# The inducible amphisome isolates viral hemagglutinin and defends against influenza A virus infection

Jumpei Omi [1], Miho Watanabe-Takahashi[1], Katsura Igai[2], Eiko Shimizu[1], Ching-Yi Tseng[1], Tomohiro Miyasaka[3], Tsuyoshi Waku[4], Shinichiro Hama[1], Rieka Nakanishi[1], Yuki Goto[1], Yuri Nishino[5], Atsuo Miyazawa[5], Yasuhiro Natori[6], Makoto Yamashita[7] & Kiyotaka Nishikawa[1]*

The emergence of drug-resistant influenza type A viruses (IAVs) necessitates the development of novel anti-IAV agents. Here, we target the IAV hemagglutinin (HA) protein using multivalent peptide library screens and identify PVF-tet, a peptide-based HA inhibitor. PVF-tet inhibits IAV cytopathicity and propagation in cells by binding to newly synthesized HA, rather than to the HA of the parental virus, thus inducing the accumulation of HA within a unique structure, the inducible amphisome, whose production from the autophagosome is accelerated by PVF-tet. The amphisome is also produced in response to IAV infection in the absence of PVF-tet by cells overexpressing ABC transporter subfamily A3, which plays an essential role in the maturation of multivesicular endosomes into the lamellar body, a lipid-sorting organelle. Our results show that the inducible amphisomes can function as a type of organelle-based anti-viral machinery by sequestering HA. PVF-tet efficiently rescues mice from the lethality of IAV infection.

[1] Department of Molecular Life Sciences, Graduate School of Life and Medical Sciences, Doshisha University, Kyoto 6100394, Japan. [2] Department of International Health, Institute of Tropical Medicine, Nagasaki University, Nagasaki 8528523, Japan. [3] Department of Neuropathology, Graduate School of Life and Medical Sciences, Doshisha University, Kyoto 6100394, Japan. [4] Department of Genetic Code, Graduate School of Life and Medical Sciences, Doshisha University, Kyoto 6100394, Japan. [5] Graduate School of Life Science, University of Hyogo, Hyogo 6781297, Japan. [6] Department of Health Chemistry, School of Pharmacy, Iwate Medical University, Iwate 0208505, Japan. [7] Department of Microbiology and Immunology, Institute of Medical Science, University of Tokyo, Tokyo 1088639, Japan. *email: knishika@mail.doshisha.ac.jp

nfluenza A virus (IAV) is a major pathogen responsible for three to five million severe respiratory infections and 250,000–500,000 fatalities worldwide annually[1,2]. The most common currently used therapeutic agents are inhibitors of neuraminidase (NA), a viral coat protein that is responsible for the release of newly synthesized virus from the plasma membrane of infected cells[3–5]. The recent emergence and spread of IAV strains that are resistant to NA inhibitors makes it necessary to establish therapeutic strategies that target other IAV replication processes[6].

Hemagglutinin (HA) is a viral coat protein that is responsible for the binding of IAV to target cells and the subsequent fusion of the viral and cellular membranes. HA functions as a homo-trimer; each subunit binds to one molecule of sialic acid, a cell-surface receptor that is generally present at the terminus of a sugar chain of glycolipids or glycoproteins[7,8]. Thus, the HA trimer can bind three sialic acid molecules through a multi-valent interaction, sometimes referred to as the clustering effect, which increases its binding affinity for its receptor by thousands fold[9–11]. Consequently, several HA trimers on the virus particle simultaneously bind to receptors to further strengthen the clustering effect[9–11]. Although a series of HA inhibitors have been reported[12–14], none of them has been screened for a clustering effect on its direct binding to the HA trimer.

Previously, we developed a method to screen multivalent peptide libraries to identify tetravalent peptides that can directly bind to a target molecule that functions with a clustering effect, such as Shiga toxin (Stx), a major virulence factor of enter-ohemorrhagic *Escherichia coli*[15]. Stx has a B-subunit pentamer that markedly increases its binding affinity for cell-surface receptors through a multivalent interaction. We identified tetra-valent peptides that bound to the B-subunit pentamer of Stx with high affinity and inhibited the toxicity of Stx in vitro and in vivo[15–17]. Furthermore, we established a technique to use information obtained from the multivalent peptide library screen to synthesize hundreds of potentially target-specific tetravalent peptides on a cellulose membrane[18]. By screening the membrane, we identified a series of selective Stx neutralizers that can dis-tinguish among closely related Stx subtypes[19]. Based on those results, we hypothesized that the combination of the multivalent peptide library screen and the cellulose membrane-based synth-esis of candidate peptides could be used to identify highly specific neutralizers against IAV HA.

After IAV infection, HA is newly synthesized in the endo-plasmic reticulum (ER) and then glycosylated and palmitoy-lated in the Golgi[20,21]. In the trans-Golgi network (TGN) compartments, HA is recruited to the cholesterol-rich mem-brane domain, generally referred to as a lipid raft, and then transported to the apical surface of the plasma membrane via vesicular trafficking[22–24]. Some intracellular vesicles, such as late endosomes, can fuse with autophagosomes to form the amphisome, an intermediate organelle observed in various type of cells, which subsequently fuses with the lysosome, leading to the degradation of the vesicular contents[25–27].

Alveolar type-II epithelial cells (AT-II cells) are major target cells of IAV infection[28]. Multivesicular endosomes can fuse with the autophagosome in AT-II cells to produce the amphi-some, which subsequently matures into the lamellar body, a lipid-storing organelle that supplies alveolar surfactant[29–31]. During the maturation process, one member of the ATP-binding cassette (ABC) transporter family, ABCA3, has an essential role by transporting disaturated phosphatidylcholine, cholesterol, and sphingolipid into the amphisome[32]. Recently, it was reported that in AT-II cells in mice, IAV alters the mor-phology of the lamellar body to give it a more electron-dense and disrupted structure[33,34]. The involvement of HA in the formation of the altered lamellar body structure and in the pathological role of the amphisome during IAV infection has not been elucidated.

In this study, we use multivalent peptide library screens combined with membrane-based synthesis and screening of candidate multivalent peptides to identify PVF-tet, a peptide-based HA inhibitor with high affinity for HA based on the clustering effect. PVF-tet inhibits IAV replication by sequestering HA into amphisomes, whose production is enhanced by PVF-tet. In the absence of PVF-tet, the overexpression of ABCA3 in response to IAV infection induces amphisomes with similar anti-IAV properties. Thus, the inducible amphisome functions as a type of anti-IAV defense machinery by sequestering HA. We confirm the anti-IAV action of the amphisome in AT-II cells in mice.

## Results

**Peptide library screens identified a specific HA inhibitor.** Previously, we developed a library of tetravalent peptides that exhibit the clustering effect[15]. The tetravalent peptides have a polylysine core and four randomized peptides connected to the core via a spacer (Fig. 1a). We screened the tetravalent peptide library for the ability to bind with high affinity to recombinant HA from IAV H1N1 strain A/Puerto Rico/8/1934 (PR8) but not to a related mutant HA with a Leu to Ala amino-acid substitution at residue 194 (HA L194A), which results in complete abolish-ment of the receptor-binding activity[35]. For the first round of selections, Lys was selected at all variable positions, followed by Val at positions 1 and 2, and Gly at position 4, respectively (Fig. 1a). Based on that, we constructed a second set of tetravalent peptide libraries with fixed Gly, Lys, or Val at position 4 in an expanded variable region and screened them for HA-binding affinity to further refine the peptide selections. Screening of the peptide libraries with fixed Lys or Val at position 4 in the variable region identified strongly selected amino acids at all of the vari-able positions. Based on those results, we synthesized tetravalent peptides with the same core structure and the motifs HHTKRRR or RRRVNHH (TKR-tet or RVH-tet, respectively) and examined them for inhibitory activity against IAV cytopathicity in Madin-Darby canine kidney (MDCK) cells. RVH-tet efficiently inhibited the cytopathicity in a dose-dependent manner and to a greater degree than TKR-tet (Fig. 1b).

To further improve the anti-IAV activity of RVH-tet, we synthesized tetravalent peptides on a cellulose membrane based on the HA-binding motif of RVH-tet. We optimized the structure of the membrane-attached tetravalent peptides for HA (Fig. 1c). After two steps of affinity maturation-based screening, we identified five HA-binding motifs: RRPVNHD, RRPMNHH, RRPVNHN, RRPVNHF, and RRPVNHP (Fig. 1c, and Supple-mentary Figs. 1a and 1b). We then synthesized tetravalent peptides with those motifs, referring to them as PVD-tet, PMH-tet, PVN-tet, PVF-tet, and PVP-tet, respectively, and tested them for the ability to inhibit HA function. All of the tetravalent peptides efficiently bound to HA and inhibited the interaction between HA and α2,3-sialyllactose polymer (also α2,6-sialyllac-tose polymer), a molecular mimic of the HA receptor, indicating that the peptides directly bind to the functional receptor-binding region of HA, which includes L194 (Fig. 1d, e, Supplementary Fig. 1c). Among the tetravalent peptides, PVF-tet had the greatest inhibitory effect on IAV cytopathicity with comparable efficacy to zanamivir, a representative NA inhibitor (Fig. 1f). In contrast, MA-tet, which has the same core structure but lacks any HA-binding motifs, did not inhibit the cytopathicity. None of these compounds showed cytotoxicity up to 60 μM (Supplementary Fig. 1d).

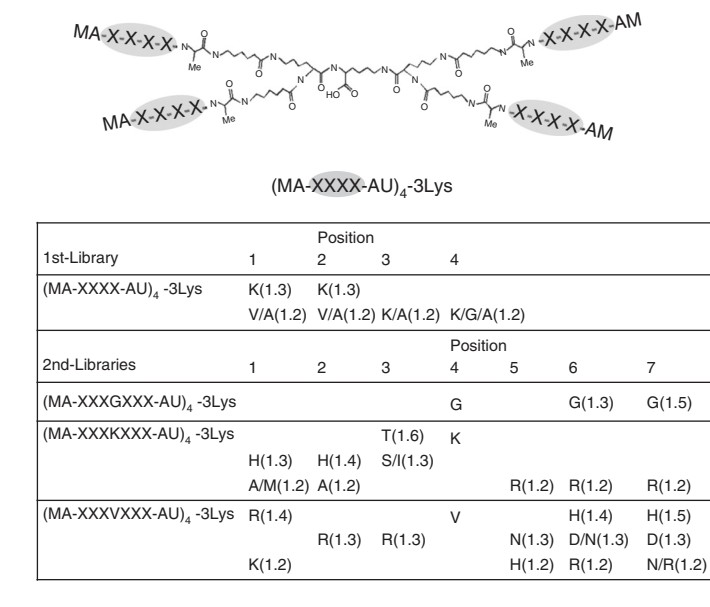

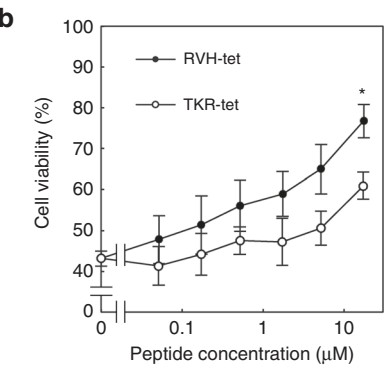

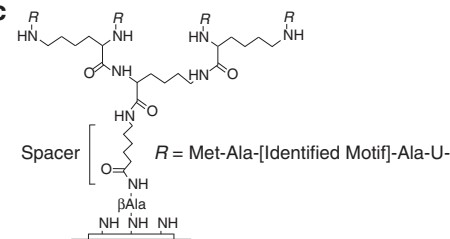

**a**

(MA-XXXX-AU)₄-3Lys

| 1st-Library | Position | | | |
|---|---|---|---|---|
| | 1 | 2 | 3 | 4 |
| (MA-XXXX-AU)₄ -3Lys | K(1.3) | K(1.3) | | |
| | V/A(1.2) | V/A(1.2) | K/A(1.2) | K/G/A(1.2) |

| 2nd-Libraries | Position | | | | | | |
|---|---|---|---|---|---|---|---|
| | 1 | 2 | 3 | 4 | 5 | 6 | 7 |
| (MA-XXXGXXX-AU)₄ -3Lys | | | | G | | G(1.3) | G(1.5) |
| (MA-XXXKXXX-AU)₄ -3Lys | | | T(1.6) | K | | | |
| | H(1.3) | H(1.4) | S/I(1.3) | | | | |
| | A/M(1.2) | A(1.2) | | | R(1.2) | R(1.2) | R(1.2) |
| (MA-XXXVXXX-AU)₄ -3Lys | R(1.4) | | | V | | H(1.4) | H(1.5) |
| | | R(1.3) | R(1.3) | | N(1.3) | D/N(1.3) | D(1.3) |
| | K(1.2) | | | | H(1.2) | R(1.2) | N/R(1.2) |

**c**

Spacer    R = Met-Ala-[Identified Motif]-Ala-U-

| Identified motifs | Tetravalent form |
|---|---|
| RRPVNHD | PVD-tet |
| RRPMNHH | PMH-tet |
| RRPVNHN | PVN-tet |
| RRPVNHF | PVF-tet |
| RRPVNHP | PVP-tet |

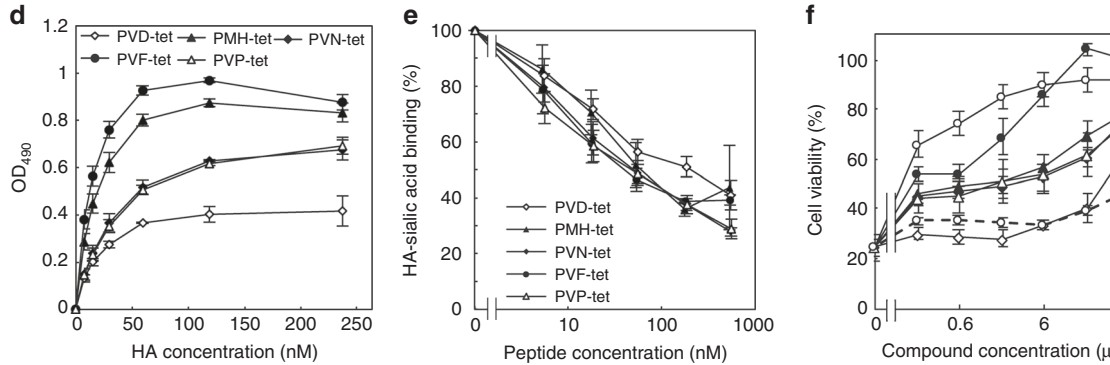

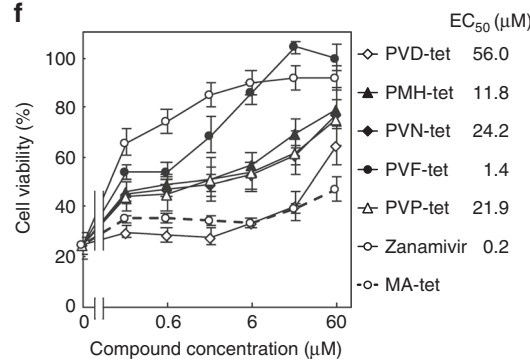

**f**

EC₅₀ (μM)

| | |
|---|---|
| PVD-tet | 56.0 |
| PMH-tet | 11.8 |
| PVN-tet | 24.2 |
| PVF-tet | 1.4 |
| PVP-tet | 21.9 |
| Zanamivir | 0.2 |
| MA-tet | |

**Fig. 1 Multivalent peptide library screen identified peptide-based HA inhibitors. a** The tetravalent peptide library for the first screen comprised tetravalent compounds with four randomized peptides of sequence Met-Ala-X-X-X-X-Ala-U (U; amino hexanoic acid), where X indicates any amino-acid except Cys. The library was screened to identify compounds that bound to HA but not to HA L194A. Three similar libraries with randomized peptides of sequence Met-Ala-X-X-X-(Gly/Lys/Val)-X-X-X-Ala-U were used for the second round of selections. Values in parentheses indicate the relative selectivity of the compound with indicated amino acids at the indicated positions. Each screen was performed twice; representative values are shown. **b** Two tetravalent peptides reduced IAV cytopathicity in MDCK cells. MDCK cells were infected with IAV strain PR8 at 10 MOI for 24 h in the presence of each peptide. Data are presented as a percentage of the control value without infection. *$P < 0.05$ (compared with TKR-tet by two-sided Student's $t$ test). **c** Structure of the tetravalent peptides synthesized on a cellulose membrane and optimized to HA. The density of the tetravalent peptide was set to 100% by using Fmoc-β-Ala-OH without Boc-β-Ala-OH for the first peptide synthesis cycle, and the number of amino hexanoic acid residues (U; spacer length) was set to one. Five HA-binding motifs identified by two steps of affinity maturation screening of the membranes are shown (lower panel). **d** ELISA to measure the binding of each tetravalent peptide (2 μM) to recombinant HA. **e** AlphaScreen assay to examine the inhibitory effect of tetravalent peptides on the binding between HA and its receptor mimic, α2–3-sialyllactose-PAA. Data are presented as a percentage of the control value. **f** The effects of tetravalent peptides and zanamivir on the IAV cytopathicity. MDCK cells were incubated with each compound for 30 min, and then infected with IAV strain PR8 at 10 MOI for 24 h. Data are presented as a percentage of the control value without infection. All data are from three independent experiments **b** and **d–f**, mean ± SEM). Source data are provided as a Source Data file.

**PVF-tet functions based on the clustering effect**. We examined the kinetics of the binding of PVF-tet or its monomer peptide (MARRPVNHFA; PVF-mono) to HA using the BIAcore system. PVF-tet, but not PVF-mono, bound to HA with high affinity (Fig. 2a). PVF-tet, but not PVF-mono, induced the formation of highly clustered HA (Supplementary Fig. 2), indicating that PVF-tet induces HA oligomer formation through intramolecular and intermolecular interactions. Consistently, PVF-mono showed no inhibitory effect on IAV cytopathicity when compared with PVF-tet (Fig. 2b), indicating that PVF-tet binds to HA and exerts its anti-IAV activity based on the clustering effect.

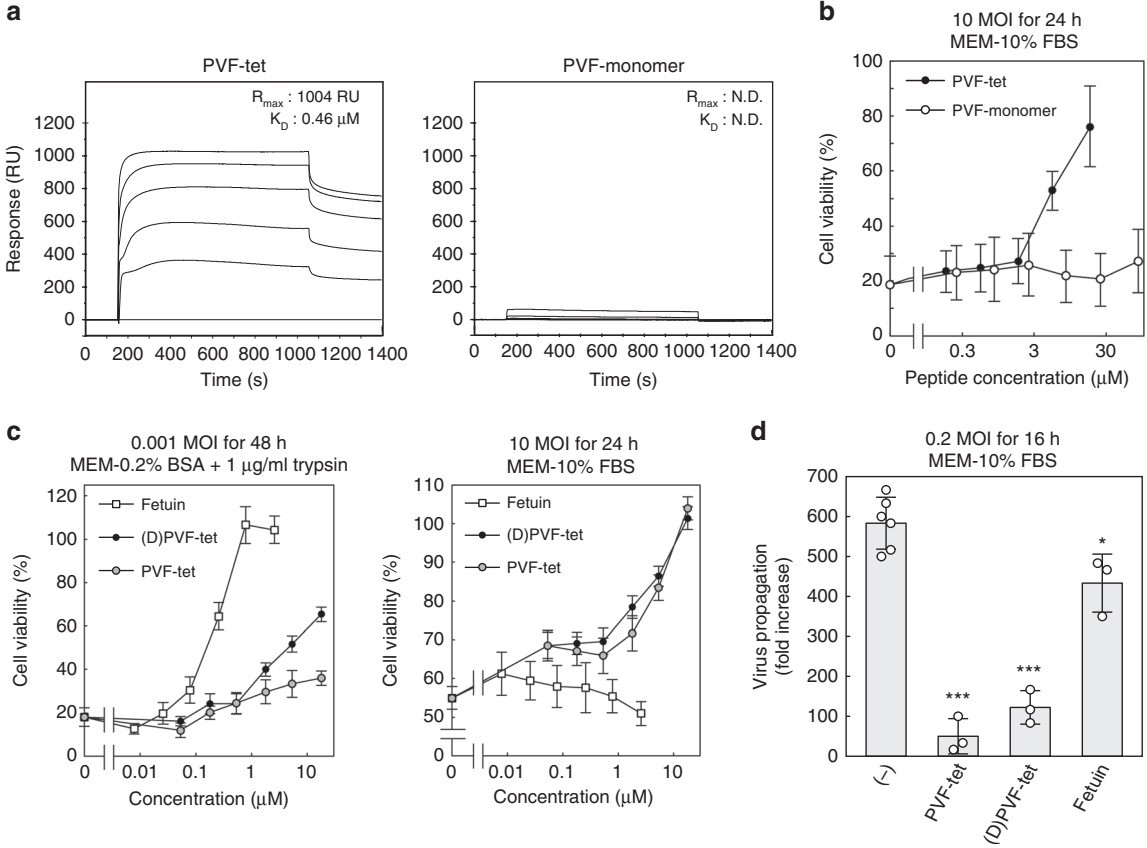

**Fig. 2 PVF-tet functions based on the clustering effect, especially under a high MOI. a** Kinetics of the binding of PVF-tet or its monomer peptide to immobilized HA analyzed using the BIAcore system. The resonance unit (RU) is an arbitrary unit used by the BIAcore system to indicate peptide binding. **b** The effects of PVF-tet or its monomer peptide on the cytopathicity induced by IAV infection. MDCK cells were incubated with the indicated concentrations of each compound for 30 min, and then infected with IAV strain PR8 at 10 MOI for 24 h in the presence of the indicated amounts of each peptide. Data are presented as a percentage of the control value without infection (mean ± SEM of three independent experiments). **c** The effects of fetuin or each tetravalent peptide on the cytopathicity induced by IAV infection. MDCK cells were incubated with the indicated concentrations of each compound for 30 min, and then infected with IAV strain PR8 at 0.001 MOI in the presence of 1 μg/ml trypsin for 48 h (left panel) or 10 MOI in the absence of trypsin for 24 h (right panel). Data are presented as a percentage of the control value without infection (mean ± SEM of three independent experiments). **d** The effect of fetuin (7.8 μM) or each tetravalent peptide (60 μM) on virus propagation after IAV infection. MDCK cells were incubated with the indicated concentrations of each compound for 30 min, and then infected with IAV strain PR8 at 0.2 MOI for 16 h. The virus titer in the supernatant was determined by a plaque assay. Data are presented as a fold increase over the initial virus titer (60,000 pfu/ml) (mean ± SD of 3–6 independent experiments). *$P < 0.05$; ***$P < 0.001$ (compared with no compound (−) treatment by ANOVA followed by one-sided Dunnett's test). Experimental details are shown above each panel. Source data are provided as a Source Data file.

Under a multi-cycle infection, i.e., a low multiplicity of infection (MOI), intact HA (HA0) on virus particles released from infected cells must be cleaved into two fragments, HA1 and HA2, by trypsin present in the culture medium, which are connected to each other via a disulfide bond (HA1/HA2). The formation of HA1/HA2 enables the virus to fuse with the endosome membrane after endocytosis by the target cells[36,37]. To avoid the proteolytic degradation of PVF-tet under low MOI (0.001 MOI), we synthesized (D)PVF-tet, a trypsin-resistant form of PVF-tet in which the first Arg of the motif is substituted with D-Arg (the enantiomer). (D)PVF-tet was completely resistant to trypsin, whereas PVF-tet was easily cleaved by trypsin (Supplementary Fig. 3).

We compared the anti-IAV activity of PVF-tet with that of fetuin, a highly sialylated glycoprotein known to effectively bind to HA and inhibit IAV infection[38,39]. (D)PVF-tet inhibited IAV cytopathicity to a greater degree than PVF-tet at 0.001 MOI, although fetuin had the greatest inhibitory effect (Fig. 2c, left panel). In stark contrast, at 10 MOI both PVF-tet and (D)PVF-tet markedly inhibited the cytopathicity, whereas fetuin showed no inhibitory

effect at all (Fig. 2c, right panel). Accordingly, PVF-tet and (D)PVF-tet both inhibited IAV propagation to a greater degree than fetuin (Fig. 2d). Thus, the anti-IAV activity of PVF-tet has a different mechanism than that of fetuin, which inhibits virus entry into cells by competing with the cell-surface receptor for HA binding.

**PVF-tet sequesters newly synthesized HA into a unique structure.** PVF-tet did not inhibit virus entry into target cells, as indicated by the percentage of cells expressing viral nucleoprotein (NP) at an early stage of IAV infection (Fig. 3a). In contrast, bafilomycin A1, which inhibits the vacuolar H⁺-ATPase and suppresses the fusion between the viral and cellular membranes after endocytosis of the virus, completely inhibited virus entry into cells. Consistently, PVF-tet did not affect the fusion process (as monitored by fluorescently labeled virus) or the following NP mRNA synthesis, whereas bafilomycin A1 efficiently inhibited both events (Supplementary Figs. 4a and 4b). Furthermore, PVF-tet treatment inhibited IAV cytopathicity even when administered up to 9 h after initial IAV infection, while

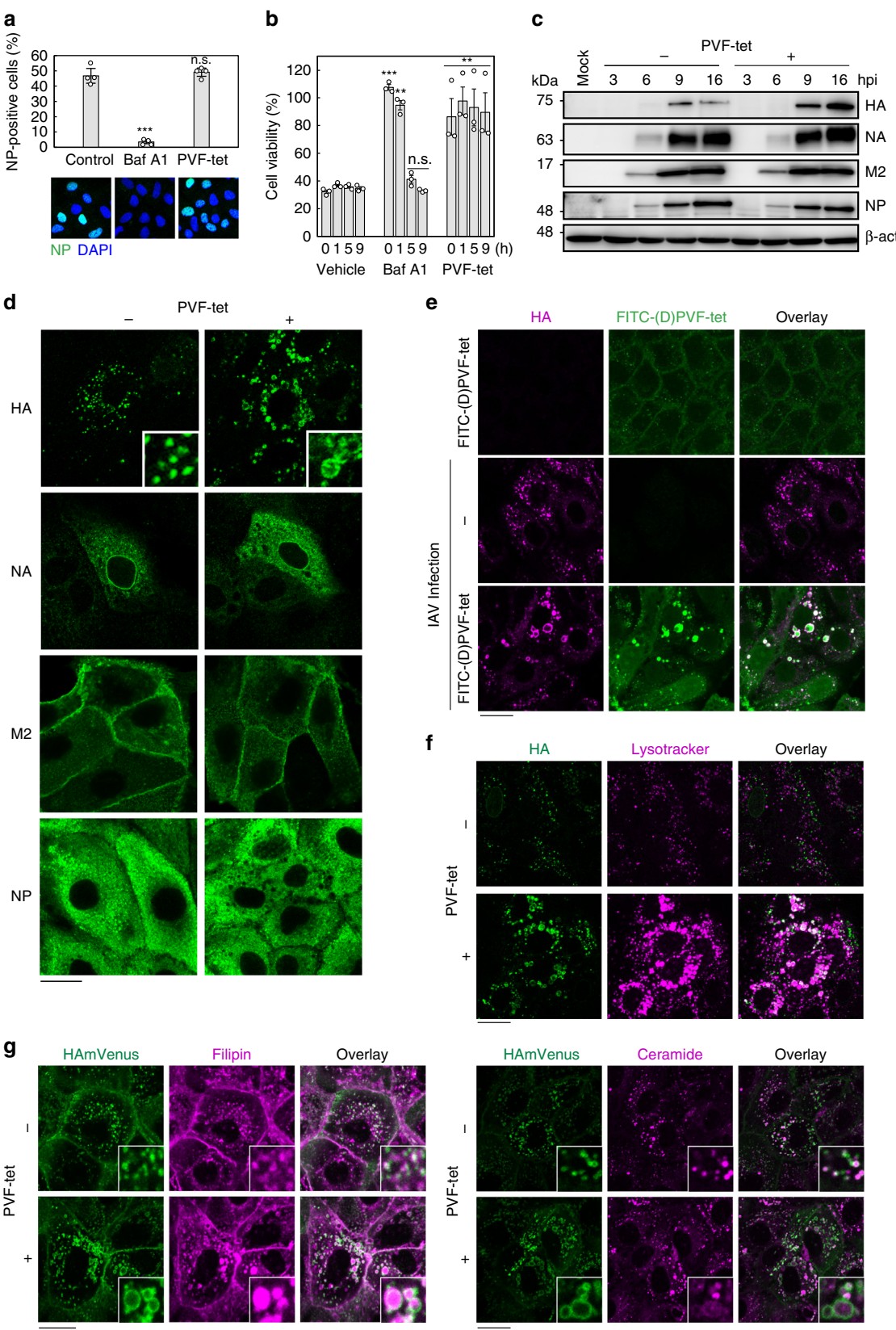

bafilomycin A1 could not inhibit the cytopathicity when administered 5 h after infection (Fig. 3b). These results indicate that PVF-tet functions in a later stage of infection.

On the basis of these results, it is possible that PVF-tet does not target HA present on infectious IAV particles. In contrast to fetuin, PVF-tet did not inhibit IAV-induced hemagglutination, supporting that hypothesis (Supplementary Fig. 4c). Furthermore, we found that PVF-tet did not bind to infectious IAV particles with HA1/HA2, but it did bind with remarkable efficacy to non-infectious IAV particles with HA0 (Supplementary Fig. 5). That

**Fig. 3 PVF-tet targets newly synthesized HA and sequesters HA into a vacuole-like structure. a** The effects of PVF-tet on IAV entry into target cells. MDCK cells were infected with IAV strain PR8 at 2 MOI in the presence of Bafilomycin A1 (Baf A1, 100 nM) or PVF-tet (20 μM) for 3 h. Intracellular NP was detected by immunocytochemistry. Percentage of NP-positive cells was measured (mean ± SD of four images acquired at random). $***P < 0.001$ (compared with untreated cells by ANOVA followed by one-sided Dunnett's test). n.s., not significant. **b** MDCK cells were infected with IAV strain PR8 at 10 MOI for 1 h at 4 °C to synchronize the IAV binding to the cells. After extensive washing, the cells were cultured at 37 °C, and then treated with 20 μM PVF-tet, 100 nM Baf A1, or vehicle at the indicated time after washing. The relative cell viability was examined 24 h after infection. Data are presented as a percentage of the control value (mean ± SEM of three independent experiments). $**P < 0.01$; $***P < 0.001$ (compared with vehicle treatment by ANOVA followed by one-sided Dunnett's test). n.s., not significant. **c–g** MDCK cells **c–f** or MDCK cells stably expressing HAmVenus **g** were infected with IAV strain PR8 at 10 MOI for 1 h. After washing, the cells were cultured in the presence or absence of 20 μM PVF-tet **c**, **d**, **f**, **g** or 2.5 μM FITC-labeled (D)PVF-tet **e**. The expression of each viral protein or β-actin in the lysates at the indicated times was analyzed by western blot. Data are representative of three independent experiments **c**. The intracellular localization of each viral protein **d**, and colocalization of HA and FITC-labeled (D)PVF-tet **e** were analyzed by immunocytochemistry. Acidification of the HA-containing vacuole-like structure was analyzed using Lysotracker **f**. Colocalization of fluorescent-labeled HA (HAmVenus) with cholesterol or sphingolipid was examined using filipin (left panel) or BODIPY-TR-C5-Ceramide (right panel), respectively **g**. Immunocytochemistry was performed 16 h after infection **d–g**. Insets show magnified fields. Scale bars represent 20 μm. Source data including uncropped western blot images are provided as a Source Data file.

stark difference can be attributed to the difference of the atomic fluctuation level, represented by the crystallographic B-factor, in the receptor-binding sites of HA0 and HA1/HA2. The normalized B-factors in the receptor-binding sites were higher in HA0 compared with HA1/HA2, allowing PVF-tet better access to the site in the HA0 conformation (Supplementary Fig. 6). Despite this observation, more direct evidence for the difference in fluctuation level remains to be obtained. Nevertheless, our results suggest that PVF-tet is more likely to target newly synthesized HA, which is produced as membrane-associated HA0 in infected cells.

We examined the effects of PVF-tet on the expression of newly synthesized viral proteins. Only HA, and not the other viral proteins, accumulated in a time-dependent manner in infected cells after PVF-tet treatment; in the absence of PVF-tet, the amount of HA in infected cells peaked at 9 h after infection and subsequently declined (Fig. 3c). We consistently observed the accumulation of HA in a vacuole-like structure following PVF-tet treatment, whereas the intracellular localization of the other viral proteins was not affected by PVF-tet treatment (Fig. 3d). In uninfected cells, fluorescein isothiocyanate (FITC)-labeled (D) PVF-tet localized mainly at the plasma membrane and partially within the cytosolic region as punctate spots. However, in IAV-infected cells, (D)PVF-tet colocalized with HA in the vacuole-like structure (Fig. 3e), suggesting that (D)PVF-tet formed a complex with newly synthesized HA. The formation of the vacuole-like structures, along with concurrent increases in cell viability, was clearly observed even when the cells were treated with (D)PVF-tet after the establishment of IAV infection (Supplementary Figs. 7a and 7b). Furthermore, PVF-tet induced the accumulation of HA within the vacuole-like structures, resulting in a marked reduction of the amount of HA that was present on the plasma membrane and available for virus production (Supplementary Fig. 7c). Those observations suggest that (D)PVF-tet made a complex with newly synthesized HA and then localized along with HA inside the vacuole-like structure, thus inhibiting the production of virus particles.

PVF-tet and the other tetravalent peptides, except for PVD-tet, can be incorporated into MDCK cells at 37 °C (Supplementary Fig. 8a). PVF-tet contains eight Arg residues in its structure and was incorporated into cells via the endocytic pathway (Supplementary Fig. 8b), which is consistent with previous studies showing that monovalent or polyvalent peptides with poly-Arg motifs have high potency to penetrate into cells via the endocytic pathway[40]. The less cell-penetrating feature of PVD-tet, which is the only one of the tetravalent peptides with an Asp in its motif, might explain why only PVD-tet failed to inhibit IAV cytopathicity (Fig. 1f). Furthermore, the incorporation of

PVF-tet into cells was enhanced by the presence of fetal bovine serum (FBS) (Supplementary Fig. 8c), which explains why (D) PVF-tet was less effective under the low MOI condition (Fig. 2c), in which FBS was excluded from the culture medium to promote multi-cycle infection, than under the high MOI condition. That observation is consistent with a previous report, demonstrating that cell-penetrating peptides can be efficiently incorporated into cells in the presence of FBS[41]. In Caco-2 cells, where IAV can efficiently replicate even in the presence of 10% FBS[42], PVF-tet equally inhibited cytopathicity induced by both low and high MOI infection (Supplementary Fig. 8d), indicating that the anti-viral activity of PVF-tet is independent of the infectious MOI. Thus, the cell-penetrating feature of PVF-tet substantially contributes to the ability of PVF-tet to suppress virus production through the formation of HA-containing vacuole-like structures.

We performed a cytochemical analysis of the HA-containing vacuole-like structure induced by PVF-tet treatment. We found that lysosome-associated membrane protein 1 (LAMP1) partially localized within the structure, whereas EEA1 (an early endosome marker), HSP47 (an ER marker), calnexin (an ER marker), p230 (a TGN marker), GS28 (a TGN marker), and GM130 (a cis-Golgi network marker) did not (Supplementary Figs. 9a–g). Lysotracker (an acid-tropic dye) also accumulated within the vacuole-like structure (Fig. 3f), indicating that the structure is highly acidified. HA was not degraded within the structure, as treatment with lysosomal protease inhibitors did not affect the amount of HA inside the structure (Supplementary Fig. 9h). The PVF-tet-induced HA oligomer formation mediated by the intramolecular and intermolecular interactions (Supplementary Fig. 2) might contribute to the HA resistance to proteolytic degradation.

To further characterize the HA-containing vacuole-like structure, we used MDCK cells stably expressing fluorescent-labeled HA (HAmVenus); otherwise, the colocalization of viral HA and cellular lipids could not be analyzed because of the permeabilization process involving a detergent. Consistent with previous reports demonstrating that newly synthesized HA associates with lipid rafts containing cholesterol and sphingolipid during its vesicular transport to the apical membrane[22,43], HAmVenus colocalized with filipin (a probe for unesterified cholesterol) and with fluorescent-labeled ceramide (a probe for sphingolipid). The localization of HAmVenus was not affected by IAV infection (Fig. 3g, upper panels). After IAV infection in the presence of PVF-tet, we found that HAmVenus was isolated within the vacuole-like structure (Fig. 3g, lower panels). Treatment with PVF-tet caused those lipid probes to accumulate along with HAmVenus in the vacuole-like structure. All of those results indicate that the HA-containing vacuole-like structure is a lysosome-related organelle enriched with lipids.

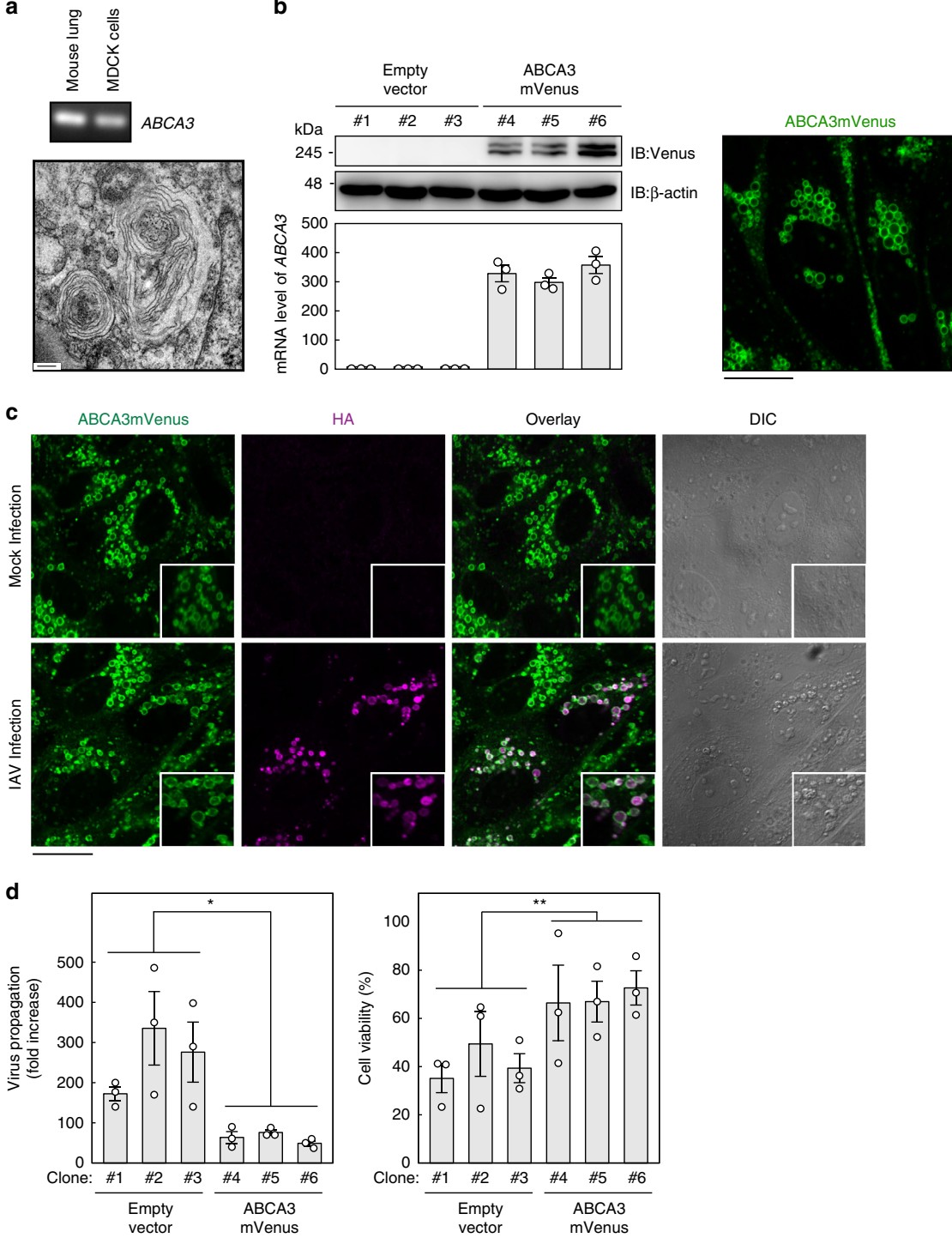

**Overproduction of the lamellar body also sequesters HA**. The lamellar body, which is generally present in AT-II cells, is a typical lysosome-related organelle enriched with lipids[44]. We hypothesized that the HA-containing vacuole-like structure that we observed following IAV infection in the presence of PVF-tet originated from the lamellar body. We could observe even small amounts of lamellar body in MDCK cells by electron microscopy, and we also detected *ABCA3* mRNA in those cells by RT-PCR (Fig. 4a).

First, we examined the effects of lamellar body overproduction on IAV infection. We established three clones from MDCK cells that stably overexpressed fluorescent-labeled ABCA3 (ABCA3mVenus)

at both the protein and the mRNA levels (Fig. 4b, left panels). Those clones displayed a large amount of lamellar body containing ABCA3mVenus (Fig. 4b, right panel). After the clones were infected with IAV, newly synthesized HA accumulated within an ABCA3mVenus-positive vacuole-like structure (Fig. 4c), resulting in inhibition of virus propagation and cytopathicity (Fig. 4d).

The characteristics of the ABCA3-positive and HA-positive structure were different from those of the lamellar body but were quite similar to those of the PVF-tet-induced HA-containing structure. Both Lysotracker and filipin accumulated in the ABCA3-positive and HA-positive structure but not in the lamellar body (Supplementary Fig. 10). Furthermore, differential

**Fig. 4 Overproduction of the lamellar body inhibits virus propagation by sequestering newly synthesized HA. a** Relative mRNA level of *ABCA3* in mouse lung and in MDCK cells analyzed by RT-PCR. The micrograph shows a typical image of a lamellar body detected in MDCK cells by electron microscopy. **b** Expression levels of ABCA3mVenus in MDCK-derived clones stably overexpressing ABCA3mVenus (clones 4–6) or transfected with empty vector (clones 1–3) analyzed by western blot using specific antibodies (IB; left upper panel). Upper and lower bands correspond to uncleaved and N-terminally cleaved forms of ABCA3mVenus, respectively. The mRNA level of ABCA3 in each clone was analyzed by quantitative RT-PCR (left lower panel). Data are presented as the relative amount of ABCA3 mRNA compared with the average of the three control clones (clones 1–3). Intracellular localization of ABCA3mVenus in clone 6 was analyzed by confocal microscopy (right panel). Similar images were obtained in clones 4 and 5. **c** Colocalization of HA with ABCA3 in clone 6. Clone 6 was not infected or infected with IAV strain PR8 at 10 MOI (lower panel) for 1 h. After washing, the cells were cultured for 15 h. Fluorescent images were analyzed using confocal microscopy. **d** The effects of ABCA3 overproduction on virus propagation and viral cytopathicity. Each MDCK clone was infected with IAV strain PR8 at 0.2 MOI for 16 h. The virus titer in the supernatant was determined by a plaque assay (left panel). Data are presented as a fold increase over the initial virus titer (60,000 pfu/ml). *$P < 0.05$ (by two-sided Student's $t$ test). Each MDCK clone was infected with IAV strain PR8 at 10 MOI for 1 h. After washing, the cells were cultured for 19 h (right panel). Cell viability was measured by cytopathicity assay. Data are presented as a percentage of the control value without infection. **$P < 0.01$ (by two-sided Student's $t$ test). All data are from three independent experiments (**b** and **d**, mean ± SEM). Scale bars represent 200 nm **a** or 20 μm **b**, **c**. Source data including uncropped western blot images are provided as a Source Data file.

interference contrast images of the lamellar body and of the ABCA3-positive and HA-positive structure were clearly different (Fig. 4c). Those results suggested that the ABCA3-positive and HA-positive structure and also the PVF-tet-induced HA-containing structure matured from the lamellar body.

To investigate that possibility, we examined the effects of *ABCA3* gene knockout on the anti-IAV activity of PVF-tet. Unexpectedly, PVF-tet retained its anti-IAV activity and continued to cause the accumulation of newly synthesized HA in the *ABCA3*-knockout clone (Supplementary Figs. 11a–c). The PVF-tet-induced HA accumulation in the *ABCA3*-knockout cells occurred in a highly acidified structure, although the average size of the HA-containing structure was clearly reduced compared with that in wild-type cells (Supplementary Fig. 11d), suggesting that ABCA3 is involved in the maturation process of the HA-containing structure but not in the initial formation of the structure.

The results indicated that our observations involved two independent but closely related machineries that inhibit IAV propagation. One is the PVF-tet-induced HA-containing vacuole-like structure, whose anti-IAV activity does not essentially depend on ABCA3. The other is an HA-containing and ABCA3-containing structure originated from the lamellar body, which can be produced without PVF-tet.

**The HA-containing structures are classified as amphisomes**. We performed electron microscopic analysis to further elucidate the morphological characteristics of the HA-containing structures. Correlative fluorescence and electron microscopy clearly showed that the HA-containing structure induced by PVF-tet had not only lamellar membranous features but also electron-dense core features (Fig. 5a). In the MDCK clone overexpressing ABCA3, IAV infection induced the formation of a structure with similar features, whereas uninfected cells displayed a smooth lamellar structure (Fig. 5b).

All of the features of the HA-containing structures are consistent with those of the amphisome, which is formed by the fusion of various endosomes with autophagosomes[31,45]. First, to determine whether the PVF-tet-induced HA-containing structure is an amphisome, we examined the activation of microtubule associated protein 1 light-chain 3 (LC3-I), which can be processed to its activated form LC3-II to enable the formation of the autophagosome. As previously reported[46–51], IAV infection caused the accumulation of LC3-II (Fig. 6a), but the localization of HA did not match well with that of LC3 (Fig. 6b), indicating that the autophagosome induced by IAV infection and the HA-containing transport vesicle are different structures.

PVF-tet treatment further enhanced the formation of LC3-II and induced the accumulation of LC3 in the PVF-tet-induced HA-containing structure (Fig. 6a, b). Under that condition, treatment with protease inhibitors did not affect the amount of LC3-II in the cells, suggesting that the proteolytic degradation of LC3-II does not occur in the HA-containing compartment (Fig. 6a). Those results are consistent with the previous observation that PVF-tet did not cause the degradation of HA in the HA-containing structure (Fig. 3c and Supplementary Fig. 9h). In the MDCK clone overexpressing ABCA3, LC3 clearly accumulated in the ABCA3-positive structure after IAV infection (Supplementary Fig. 12a). Another prototypic autophagosome marker, p62/sequestosome 1, was partially accumulated in the HA-containing structure induced by PVF-tet treatment (Supplementary Fig. 12b). Furthermore, dextran, which is widely known to be incorporated into cells via macropinocytosis and transported to endosomes, was efficiently incorporated into the HA-containing structure (Supplementary Fig. 12c). Because it is well known that amphisomes are formed by the fusion of endosomes with the autophagosome containing LC3-II and p62, these results indicate that the two types of HA-containing structures can be classified as amphisomes.

To confirm that assertion, we examined the effects of gene knockout of *PIK3C3*, which encodes a protein involved in the nucleation of pre-autophagosomal structures[52], and *ULK1*, which encodes a direct activator of PIK3C3 complex[53,54], on the anti-viral activity of PVF-tet. The MDCK-derived knockout clones of these genes were confirmed to have suppressed protein expression (Fig. 6c left panels) and resistance to starvation-induced LC3 (+)-puncta formation (Supplementary Fig. 13a). In these clones, the formation of an HA-containing vacuole-like structure induced by PVF-tet was markedly inhibited (Fig. 6c right panel), although the total amounts of newly synthesized HA were similar to that of parental cells. All of these clones were found to be less sensitive to IAV cytopathicity, consistent with previous reports indicating that activation of autophagy is required for IAV-induced apoptosis[55] (Supplementary Fig. 13b). Although IAV infection reduced the cell viabilities of these clones to 30% or less after 48 h of infection, which is comparable to the cytopathicity in parental MDCK cells 24 h after infection, the anti-viral activity of PVF-tet was significantly suppressed in the clones compared with its dose-dependent activity in parental cells (Fig. 6d). Furthermore, SAR405, a specific inhibitor of PIK3C3, also suppressed the dose-dependent anti-viral activity of PVF-tet (Supplementary Fig. 13c). In the MDCK clone overexpressing ABCA3, SAR405 inhibited the accumulation of HA in the ABCA3-positive structure after IAV infection (Supplementary Fig. 14). Thus, autophagosome formation, which is a prerequisite for amphisome

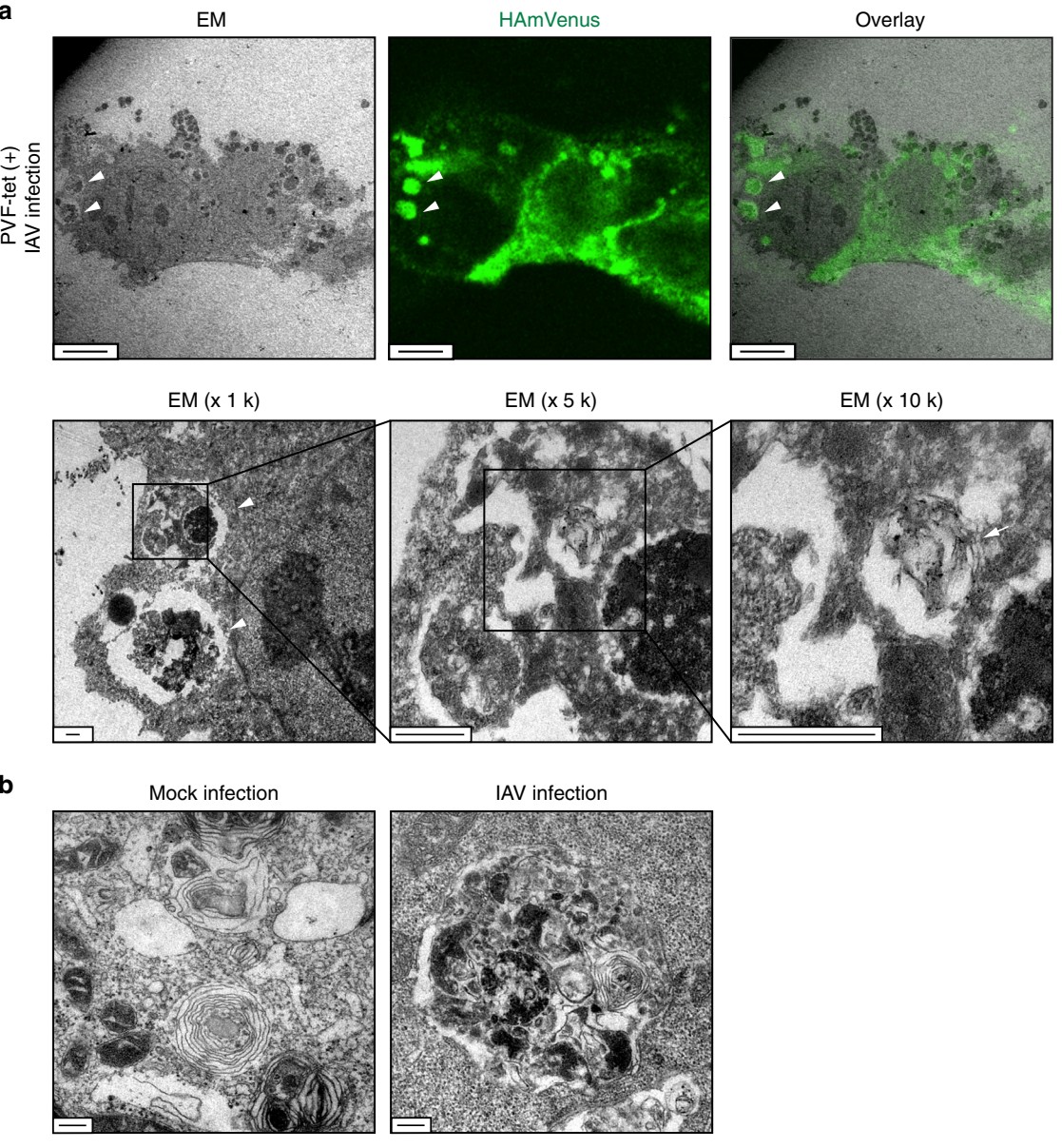

**Fig. 5 The HA-containing structures have not only lamellar membranous features but also electron-dense core features. a** Correlative fluorescence and electron microscopy (EM) of MDCK cells stably expressing HAmVenus. The cells were infected with IAV strain PR8 at 10 MOI for 1 h. After washing, the cells were cultured in the presence of PVF-tet (20 μM) for 15 h. Arrowheads indicate the HA-containing structures; the arrow in the last panel indicates the lamellar membrane. **b** Electron microscopy of MDCK clone 6. The cells were not infected or infected with IAV strain PR8 at 2 MOI for 1 h. After washing, the cells were cultured for 15 h. Scale bars represent 6 μm (**a**, upper panel), 500 nm (**a**, lower panel), or 200 nm **b**, respectively.

formation, is essential for the induction of both types of HA-containing structures with anti-IAV activities, further confirming that the structures are amphisomes.

**(D)PVF-tet protects mice from the lethal IAV infection**. Intranasal infection of BALB/c mice with the mouse-adapted IAV strain PR8 caused acute weight loss from day 3 after infection and 100% mortality within 8 days. Intranasal co-injection of 1.25 or 2.5 mg/kg (D)PVF-tet partially attenuated the weight loss and rescued 10% or 50% of mice from mortality, respectively (Fig. 7a). Under the same conditions, zanamivir attenuated the weight loss and rescued 100% of mice from mortality (Supplementary Fig. 15a). Treatment with 2.5 mg/kg (D)PVF-tet 6 h after IAV infection still partially attenuated the weight loss and rescued 50% of mice from mortality (Supplementary Fig. 15b). (D)PVF-tet inhibited the propagation of the virus in the lung both 3 days and

5 days after infection (Fig. 7b). Furthermore, (D)PVF-tet efficiently attenuated weight loss and prevented mortality in mice infected with two other mouse-adapted IAV strains, A/California/04/2009 (H1N1) and A/Aichi/2/1968 (H3N2) (Fig. 7c), whose cytopathicity in MDCK cells was also inhibited by PVF-tet through the formation of inducible amphisomes (Supplementary Fig. 16). Those results indicate that (D)PVF-tet has broad anti-IAV activity in vivo.

Recent reports have shown that the main target of IAV in the lung is AT-II cells[28]. We examined the effects of (D)PVF-tet treatment on IAV-induced damage to AT-II cells. We used the expression of Pro-surfactant protein C (SP-C) as a marker of AT-II cells. Consistent with the anti-IAV activity in mice, (D)PVF-tet inhibited the loss of Pro-SP-C-expressing cells in the alveolar region following IAV infection (Fig. 7d and Supplementary Fig. 17a), suggesting that (D)PVF-tet protects AT-II cells from

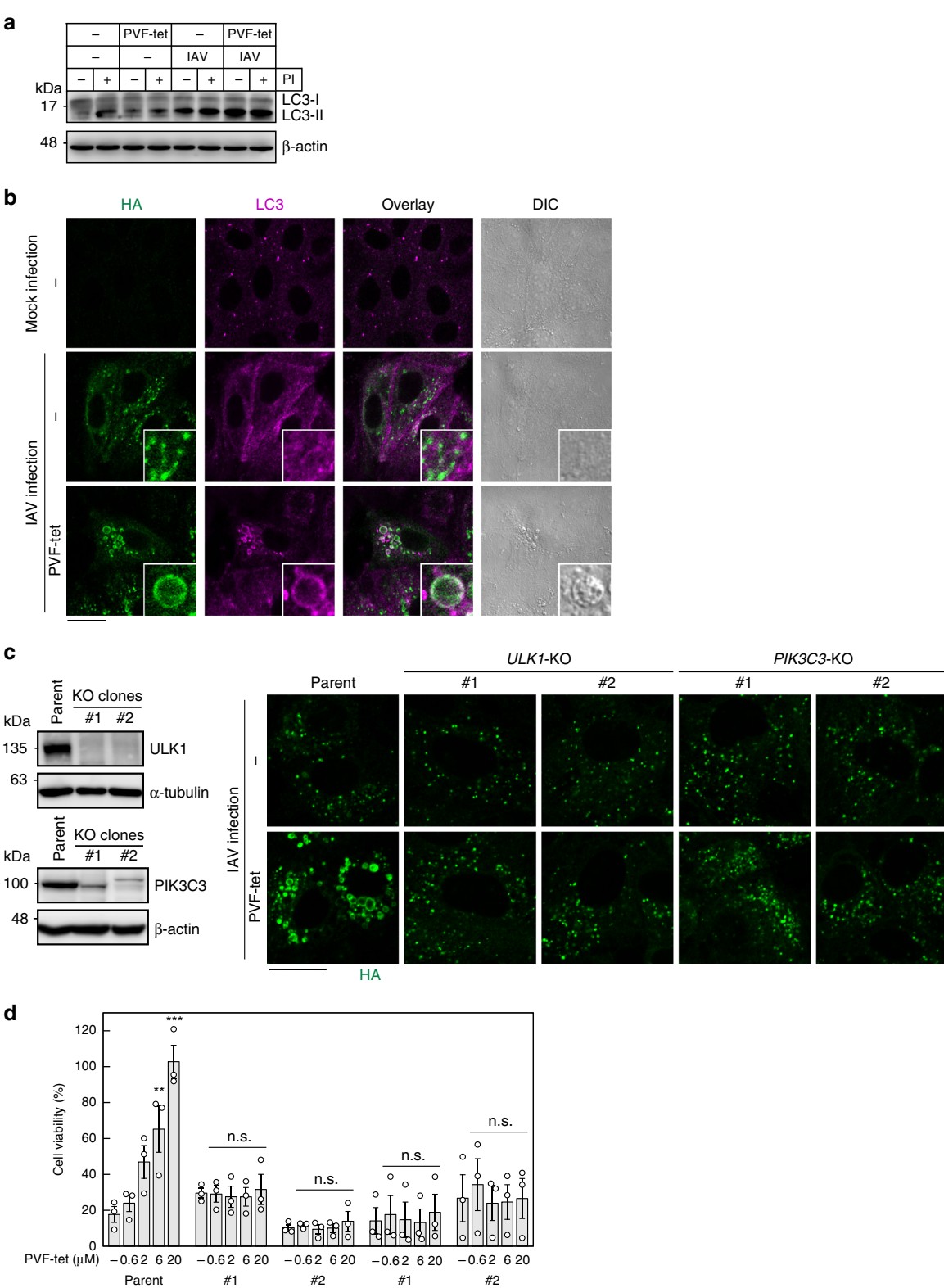

IAV cytopathicity. Furthermore, immunohistochemical analysis clearly showed that (D)PVF-tet colocalized with not only HA but also ABCA3 (Fig. 7e), which is an exclusive marker of AT-II cells in the lung, indicating that (D)PVF-tet facilitates the formation of an HA-positive and ABCA3-positive structure, a hybrid of the two types of amphisomes, to exert its anti-IAV activity in vivo. We examined the content of LC3-I and LC3-II in the cell population enriched with alveolar type-II epithelial cells isolated from IAV-infected or IAV and (D)PVF-tet co-injected mice. LC3-II production was enhanced by IAV infection, and the ratio of LC3-II to LC3-I, which correlates with the progression of autophagy, was substantially increased in IAV and (D)PVF-tet co-injected mice compared to IAV-infected mice (Supplementary Figs. 17b and 17c), confirming that (D)PVF-tet facilitates the

**Fig. 6 The HA-containing structures can be classified as amphisomes. a** Production of LC3-II and the effects of protease inhibitors. MDCK cells were infected (IAV) or not infected (−) with IAV strain PR8 at 10 MOI for 1 h. After washing, the cells were cultured in the presence or absence of 20 μM PVF-tet for 2 h. The cells were treated (+) or not treated (−) with lysosomal protease inhibitor cocktail containing 30 μM E-64, 15 μM pepstatin A, and 20 μM leupeptin and further cultured for 13 h. The cell lysates were analyzed by western blot. **b** The effect of PVF-tet on the intracellular localization of LC3. MDCK cells were not infected or infected with IAV strain PR8 at 10 MOI for 1 h. After washing, the cells were cultured in the presence or absence of PVF-tet (20 μM) for 15 h. Localization of HA and LC3 was analyzed by immunocytochemical staining. **c−d** The effects of *ULK1* or *PIK3C3* knockout on the formation of the PVF-tet-induced HA-containing structure and the anti-IAV activity of PVF-tet. The expression levels of ULK1 in MDCK-derived *ULK1*-knockout clones (**c**, left upper panel) and PIK3C3 in MDCK-derived *PIK3C3*-knockout clones (left lower panel) were analyzed by western blot using specific antibodies. Parental MDCK cells or each MDCK-derived knockout clone was infected with IAV strain PR8 at 10 MOI for 1 h. After washing, the cells were cultured in the presence of 20 μM PVF-tet for 15 h and fluorescent images were analyzed using laser scanning confocal microscopy (**c**, right panel), or cultured for 23 h (MDCK cells) or 47 h (knockout clones) and cell viability was measured by a cytopathicity assay **d**. Data are presented as a percentage of the control value without infection (mean ± SEM of three independent experiments). **P < 0.01; ***P < 0.001 (compared with untreated cells by ANOVA followed by one-sided Dunnett's test). n.s., not significant. Scale bars represent 20 μm. Source data including uncropped western blot images are provided as a Source Data file.

formation of amphisomes with HA accumulation through the autophagic machinery to exert anti-IAV activity in vivo.

## Discussion

We developed PVF-tet, a novel peptide-based HA inhibitor, by targeting the receptor-binding region of HA using a multivalent peptide library-screening method. PVF-tet markedly inhibited the cytopathicity of IAV infection by sequestering newly synthesized HA into the amphisome (Fig. 8). Even in the absence of PVF-tet, the production of amphisomes with anti-IAV activity was induced by the overexpression of ABCA3. Thus, the inducible amphisome can function as a type of anti-viral machinery.

Several peptide-based HA inhibitors have been developed. The sialic acid mimic peptide C18-s2(1–5), which was identified using a phage-display library screen by targeting the receptor-binding region of HA, inhibited the binding of IAV to target cells[12]. A cyclic peptide, which was designed based on the complementarity determining regions of anti-HA antibodies raised against the stem region of HA, inhibited the membrane-fusion process of IAV infection[14]. These HA inhibitors can inhibit IAV infection in cultured cells, but the effectiveness in vivo has not been shown. Recently, a computationally designed high-avidity trimeric protein that binds to the receptor-binding region of HA was developed to function based on the clustering effect; this molecule was found to inhibit IAV infection both in vitro and in vivo[56]. However, all of these inhibitors cannot be effective after viral entry because they target the HA of parental IAV particles to inhibit the entry of the virus into target cells. PVF-tet functions both in vitro and in vivo by targeting newly synthesized HA rather than the HA of parental virions, providing a promising strategy to regulate the propagation of IAV.

PVF-tet functions based on the clustering effect to bind and inhibit HA. PVF-mono, the monomer form of PVF-tet, did not bind to HA nor did it inhibit the cytopathicity of IAV infection (Fig. 2a, b). PVF-tet can penetrate into cells and directly bind to newly synthesized HA in the ER or the Golgi, resulting in the formation of highly clustered complexes containing intermolecularly cross-linked HA. Previously, newly synthesized HA in the ER was shown to be transported to the Golgi, and then to the apical membrane, through transport vesicles whose membranes are enriched with cholesterol and sphingolipids and which originate from the TGN[22,43,57]. Thus, it is possible that the PVF-tet-induced HA-containing vesicle is also produced from the TGN, which, in this case, is enriched with clustered complexes of PVF-tet and HA in addition to cholesterol and sphingolipids. Then, the vesicle matures into the large and highly acidified amphisome after fusion with the autophagosome.

The precise mechanism by which the vesicle is not transported to the apical membrane but instead stays in the cytosol to maturate into the amphisome remains to be elucidated. The vesicles that transport HA to the apical membrane require the assembly of sorting proteins such as Annexin XIIIb and COOH-terminal motor domain-type kinesin superfamily protein 3[23,24]. The formation of clustered complexes of PVF-tet and HA in the TGN may affect the function and/or assembly of those sorting proteins, resulting in vesicles that cannot be transported to the apical membrane.

Even in the absence of PVF-tet, the overproduction of the lamellar body induced by the overexpression of ABCA3 can induce the formation of highly acidified, lipid-enriched amphisomes that sequester HA and inhibit virus propagation. In general, newly synthesized ABCA3 in the ER is transported to the Golgi and then to the multivesicular endosome (MVE)[32,44,58]. The MVE fuses with the autophagosome to form the amphisome, which maturates to the lamellar body[29–31]. Thus, in ABCA3-overexpressing cells, which constitutively produce a large amount of lamellar body enriched with cholesterol and sphingolipid, newly synthesized HA, which is preferentially sorted to the lipid raft, can be easily trapped into the MVE and then into the amphisome, resulting in the colocalization of HA with ABCA3.

In MDCK cells, after transport of newly synthesized HA to the plasma membrane, HA has been reported to bind to sulfatide, one of the major sulfated glycolipids present in abundance in lipid rafts of the plasma membrane, and then induce apoptosis[59]. Blocking the formation of this complex with an anti-sulfatide or anti-HA antibody results in inhibition of virus amplification, suggesting that the interaction of HA with sulfatide on the plasma membrane mainly contributes to the virus-induced cell death in MDCK cells[60]. Thus, sequestration of HA induced by PVF-tet or ABCA3 overexpression can block this pathway to promote cell survival.

We found that autophagosome formation is an essential step for the induction of the anti-viral amphisome. In contrast, previous reports have shown that IAV infection induces the accumulation of LC3-II and then the autophagosome formation, and that PI3K inhibitors prevent those events and suppresses virus propagation[46,47,49–51,55], suggesting that autophagosome formation can function to aid viral propagation. We found that the autophagosome induced by IAV infection alone did not contain HA, whereas the autophagosome/amphisome induced by PVF-tet contained a large amount of HA, indicating that those two types of autophagosomes are structurally and functionally different. In addition, PVF-tet dramatically changed the intracellular localization so that LC3 was exclusively colocalized with HA after IAV infection. Thus, PVF-tet can exert anti-IAV activity not only by inducing the formation of the anti-viral amphisome, but also by inhibiting the formation of the IAV-induced autophagosome, which may promote viral propagation.

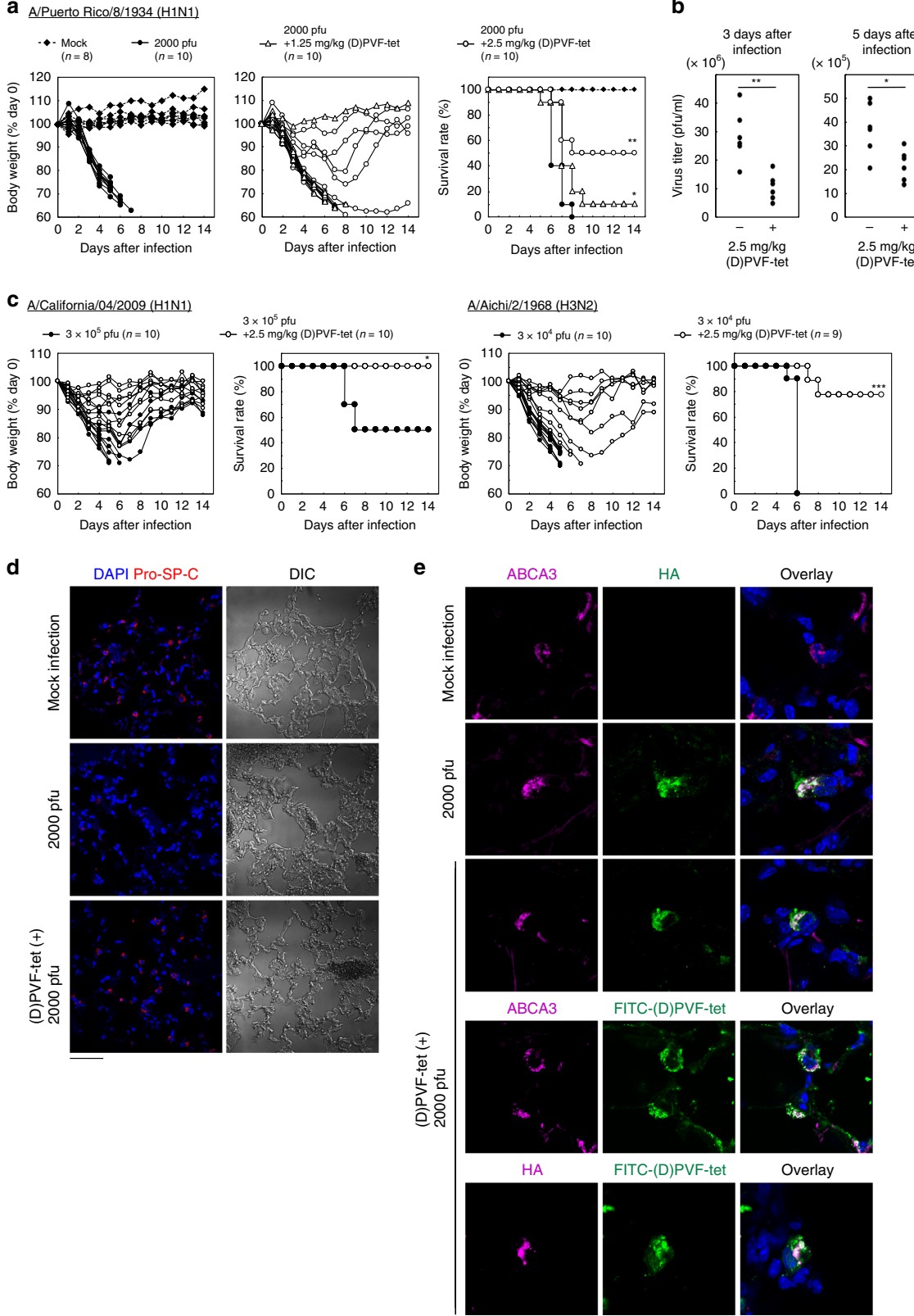

We found that IAV infection caused severe damage to AT-II cells in mice, although the AT-II cells were confirmed to express ABCA3 and SP-C, both of which are exclusive markers for the lamellar body. The result suggests that the amount of lamellar body in AT-II cells is not enough to produce a sufficient amount of anti-viral amphisome in vivo. Previous data have shown that

IAV infection causes damage to AT-II cells and also significant alterations in surfactant lipid metabolism, such that the levels of major surfactant phospholipids (phosphatidylcholine and phosphatidylglycerol) were reduced while those of minor phospholipids (phosphatidylserine and sphingomyelin) and cholesterol were increased[34]. It is likely that after infection, PVF-tet functions

**Fig. 7 (D)PVF-tet rescues mice from the lethality of IAV infection by specifically targeting AT-II cells. a, b** Female BALB/c mice were intranasally infected or not infected (Mock) with 2000 pfu IAV strain PR8 with or without 1.25 or 2.5 mg/kg (D)PVF-tet. The body weight of each mouse is presented as a percentage of the body weight on day 0 (**a**, left panel). The survival rate of each group is shown (**a**, right panel). Each group contained 8–10 mice. *$P < 0.05$; **$P < 0.01$ (by Log rank test). The virus titer in the mouse lungs harvested 3 or 5 days after infection was determined by plaque forming assay **b**. Each group contained six mice. *$P < 0.05$; **$P < 0.01$ (by Mann–Whitney $U$ test). **c** Female BALB/c mice were intranasally infected or not infected (Mock) with mouse-adapted IAV strain A/California/04/2009 (H1N1 pdm, $3 \times 10^5$ pfu) or mouse-adapted IAV strain A/Aichi/2/1968 (H3N2, $3 \times 10^4$ pfu) with or without intranasal co-injection with 2.5 mg/kg (D)PVF-tet. The body weight of each mouse is presented as a percentage of the body weight on day 0. The survival rate of each group is shown. Each group contained 9–10 mice. *$P < 0.05$; ***$P < 0.001$ (by log rank test). **d, e** Immunohistochemical analysis of the lungs harvested 3 days after infection. AT-II cells were stained using anti-Pro-SP-C antibody **d**. Localization of ABCA3, HA, or FITC-(D)PVF-tet in lung sections was examined by immunohistochemical analysis **e**. The nuclei were stained with DAPI. Scale bars represent 50 μm in **d** and 20 μm in **e**. Source data are provided as a Source Data file.

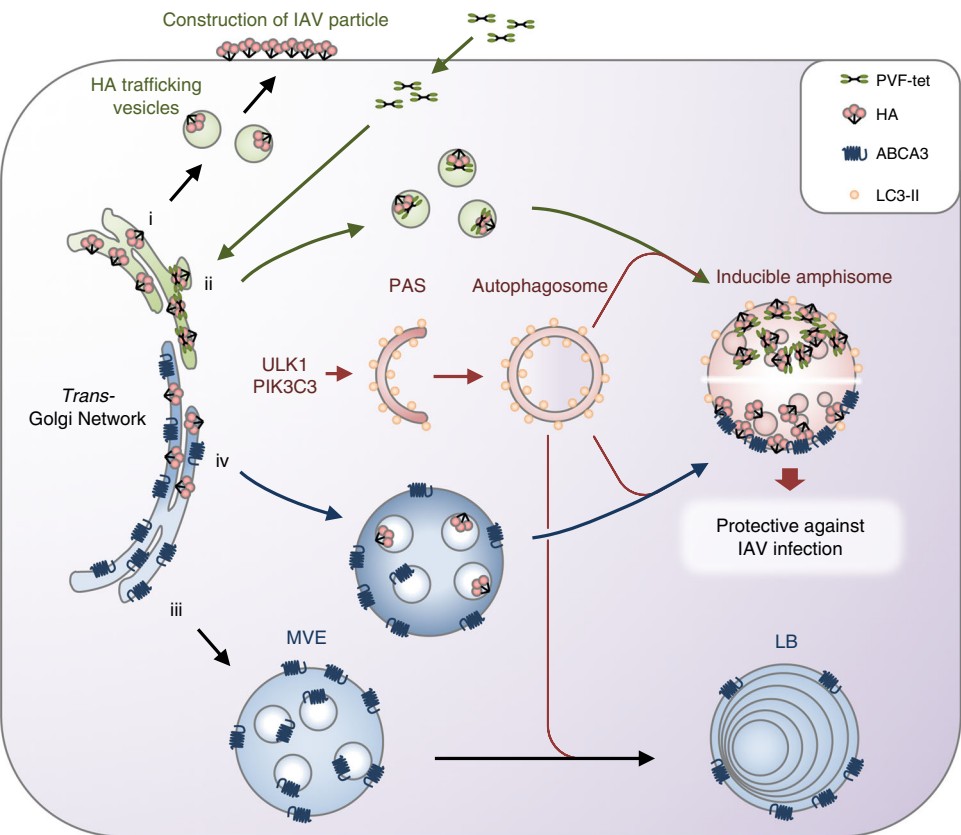

**Fig. 8 Mechanism of formation of the inducible amphisome with anti-IAV activity.** After IAV infection, newly synthesized HA is transported to the apical surface of the plasma membrane via vesicular trafficking (i). PVF-tet can penetrate into the cells and directly bind to the newly synthesized HA in the *trans*-Golgi Network to form an HA-containing structure. This structure subsequently matures into the amphisome, resulting in the sequestration of HA in this structure to inhibit virus propagation. Autophagic machinery is essential for this maturation step, because gene knockout of PIK3C3 and ULK1, both of which are indispensable for the formation of an autophagosome from a pre-autophagosomal structure, abrogates the sequestration (ii). In general, ABCA3 has an essential role in the maturation of the multivesicular endosome into the lamellar body through fusion with the autophagosome (iii). After IAV infection, even in the absence of PVF-tet, the overproduction of lamellar bodies promoted by overexpression of ABCA3 can induce the formation of amphisomes with anti-IAV activity depending on the autophagic machinery (iv).

to manage the lipid compartment containing ABCA3, in which cholesterol and sphingolipid are enriched, to be preferentially used to form the anti-viral amphisome that can efficiently isolate HA, although the precise mechanism remains to be clarified.

There are previous reports of organelle-based anti-microbial machineries such as the direct degradation of microbes through autophagy/xenophagy[61] and the mitochondrial anti-viral signaling system, which is activated by RIG-like receptor-mediated RNA sensing and coordinates signals on the mitochondria, peroxisomes, and ER[62,63]. Our results indicate that the inducible amphisome is another type of organelle-based defense machinery against IAV infection, which functions by specifically inactivating the viral HA protein. The precise regulation of the formation of

the amphisome could provide a new strategy to combat IAV infection.

## Methods

**Antibodies**. Antibodies were obtained from the vendors and used at the indicated dilution as follows: mouse monoclonal anti-Hemagglutinin (1:2000, clone C102, Genetex, Cat# GTX28262), rabbit polyclonal anti-Hemagglutinin (1:2000, Genetex, Cat# GTX127357), mouse monoclonal anti-Neuraminidase (1:2000, clone GT288, Genetex, Cat# 629696), rabbit polyclonal anti-Matrix protein 2 (1:2000, Genetex, Cat# 125951), mouse monoclonal anti-Nucleoprotein (1:2000, clone HT103, Kerafast, Cat# EMS010), mouse monoclonal anti-6xHistidine-tag (1:8000, clone 9C11, Wako, Cat# 015–23094), mouse monoclonal anti-GFP (1:1000, clone B-2, Santa Cruz Biotechnology, Cat# sc-9996), rabbit polyclonal anti-Fluorescein (1:500, Molecular Probes, Cat# A-889), rabbit polyclonal anti-β-actin (1:3000, MBL

International, Cat# PM-053), mouse monoclonal anti-ABCA3 (1:1000, clone 3C9, BioLegend, Cat# 911001), rabbit polyclonal anti-LAMP1 (1:500, Abcam, Cat# ab24170), rabbit polyclonal anti-EEA1 (1:1000, Thermo Fisher Scientific, Cat# PA1–063A), rabbit polyclonal anti-Calnexin (1:500, Santa Cruz Biotechnology, Cat# sc-11397), mouse monoclonal anti-p230 (1:500, BD Biosciences, Cat# 611280), mouse monoclonal anti-LC3 (1:250, Nano tools, Cat# 0231–100/ LC3–5F10), rabbit polyclonal anti-LC3 (1:1000, MBL International, Cat# PM036B), rabbit polyclonal anti-Pro-surfactant protein C (1:2000, Abcam, Cat# ab90716), rabbit polyclonal anti-p62 (1:1000, MBL International, Cat# PM045), mouse monoclonal anti-HSP47 (1:500, clone M16.10A1, Enzo Lifesciences, Cat# ADI-SPA-470-D), mouse monoclonal anti-GM130 (1:500, clone 35/GM130, BD Biosciences, Cat# 610822), mouse monoclonal anti-GS28 (1:500, clone 1/GS28, BD Biosciences, Cat# 611185), mouse monoclonal anti-ULK1 (1:1000, clone F-4, Santa Cruz Biotechnology, Cat# sc-390904), rabbit monoclonal anti-PI3 Kinase Class III (1:1000, clone D9A5, Cell Signaling Technology, Cat# 4263S), goat anti-rabbit IgG, horseradish peroxidase (HRP)-linked antibody (1:3000, Cell Signaling Technology, Cat# 7074S), horse anti-mouse IgG, HRP-linked antibody (1:2000, Cell Signaling Technology, Cat# 7076S), Alexa Fluor 488 goat anti-mouse IgG (1:2000, Thermo Fisher Scientific, Cat# A-11001), Alexa Fluor 546 goat anti-rabbit IgG (1:2000, Thermo Fisher Scientific, Cat# A-11010), Alexa Fluor 488 goat anti-rabbit IgG (1:2000, Thermo Fisher Scientific, Cat# A-11008), Alexa Fluor 546 goat anti-mouse IgG (1:2000, Thermo Fisher Scientific, Cat# A-11003). Prior to using, we tested the reactivity of the purchased antibodies against the samples alongside the positive control listed in the manufacturer's data sheet.

**Cell culture experiments**. Sf21 cells (Clontech) were maintained at 28 °C in TNM-FH (Sigma-Aldrich) supplemented with 10% FBS, 50 units/ml penicillin, and 50 μg/ml streptomycin. Madin-Darby canine kidney (MDCK) cells (CCL-34, ATCC) were maintained at 37 °C in minimum essential medium (MEM; Nacalai tesque, Japan) supplemented with 10% heat-inactivated FBS, 50 units/ml penicillin, and 50 μg/ml streptomycin. All cell lines used in this study were negative for mycoplasma contamination test.

**Virus preparation**. IAV strain A/Puerto Rico/8/1934 (PR8) was prepared as follows[5]. Confluent MDCK cell monolayers were infected with the virus at 0.001 MOI, which is defined as the number of fully infectious particles (i.e., pfu) per cell, in serum-free MEM containing 1 μg/ml trypsin, 0.2% bovine serum albumin (BSA), 25 mM 4-(2-hydroxyethyl)-1-piperazineethanesulfonic acid (HEPES) buffer, 50 units/ml penicillin, and 50 μg/ml streptomycin. After 72 h of incubation, the culture medium was recovered and stored at − 80 °C. Mouse-adapted IAV strain A/California/04/2009 (H1N1 pdm), A/Tokyo/UTHP013/2016 (H1N1 pdm), and mouse-adapted A/Aichi/2/1968 (H3N2) were kindly provided by Drs. Yoshihiro Kawaoka and Mutsumi Ito (Department of Microbiology and Immunology, Institute of Medical Science, University of Tokyo, Japan) and prepared as described above.

**Preparation of recombinant HA**. Recombinant histidine-tagged full-length HA (HA0) originated from IAV H1N1 strain A/Puerto Rico/8/1934 (PR8) was expressed in Sf21 cells using baculovirus as follows. A DNA fragment encoding full-length HA was obtained from cDNA of infected MDCK cells by PCR using specific primers (Supplementary Table. 1) and then cloned into a pBacPAK8 transfer vector (Clontech). A mutant construct with an amino-acid substitution of Leu194 to Ala (HA L194A) was prepared using a QuikChange site-directed mutagenesis kit (Agilent). Recombinant baculovirus was generated using the BacPAK Baculovirus Expression System (Clontech). Sf21 cells were infected with each recombinant baculovirus for 3 days and then lysed in lysis buffer (1% NP-40, 50 mM Tris-HCl pH 8.0, 135 mM NaCl, and complete protease inhibitor cocktail (Roche). The supernatants were incubated with Ni$^{2+}$-sepharose high-performance (GE Life Science) for 16 h at 4 °C. After the beads were washed, each immobilized HA was prepared. For the preparation of soluble HA, immobilized HA was eluted with elution buffer (0.1% NP-40, 20 mM Tris-HCl pH 8.0, 500 mM NaCl, and 1 M imidazole) followed by desalting.

**Peptides and library screening**. Tetravalent peptide libraries, peptide monomers, and tetravalent peptides were synthesized using N-α-Fmoc-protected amino acids and standard BOP/HOB coupling chemistry[15]. The synthesized peptides were validated by mass spectrometry analysis using the autoflexII TOF/TOF system (Bruker Daltonics) (Supplementary Fig. 18). The Met-Ala sequence at the amino terminus of the library peptides was included to verify that the peptides from the mixture were being sequenced and to qualify the peptides. The Ala situated next to the last degenerate position provided an estimate of peptide loss during sequencing.

Recombinant HA (HA0) or HA L194A (0.5 mg protein) bound to Ni$^{2+}$-sepharose beads was incubated with 300 μg of a given library peptide in phosphate-buffered saline (PBS) overnight at 4 °C. After extensive washing, the bound peptides were eluted with 30% acetic acid and sequenced on an Applied Biosystems model 477A protein sequencer. The molar ratio of each amino acid recovered from each degenerate position was calculated. The molar ratio of that obtained from the HA beads to that obtained from the HA L194A beads was calculated, and the sum of each ratio was normalized to 19 (the number of total

amino acids) to evaluate the relative amino acid preference at each degenerate position. Each amino acid would have a value of 1 in the absence of selectivity.

**Screening of tetravalent peptides synthesized on a membrane**. Spot synthesis of peptides on a cellulose membrane was performed using a ResPep SL SPOT synthesizer (INTAVIS Bioanalytical Instruments AG, Koeln, Germany)[18,19]. Fmoc-β-Ala-OH (Watanabe Chemical Industries, Japan) was used in the first cycle, followed by amino hexanoic acid as a spacer. Fmoc-Lys(Fmoc)-OH (Watanabe Chemical Industries) was used for the next two cycles to form four branches in the peptide chain for subsequent synthesis of the various motifs. After blocking with 5% skim milk in PBS, the membrane was blotted with $^{125}$I-labeled HA or $^{125}$I-labeled HA L194A (1 μg/ml) prepared using Iodintion Reagent (Thermo Fisher Scientific) according to the manufacturer's instruction. After extensive washing, the radioactivity of each bound peptide spot was quantified as a pixel value using a bio-imaging analyzer BAS-2500 (GE Healthcare Sciences, USA).

**Cytopathicity assay**. Confluent MDCK cell monolayers cultured on a 96-well plate were incubated with various concentrations of compounds at 37 °C for 30 min, and then infected with IAV strain PR8 at 10 MOI in MEM supplemented with 10% heat-inactivated FBS, 50 units/ml penicillin, and 50 μg/ml streptomycin (Trypsin(-)-MEM), or at 0.001 MOI in serum-free MEM supplemented with 1 μg/ml trypsin, 0.2% BSA, 25 mM HEPES, 50 units/ml penicillin, and 50 μg/ml streptomycin (Trypsin(+)-MEM). At the indicated time periods, the relative numbers of living cells were determined using cell count reagent SF (Nacalai tesque, Japan) according to the manufacturer's instruction. Fetuin, Bafilomycin A1, and zanamivir were purchased from Sigma-Aldrich, Cayman Chemicals, and Tokyo Chemical Industry, respectively.

**ELISA of the binding between HA and tetravalent peptides**. The indicated tetravalent peptides (2 μM) were applied onto each well of a 96-well enzyme-linked immunosorbent assay (ELISA) plate and incubated overnight for 24 h. After blocking, the plate was incubated with various concentrations of recombinant HA (HA0) for 1.5 h at room temperature. Bound HA was detected using mouse monoclonal anti-His-tag antibody (clone: 9C11, Wako Pure industries, Japan) and HRP-conjugated horse anti-mouse IgG antibody (Cell Signaling Technology).

**Binding assay between HA and its receptor mimic**. The AlphaScreen assay was used to assess the binding between recombinant HA and its receptor mimic, biotinylated sialyllactose polymer α2–3-sialyllactose-polyacrylamide (PAA) (Glycotech) as follows[17]. Various concentrations of tetravalent peptides were incubated with HA (10 μg/ml) in individual wells of an OptiPlate-384 (PerkinElmer) for 30 min at room temperature. After the addition of α2–3-sialyllactose-PAA (16 nM), the plate was further incubated for 1.5 h. The samples were then incubated with nickel chelate acceptor beads (20 μg/ml; PerkinElmer) for 30 min and then with streptavidin donor beads (20 μg/ml; PerkinElmer) for 1 h at room temperature in the dark. The plate was then subjected to excitation at 680 nm, and emission from the wells was monitored at 615 nm with an EnVision system (PerkinElmer). Data were obtained as the AU of signal intensity (count per second) used by the EnVision system.

**Kinetics analysis of the binding between peptides and HA**. The binding of inhibitory peptides to immobilized recombinant HA was quantitated using a BIAcore T100 system instrument (GE Healthcare Sciences, USA). Purified His-tagged HA (10 μg/ml) was injected into the system and fixed on the Ni$^{2+}$-chelate sensor chip. Various concentrations of PVF-tet were subsequently injected over the immobilized HA at a flow rate of 20 μl/min until reaching a plateau at 25 °C. The resonance unit is an arbitrary unit used by the BIAcore system. Binding kinetics were analyzed using BIAevaluation software, v1.1.1 (GE Healthcare Sciences).

**Measurement of virus propagation**. Confluent MDCK cell monolayers cultured on a 24-well plate were incubated with various concentrations of peptide or compound at 37 °C for 30 min The cells were infected with IAV at 0.2 MOI in Trypsin (-)-MEM for 16 h. The culture medium was used for the determination of the virus titer using a regular plaque forming assay using the MDCK cell monolayers[5].

**Western blot**. MDCK cells cultured on a 24-well plate were infected with IAV strain PR8 at 10 MOI for 1 h. After washing, the cells were cultured in the presence or absence of PVF-tet (20 μM) for the indicated time periods. The cells were lysed in a lysis buffer containing 1% NP-40, 0.2% sodium dodecyl sulfate (SDS), 10% glycerol, 25 mM Tris-HCl pH 8.0, 135 mM NaCl, and complete protease inhibitor cocktail (Roche). The lysates (20 μg protein) were separated by SDS-polyacrylamide gel electrophoresis and transferred to a polyvinylidene difluoride membrane. After blocking with 5% BSA, the membrane was immunoblotted with the indicated primary antibodies, followed by HRP-labeled goat anti-rabbit IgG or horse anti-mouse IgG (Cell Signaling Technology). The membrane was finally visualized by enhanced chemiluminescence (ECL) plus reagent (PerkinElmer), and analyzed using an LAS500 CCD imager (GE Healthcare Sciences). The amount of

β-actin or α-tubulin was used as an internal control. Source data including uncropped western blot images are provided as a Source Data file.

**Immunocytochemical analysis.** MDCK cells cultured in a 35 mm glass-bottom dish (IWAKI, Japan) were infected with IAV strain PR8 at 10 MOI for 1 h. After washing, cells were cultured in the presence or absence of PVF-tet (20 μM) for 15 h. The cells were fixed with 3.75% paraformaldehyde, permeabilized with 50 μg/ml digitonin (for detecting LC3 or p62) or 0.1% TritonX-100, and then incubated with specific antibodies against LC3 (PM036, MBL International), p62 (PM045, MBL International), HA (C102, Genetex), NA(GT288, Genetex), NP(HT103, Kerafast), M2 (Genetex), FITC (Molecular Probes), LAMP1 (Abcam), EEA1 (Thermo Fisher Scientific), GS28 (BD Biosciences), p230 (BD Biosciences), GM130 (BD Biosciences), HSP47 (Enzo Lifesciences), or Calnexin (Santa Cruz Biotechnology), followed by Alexa488-labeled or Alexa546-labeled secondary antibodies (Thermo Fisher Scientific). The nuclei were stained with 1 μg/ml DAPI. For the detection of the intracellular acidic compartment, the cells were treated with 75 nM Lysotracker DND-99 (Molecular probes) for 30 min followed by the PFA fixation. Then, the cells were permeabilized with 50 μg/ml digitonin (Wako Pure industries). For the detection of cellular cholesterol, the cells were treated with 50 μg/ml filipin III (Cayman chemical) for 30 min after PFA fixation. For the detection of ceramide, the cells were treated with BODIPY-TR-C5-Ceramide (Molecular Probes) according to the manufacturer's instruction and then infected with IAV strain PR8. All of the fluorescent images were analyzed using LSM710 laser scanning confocal microscopy (Zeiss).

**Establishment of MDCK cell clones.** MDCK cells stably expressing fluorescent-labeled HA (HAmVenus) or fluorescent-labeled ABCA3 (ABCA3mVenus) were prepared as follows. DNA fragments encoding HA from the cDNA of infected MDCK cells or mVenus from the mVenus N1 vector were amplified by PCR using specific primers (Supplementary Table. 1). mVenus N1 vector was a gift from Steven Vogel (Addgene plasmid #27793)[64]. DNA fragments encoding ABCA3 were obtained from the cDNA of MDCK cells by PCR using specific primers (Supplementary Table. 1). The DNA fragment of HA or ABCA3 was assembled with the DNA fragment of mVenus and then cloned into pcDNA3.1(-) vector (Thermo Fisher Scientific) using the Gibson assembly technique according to the manufacturer's protocol (New England Biolabs). MDCK cells were transfected with the empty vector (pcDNA3.1) or with the respective expression vectors encoding HAmVenus or ABCA3mVenus using ScreenFectA plus transfection reagent (Wako pure industries). After 48 h, the cells were treated with 500 μg/ml G418 (Nacalai tesque) for 1 week. The obtained G418-resistant and Venus fluorescence-positive cells were then sorted onto 96-well plates using an FACSAria II cell sorter (BD Biosciences). The resulting MDCK clones were maintained in the presence of 500 μg/ml G418 for at least 10 days.

MDCK-derived knockout clones of *PIK3C3* and *ULK1* were prepared using CRISPER-Cas9 genome editing[65]. Non-homologous end joining-mediated indel mutations were introduced into the exon of each gene. For the construction of sgRNA-Cas9 co-expression vector, DNA fragments (Supplementary Table. 1) were inserted into pSpCas9(BB)-2A-GFP (px458) vector according to original protocols. px458 vector (Addgene plasmid #48138) was a gift from Feng Zhang[65]. MDCK cells were transfected with sgRNA-Cas9 co-expression vector. After 72 h, the enhanced green fluorescent protein-positive cells were sorted into 96-well plates using an FACSAria II cell sorter. Each clone was screened based on protein expression analyzed by western blot using specific antibodies against ULK1 (Santa Cruz Biotechnology) or PIK3C3 (Cell Signaling Technology).

**Electron microscopy.** MDCK cells cultured in a 100-mm dish were harvested using 0.25% trypsin and fixed with 2% PFA and 1% glutaraldehyde in PBS(-) at room temperature for 1 h. After washing, the cells were embedded in 5% SeaPlaque GTG Agarose (Lonza) and then post-fixed with 2% OsO4 (TAAB Laboratories Equipment Ltd.) at room temperature for 1 h. After dehydration in a graded series of ethanol and propylene oxide, the cells were embedded in EPON812 (TAAB Laboratories Equipment Ltd) resin. The blocks were sectioned to 50-nm thickness with an ultramicrotome (EM UC6, Leica Microsystems) and stained with uranyl acetate and lead citrate. The ultra-thin sections were examined with a transmission electron microscope (JEM-2010, JEOL).

For correlative fluorescence and electron microscopy, MDCK cells expressing HAmVenus were grown on a carbon-coated gridded coverslip on a 24-well plate and infected with IAV strain PR8 at 10 MOI in the presence of PVF-tet (20 μM) for 16 h. After brief fixation with 3.75% PFA in PBS(−) at room temperature for 15 min, fluorescent images were obtained using confocal microscopy. After the imaging, the cells were fixed with 2% PFA and 1% glutaraldehyde in PBS(−) and post-fixed with 2% OsO4. The ultra-thin sections of cells were prepared and analyzed with a transmission electron microscope as described above.

**Quantitative polymerase chain reaction.** Mouse lung or MDCK cells were harvested into Sepasol RNAI Super G (Nacalai tesque, Japan). Lung homogenates were prepared using Micro Smash MS-100R (TOMY Seiko, Japan) with a zirconia bead (φ 5.0 mm). Then, total RNA was extracted, treated with DNaseI (Promega), and transcribed into cDNA using SuperScript III reverse transcriptase (Invitrogen)

according to the manufacturer's protocols. PCR was performed for 40 cycles using the obtained cDNA as a template and specific primers (Supplementary Table. 1). The obtained products were separated by 1% agarose gel electrophoresis for the detection of mouse or canine *ABCA3*. Alternatively, mRNA levels of *ABCA3* were quantified by RT-qPCR using SYBR green dye (Applied Biosystems) and the same primers. Data were analyzed by relative quantification based on the ddCt methods using *GAPDH* as a reference gene, and expressed as the fold increase over the average mRNA levels of three MDCK clones, which were transfected with an empty vector.

**Infection of mice with IAV.** Female 6- to 8-week-old, specific pathogen-free BALB/c mice (Shimizu Laboratory Supplies, Japan) were randomly assigned to each group prior to the experiments. Mice were anesthetized with isoflurane and intranasally infected with IAV strains PR8 (2000 pfu, equivalent to 10 LD$_{50}$ values), mouse-adapted A/California/04/2009 ($3 \times 10^5$ pfu), or mouse-adapted A/Aichi/2/1968 ($3 \times 10^4$ pfu) in the presence or absence of the indicated amounts of (D)PVF-tet. The survival rate and body weight of each mouse were monitored for 2 weeks. The survival rate was analyzed by Kaplan–Meier survival analysis. Under the same condition, the lung was harvested 3 or 5 days after infection and homogenized in 1 ml serum-free MEM using Micro Smash MS-100R (TOMY Seiko, Japan) with a zirconia bead (φ 5.0 mm). After centrifugation, the obtained supernatant was used for the determination of virus titer as described above. All animal experiments were approved by the animal ethics committee of Doshisha University prior to their commencement, and performed in accordance with approved protocols. No blinding method was applied in this study.

**Immunohistochemical analysis.** The lung prepared as described above was instilled with 1 ml 4% paraformaldehyde in PBS(+) and fixed with 4% paraformaldehyde for 2 days at room temperature. The lung was embedded in 5% Seaplaque agarose (Lonza) or paraffin and then sectioned to 100 μm or 6 μm thickness, respectively, with a microslicer (DOSAKA EM). Paraffin sections were deparaffinized with xylene and ethanol and then rehydrated. The obtained sections were permeabilized with 0.5% TritonX-100 in PBS(+) for 1 h, stained with specific antibodies against HA (C102), FITC, ABCA3 (3C9), or Pro-SP-C (Abcam), followed by Alexa488-labeled or Alexa546-labeled secondary antibodies, and then mounted with ProLong Diamond reagent (Thermo Fisher Scientific). All images were analyzed using LSM710 laser scanning confocal microscopy (Zeiss).

**Statistical analysis and general methods.** Prior to parametric statistical analysis, the normality of data was analyzed using the Shapiro–Wilk normality test. Significant differences between two groups were analyzed using unpaired two-sided Student's *t* test or Welch's *t* test based on the equality of two variances as tested by F-test. Multiple comparisons of differences among every group were analyzed using one-way analysis of variance (ANOVA) followed by Tukey's test. Significant differences between each group and the control group were analyzed using one-way ANOVA followed by Dunnett's test. These parametric statistical analyses were performed based on the assumption of normality of data distribution. The nonparametric Mann–Whitney *U* test was also used to analyze significant differences in data distribution between two groups and the *P* values were adjusted by Bonferroni correction. Significant differences of survival rate were analyzed using the log rank test. All statistical analysis was performed using IBM SPSS Statistics software (ver. 24.0.0.0).

No statistical methods were used to determine the sample size. We repeated each in vitro experiment at least three times and confirmed the reproducibility of each result. For the animal experiments, 5–10 mice were used, which should be sufficient to identify the effects. No data were excluded in this study.

**Reporting summary.** Further information on research design is available in the Nature Research Reporting Summary linked to this article.

## Data availability

All data that support the findings of this study are available upon request. The source data for Figs. 1a, 1b, 1d–f, 2b–d, 3a–c, 4b, 4d, 6a, 6c, 6d, and 7a–c and Supplementary Figs. 1a–d, 4b, 5a, 5b, 6b, 6e, 7b, 8d, 9h, 11b–d, 13a–c, 15a, 15b, 16a–d, 17a, and 17c are provided as a Source Data file.

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

## Acknowledgements

We thank Drs. Yoshihiro Kawaoka and Mutsumi Ito (Department of Microbiology and Immunology, Institute of Medical Science, University of Tokyo, Japan) for providing virus strain resources. We thank Dr. Shuku Kubo (Daiichi Sankyo Co., Ltd., Tokyo, Japan) for fruitful discussion. We also thank Mika Fukumoto (Doshisha University, Kyoto, Japan.) for technical assistance. This work was supported by grants from the Japan Society for the Promotion of Science (JSPS) KAKENHI Grant Numbers 18390044, 17H04002 and 18H04682; a grant from Harris Science Research Institute, Doshisha University; the Grant-in-Aid for Scientific Research on Innovative Areas "Brain Protein Aging and Dementia Control" Grant Number 26117004.

## Author contributions

J.O. and K.N. performed the experiments, analyzed and interpreted the data, and wrote the manuscript. M.W.-T., K.I., and C.Y.T. prepared the recombinant protein and performed peptide library-screening experiments. M.W.T. and E.S. synthesized the peptides and the peptide libraries. S.H. assisted with the in vivo experiments. Y.G. assisted with the binding analysis of peptides. R.N. assisted with the immunohistochemical analysis, with supervision from T.M., Yu.N., and A.M. performed electron microscopy imaging analysis. T.W. performed structure-based analysis, interpreted the data, and wrote the manuscript. K.N., Y.N., and M.Y. supervised the project.

## Competing interests

The authors declare no competing interests.
