## [Peer Review File · Nature Communications]

Reviewers' comments:

Reviewer #1 (Remarks to the Author):

The authors screened multivalent peptide libraries for tetravalent peptides which can bind influenza A haemagglutinin trimer with a clustering effect through multivalent interaction. Based on the initially found RVH-tet, they finally found a potent PVF-TET which can bind HA on solid phase assay, inhibit the interaction of HA with 2,3 sialic acid polymer and inhibit H1N1 PR8 virus induced cytopathic effects with EC50 at μM concentrations. However only PVF-TET but not PVF-monomer have such anti-HA and antiviral activity. They also modified the PVF-tet with D-arginine so that it becomes (D)PVF-tet and becomes resistant to degradation by trypsin in the cell culture medium which is essential for multicycle influenza virus infection. Moreover the PVF-tet or (D)PVF-tet are more active at a higher MOI while the control fetuin is less active at high MOI. Time of addition experiment using viral nucleoprotein expression suggested that PVF-tet does not affect virus entry but is active when added up to 9 hour after virus challenge in cell culture. Furthermore the amount of HA but not other viral proteins accumulated in infected cells and peaked at 9 hr after virus challenge before it declined. The authors observed that PVF-tet bind newly synthesized HA as punctate spots in form of vacuole like structures in infected cells. Cytochemical analysis suggested that the intracellular HA was associated with cholesterol and sphingolipid as shown by lipid probes in these LAMP positive and lysotracker positive (highly acidified) vacuoles. These vacuoles were likely to be originated from lamellar body. ABCA3 knockout cells had a reduced size of these HA containing vacuoles as ABCA3 is involved in the formation of lamellar body and thus the maturation of these HA containing structure. Electron microscopy showed that these vacuoles had both lamellar membrane and electron dense core structures which are compatible with amphisomes (fusion of endosomes with autophagosomes). Experiments with specific inhibitors 3MA and wortmannin also suggest that these are amphisomes. Finally intranasal administration of (D)PVF partially rescued the virus challenged mice, decreased weight loss, reduced lung viral load and loss of AT-II cells.

Strength

1. The PVF-tet has some antiviral activity in both cell culture and mice challenge experiments.
2. The authors have track record in peptide library screening and publication on shigatoxin.
3. There are lots of experiments performed and the paper is well written.

Weakness

The authors claimed that (D)PVF-tet inhibited viral infection by binding to newly synthesized HA due to the clustering effect, but not the HA of free infectious extracellular mature virus. And the HA-PVF-

TET complexes induced the amphisome formation. If this observation is true, this is a novel finding. However, the data were not solid enough to support these findings.

Firstly, why does a PVF-TET, which can bind to the viral receptor site, not bind to extracellular free parent virus with mature trimeric HA on the surface but only bind to newly synthesized HA? They have not shown the so called clustering effect.

Secondly, how does this PVF-TET penetrate into cells to bind the newly synthesized HA inside cells? What is the mechanism of uptake of PVF-TET into cells?

Thirdly, what makes the difference in terms of the mechanism of the PVF-tet inhibiting viral infection but not the monomer-PVF? The authors did not very clearly show that PVF-Tet could not inhibit virus entry. Fig. 3a and Fig. 3b were not the good methods to identify entry. Not inhibiting entry is most important key step in their study. If this is true, they will have novel finding. If this is not true, they may have wrong conclusion)Time of addition experiments.

Finally, the co-localization assay did not convincingly indicate that PVF-tet binds to newly synthesized HA to form the amphisome.

Detailed comments:

1, It is not clear why all identified motifs have P amino acid in Fig. 1c?

2, The effective concentration for antiviral of PVF-TET in Fig. 1f is more than 5uM as the cell viability is . This may not be a good antiviral. No control shown

3, In line 136, why does the antiviral activity need the PVF-TET clustering? No data shown to demonstrate the clustering effect and to explain the mechanism behind this observation which is the foundation of all subsequent findings.

4, The cell viability assay in Fig. 2 showed very different results in Fig. 2b,c,d. Why were the viabilities of untreated virus control so different from 20% to >50%? How could the viability of cells infected by 0.001 MOI cause 80% cell death at 24h post infection but the viability of cells infected by 10 MOI cause only 50% (Fig. 2c) cell death at 24h post infection? This does not seem logical at all.

5, In Fig. 2c and 2d, it is hard to say that the antiviral activity of PVF-TET had a different mechanism than that of fetuin. The lack of antiviral activity may be because the 7.5uM of fetuin used in Fig. 2d was too low to inhibit viral infection according the Fig. 2c (right Fig). It is not clear how the author treated the cells and virus with the PVF-TET and fetuin: before, during or after infection? All these points must be clarified.

6, In line 154, is there any data to show that PVF-tet cannot bind to free infectious virions, but only bind to newly synthesized intracellular HA? This is one of the most controversial issue in this study.

7, In line 156, Fig. 3a did not indicate that PVF-tet cannot inhibit virus entry, because the author only measured NP expression at 16 hr and measured viability at 3 time points with multi-cycle infection assay. The pNP percentage expressed at 16 hr does not indicate the lack of PVF-TET activity against viruz entry. However, the cell protection in Fig. 2b indicated the amount of cell survival but not indicated the percentage of infected cells.

8, In line 160, Fig. 3b may not indicate that PVF-TET functions in a later stage. It might indicate that earlier treatment had stronger antiviral activity. Same problem happens here: measuring the result at 24h post-infection within multi-life cycle did not indicate that PVF-tet functions at a later stage. Making such conclusion with these data is wrong.

9, Why did only HA protein in the untreated virus sample decrease whereas other viral proteins increase in 16 h post-infection in Fig. 3c (Left pane of Fig. 3c)? It is difficult to understand why PVF-TET (right panel of Fig. 3c) could stick HA inside cells but not inhibit other viral protein production (All other viral proteins were increased in cells in the right panel of Fig. 3c. How can we see the antiviral activity of PVF-TET).

10, In Line 170, Fig. 3e, in uninfected cells, PVF-TET mainly localized at the membrane. This tends to suggest that PVF-TET may not penetrate into cells efficiently? What is the mechanism of PVF-TET uptake? If PVF-TET can really go into cells to bind to the newly synthesized HA to interfere viral replication, the most important and direct experiment should be done by infecting cells with 2 MOI for 1h, then treating infected cells at 2 h post-infection with the PVF-TET, and removing the PVF-TET media at 6h post infection. After these treatment, evaluate if the PVF-TET go into cells to inhibit viral infection and accumulate the newly synthesized HA in the vacuole at 8 h or 10 h post-infection.

12, In line 267, it seems that 3-MA treatment did not inhibit the acidification of HA-containing structure. The 3-MA inhibited the formation of big autophagosome or amphisome in Fig. 6c.

13, In line 269, why did 3-MA suppress the antiviral activity of PVF-TET, but 3-MA treated samples had low HA expression in the samples with or without PVF treatment in Fig. 6e? Was it due to the cytotoxicity of 3MA?

14, In line 280, it is not clear how many LD50s were used and what method was used to inoculate the PVF-tet to mice. Does the co-injection mean that the PVF-tet was premixed with the virus? If the antiviral mechanism of PVF-tet is to bind to newly synthesized HA, there is no need to premix virus with PVF-tet to do the animal experiment? This is another highly controversial point in this study.

Minor issue:

1. For the animal experiment, allowing a weight loss of 40% without euthanasia is inhumane and against animal ethics.

2. This is quite wrong to say that the amphisome is an antiviral machinery. The sequestration of HA by PVF-TET into the amphisome is the key mechanism leading to the interruption of the viral life cycle. Claiming a host antiviral machinery is not a scientific description of the finding and this must be deleted.

The authors did not demonstrate how they can differentiate between mature HA and newly synthesized HA. Both of which can be present inside infected cell. Thus their conclusion about PVF-TET binding to only new synthesized HA is unjustified.

Reviewer #2 (Remarks to the Author):

Based on a peptide library screening, the authors selected one peptide (PVF-tet) for mechanistic investigations. The peptide was shown to cause accumulation of influenza virus HA in a vesicular compartment which was identified, by extensive confocal microscopy studies, as an amphisome. PVF-tet displayed activity against influenza virus in cell culture and mouse models.

The data package is impressive with a wide variety of highly specialized and complementary methodologies. I feel that the data are new, interesting and thoroughly elaborated. Overall, the conclusions are supported by the data. The PVF-tet part is perhaps quite specific, but this is compensated by the broader relevance of the ABCA3 experiments.

Specific comments:

1. Why was a tetravalent strategy used, considering that HA is a homotrimer?
2. By which mechanism can this peptide enter into cells (line 317)?
3. The authors SHOULD comment on the peptides' structure-activity relationship, i.e. why do some peptide sequences work better than others (cfr Figure 1).
4. Any idea on which part of HA PVF-tet is binding? The relevance of PVF-tet as an antiviral concept would be hugely increased if the authors can demonstrate that the antiviral activity is not restricted to A/PR/8 but also valid for other HA subtypes.
5. The peptide should be tested for cytotoxicity in the MDCK cells. Then, calculate the antiviral EC50 or EC90 value and the ratio between cytotoxic and antiviral concentration (= selectivity index).
6. Some experiments lack a control compound. For instance: the 1-5-9h experiment (Fig. 3b) would be more convincing if bafilomycin was included. The mouse study should have included an antiviral drug like an NAI. Another relevant control is an inactive peptide analogue like PVD-tet [or rather its (D)form].
7. Mouse studies: what was the rationale to do coinjection of virus with peptide? Would the peptide not work when administered a few hours after virus infection?
8. Line 875. Why was NP detection done at 16 h p.i.? If the purpose was to investigate virus entry, NP staining should have been done at 1 h p.i. At 16 h p.i., a second replication round has started; the lack of inhibition by PVF-tet indicates that its antiviral effect is not convincing.
9. Line 180. Why is this consistent? What could be the reason why HA is not degraded within these vesicles?

Some minor points:

10. Line 27: IAV cytopathicity (better than: cytotoxicity). Change also at other sites in the text.
11. Line 39-40: spelling error in neuraminidase. ... is responsible for the release of ... (more correct than: budding).
12. Line 49. The multivalent interaction is not limited to one HA trimer, but is likely further strengthened by several HA trimers showing simultaneous binding to cell surface sialoglycans.
13. Introduction: the background literature is rather old. Check and update as much as possible.
14. Line 139: reformulate. Cleavage is required to render the HA fusion-competent.
15. Line 161. Was virus infection synchronized?
16. Line 124. Why was the test limited to an α 2,3 sialoglycan; an α 2,6 analogue is equally relevant.

Reviewer #3 (Remarks to the Author):

Manuscript Nr: NCOMMS-18-17953

Omi et al., "The Inducible Amphisome Isolates Viral Hemagglutinin and Defends Against Influenza A Virus Infection"

The authors demonstrate that tetravalent peptides that have been identified in screens for influenza virus hemagglutinin (HA) binders inhibit viral replication. This inhibition is associated with HA sequestration into a vesicular compartment that contains lysosomal (LAMP1) and autophagosomal (LC3) markers. HA is associated with lipid rafts and overexpression of the lamellar body inducing ABCA3 transporter assembles similar compartments and inhibits influenza virus replication. From these data the authors conclude that an amphisome results from these treatments that could be explored for therapeutic approaches against influenza virus.

These are potentially interesting findings on a new therapeutic strategy against influenza A virus. However, it remains a speculation that the observed vesicles are amphisomes and unclear how these compartments restrict influenza A virus infection.

Major comments:

1. The multivalent HA binders might aggregate newly synthesized HA in the ER or Golgi, which cannot be removed by retrograde translocation and proteasomal degradation in the cytosol. ERphagy or autophagy of Golgi might take care of these aggregates. Can the authors find ER or Golgi proteins apart from calnexin or p230 in their so-called amphisomes? Does endocytosed material gain access to these vesicles? Depending on this distinction the authors can call the vesicles amphisomes or just autophagosomes that have engulfed ER or Golgi plus aggregated HA.
2. The antiviral effect of the tetravalent peptide remains enigmatic. Why is HA not degraded in the LAMP1 positive acidified vesicles? How is HA inactivated? Can less HA be found on the cell surface of tetravalent peptide treated influenza virus infected cells after HA sequestration?

3. The authors employ PI3 kinase inhibitors (3-MA and wortmannin) to block autophagy. These are, however, by no means specific for this process. In order to argue that the autophagic machinery is involved in producing the HA containing vesicles specific Atg silencing should be performed?

4. LC3 conjugation to membranes does not automatically indicate autophagy. This attachment has been also described for a subset of phagosomes in a process called LC3 associated phagocytosis (LAP). In the present manuscript the authors do not distinguish between LAP and autophagy. Only autophagy depends on the ULK1 complex and components of this complex (ULK1 or Fip200) could be silenced. Autophagosomes contain also often sequestosome 1/p62 as a prototypic autophagy cargo. Therefore, the presence of this protein in the HA positive vesicles could be assessed.

5. Finally, the in vivo data do not connect the restriction of infection to characteristic accumulations of the in vitro observed vesicles. Does LC3-II also accumulate in pneumocytes type II after tetravalent peptide treatment during influenza infection?

Minor comments:

1. It is unclear why the original descriptions of autophagic membrane accumulation during influenza A virus infection are not cited in this study: Beale et al., *Cell Host Microbe* 2014; Gannage et al., *Cell Host Microbe* 2009.

In summary, the presented data show accumulation of HA in LC3 and LAMP1 positive vesicles after tetravalent peptide treatment. This is associated with restriction of influenza virus infection, but more data on the identity of these vesicles and how they inhibit infection should be provided.

Response to the comments of the reviewer #1:

Reviewer #1 (Remarks to the Author):

The authors screened multivalent peptide libraries for tetravalent peptides which can bind influenza A haemagglutinin trimer with a clustering effect through multivalent interaction. Based on the initially found RVH-tet, they finally found a potent PVF-TET which can bind HA on solid phase assay, inhibit the interaction of HA with 2,3 sialic acid polymer and inhibit H1N1 PR8 virus induced cytopathic effects with EC50 at uM concentrations. However only PVF-TET but not PVF-monomer have such anti-HA and antiviral activity. They also modified the PVF-tet with D-arginine so that it becomes (D)PVF-tet and becomes resistant to degradation by trypsin in the cell culture medium which is essential for multicycle influenza virus infection. Moreover the PVF-tet or (D)PVF-tet are more active at a higher MOI while the control fetuin is less active at high MOI. Time of addition experiment using viral nucleoprotein expression suggested that PVF-tet does not affect virus entry but is active when added up to 9 hour after virus challenge in cell culture. Furthermore the amount of HA but not other viral proteins accumulated in infected cells and peaked at 9 hr after virus challenge before it declined. The authors observed that PVF-tet bind newly synthesized HA as punctate spots in form of vacuole like structures in infected cells. Cytochemical analysis suggested that the intracellular HA was associated with cholesterol and sphingolipid as shown by lipid probes in these LAMP positive and lysotracker positive (highly acidified) vacuoles. These vacuoles were likely to be originated from lamellar body. ABCA3 knockout cells had a reduced size of these HA containing vacuoles as ABCA3 is involved in the formation of lamellar body and thus the maturation of these HA containing structure. Electron microscopy showed that these vacuoles had both lamellar membrane and electron dense core structures which are compatible with amphisomes (fusion of endosomes with autophagosomes).

Experiments with specific inhibitors 3MA and wortmannin also suggest that these are amphisomes. Finally intranasal administration of (D)PVF partially rescued the virus challenged mice, decreased weight loss, reduced lung viral load and loss of AT-II cells.

Strength

1. The PVF-tet has some antiviral activity in both cell culture and mice challenge experiments.
2. The authors have track record in peptide library screening and publication on shigatoxin.
3. There are lots of experiments performed and the paper is well written.

Weakness

The authors claimed that (D)PVF-tet inhibited viral infection by binding to newly synthesized HA due to the clustering effect, but not the HA of free infectious extracellular mature virus. And the HA-PVF-TET complexes induced the amphisome formation. If this observation is true, this is a novel finding. However, the data were not solid enough to support these findings.

Firstly, why does a PVF-TET, which can bind to the viral receptor site, not bind to extracellular free parent virus with mature trimeric HA on the surface but only bind to newly synthesized HA? They have not shown the so called clustering effect.

Secondly, how does this PVF-TET penetrate into cells to bind the newly synthesized HA inside cells? What is the mechanism of uptake of PVF-TET into cells?

Thirdly, what makes the difference in terms of the mechanism of the PVF-tet inhibiting viral infection but not the monomer-PVF? The authors did not very clearly show that PVF-Tet could not inhibit virus entry. Fig. 3a and Fig. 3b were not the good methods to identify entry. Not inhibiting entry is most important key step in their study. If this is true, they will have novel finding. If this is not true, they may have wrong conclusion)Time of addition experiments.

Finally, the co-localization assay did not convincingly indicate that PVF-tet binds to newly synthesized HA to form the amphisome.

Response: We would like to thank the reviewer #1 for reviewing our manuscript and offering the constructive suggestions that help strengthen our conclusion. Throughout the comments, the reviewer #1 raised 3 major concerns.

1) The shortness of data showing that (D)PVF-tet inhibited viral infection by binding to newly synthesized HA due to the clustering effect, but not the HA of free infectious extracellular mature virus. We additionally showed that PVF-tet binds to newly synthesized HA as described in the response to the detailed comments no. 6, 7, 8, 9 and 10, and in the response to the minor comments no. 2-2nd paragraph, and that PVF-tet does not bind to the HA of free infectious extracellular mature virus as described in the response to the detailed comments no. 6. Also, as described in the response to the detailed comments no. 3, we showed that PVF-tet functions through the “clustering effect” including intramolecular and intermolecular interaction with HA. We additionally showed that the HA-PVF-tet complexes induced the amphisome formation as described in the response to the detailed comments no. 12, 13, and in the response to the minor comments no. 2-1st paragraph.

2) Unclearness about the mechanism of uptake of PVF-tet into cells. We clearly showed that PVF-tet can be incorporated into the cells via endocytotic pathway as described in the response to the detailed comments no. 10.

3) Unclearness about the difference in terms of the mechanism of the PVF-tet inhibiting viral infection but not the monomer-PVF. We clarified this point as described in the response to the detailed comments no. 3.

We have addressed all of the reviewer’s concerns in the point-by-point responses as follows.

Detailed comments:

1) It is not clear why all identified motifs have P amino acid in Fig. 1c?

Response: As shown in line 122 in the original manuscript, to further improve the binding activity of RVH-tet to HA, we performed the screening of tetravalent peptides synthesized on a cellulose membrane based on the HA-binding motif (RRRVNHH) of RVH-tet. For the first screen, each amino acid present in the sequence RRRVNHH was replaced one by one by all amino acids except Cys as shown in Supplementary Fig.

1a. The top 3 sequences obtained were RRPVNHH, RRRDNHH, and RRSVNHH. Based on these 3 motifs, the second library was designed as shown in Supplementary Fig. 1b, and the top 5 sequences were identified as final HA-binding motifs as shown in Fig. 1c, all of which were obtained based on RRPVNHH motif. That is the reason why all of the identified motifs in Fig. 1c have Pro at position 3.

2) The effective concentration for antiviral of PVF-TET in Fig. 1f is more than 5uM as the cell viability is . This may not be a good antiviral. No control shown

Response: Following the reviewer's suggestion, we compared the anti-viral activity of the identified tetravalent peptides with those of zanamivir and (MA-AU)₄-3Lys (MA-tet), which has the same core structure but lacks any HA-binding motifs, as a positive and a negative control, respectively. As shown in Fig. 1f in the revised manuscript, PVF-tet efficiently inhibited the cytotoxicity induced by IAV infection with comparable efficacy to zanamivir under this condition. In contrast, MA-tet did not inhibit the cytotoxicity. We replaced Fig. 1f in the original manuscript, and accordingly changed the Results and Legends (lines 123-126 and 980-988, respectively) in the revised manuscript.

3) In line 136, why does the antiviral activity need the PVF-TET clustering? No data shown to demonstrate the clustering effect and to explain the mechanism behind this observation which is the foundation of all subsequent findings.

Response: As shown in lines 48-50 in the original manuscript, the "clustering effect" can be defined as a phenomenon in which the multivalent interaction between two molecules markedly increases the binding affinity. Thus, Fig. 2a clearly indicates that PVF-tet, a tetravalent peptide, can exert the "clustering effect" to bind to HA-trimer, because PVF-mono, the monomer form of PVF-tet, did not bind to HA at all. In this case, in addition to the intramolecular interaction in the complex of HA and PVF-tet, the intermolecular cross-linking of HA by PVF-tet might be involved in this "clustering effect" and then in its efficient anti-IAV activity. To elucidate this point and also answer the reviewer's comment, we examined whether PVF-tet has a potency to induce

the oligomer-formation of HA through the intermolecular cross-linking. As shown in Supplementary Fig. 2, PVF-tet, but not PVF-mono, could induce the formation of highly clustered HA after incubation with HA, supporting that PVF-tet functions through the intramolecular and intermolecular interaction. To clarify this point, we added Supplementary Fig. 2, and accordingly changed the Results (lines 132-135) in the revised manuscript.

- 4) The cell viability assay in Fig. 2 showed very different results in Fig. 2b,c,d. Why were the viabilities of untreated virus control so different from 20% to >50%? How could the viability of cells infected by 0.001 MOI cause 80% cell death at 24h post infection but the viability of cells infected by 10 MOI cause only 50% (Fig. 2c) cell death at 24h post infection? This does not seem logical at all.

Response: In the left panel of Fig. 2c in the original manuscript, cells were infected for 48 h, not for 24 h. We sincerely apologize for the mistake, and changed the Legends (line 1002) in the revised manuscript.

- 5) In Fig. 2c and 2d, it is hard to say that the antiviral activity of PVF-TET had a different mechanism than that of fetuin. The lack of antiviral activity may be because the 7.5uM of fetuin used in Fig. 2d was too low to inhibit viral infection according the Fig. 2c (right Fig). It is not clear how the author treated the cells and virus with the PVF-TET and fetuin: before, during or after infection? All these points must be clarified.

Response: Following the reviewer's comment, we clarified the experimental conditions in the legends for Fig. 2c and 2d in the revised manuscript. In both cases, MDCK cells were incubated with various concentrations of fetuin or tetravalent peptide for 30 min before the IAV infection. A possible reason why fetuin could not inhibit the cytotoxicity at high MOI condition is that NA present on the virus particles could release sialic acid from fetuin. Following the reviewer's comment, we performed additional experiments showing that PVF-tet has a different mechanism than that of fetuin as shown below (response no. 6).

- 6) In line 154, is there any data to show that PVF-tet cannot bind to free infectious virions, but only bind to newly synthesized intracellular HA? This is one of the most controversial issue in this study.

Response: Following the reviewer's comment, we examined whether PVF-tet directly binds to infectious IAV particles. IAV particles were prepared from the culture medium of MDCK cells which were infected with IAV strain PR8 at 0.001 MOI in the presence of 1 µg/ml trypsin for 72 h. As clearly shown in Supplementary Fig. 4a, PVF-tet did not efficiently bind to IAV particles, while fetuin did bind with remarkable potency (left panel). On the other hand, both PVF-tet and fetuin bound to recombinant HA with similar efficacy (right panel). Furthermore, we also found that the proteolytic cleavage of HA into HA1 and HA2 fragments by trypsin, which mimics the state of HA present on the surface of infectious IAV particles, markedly reduced the binding of PVF-tet to HA (Supplementary Fig. 4b). This result could explain, at least in part, why PVF-tet does not efficiently bind to infectious IAV particles. All of these observations further support the contention that PVF-tet only binds to newly synthesized HA present in the Golgi or the ER, which is not cleaved into HA1 and HA2 fragments yet. We added Supplementary Figs. 4a and 4b, and accordingly changed the Results (lines 166-172) in the revised manuscript.

- 7) In line 156, Fig. 3a did not indicate that PVF-tet cannot inhibit virus entry, because the author only measured NP expression at 16 hr and measured viability at 3 time points with multi-cycle infection assay. The pNP percentage expressed at 16 hr does not indicate the lack of PVF-TET activity against virus entry. However, the cell protection in Fig. 2b indicated the amount of cell survival but not indicated the percentage of infected cells.
- 8) In line 160, Fig. 3b may not indicate that PVF-TET functions in a later stage. It might indicate that earlier treatment had stronger antiviral activity. Same problem happens here: measuring the result at 24h post-infection within multi-life cycle did not indicate that PVF-tet functions at a later stage. Making such conclusion with these data is wrong.

Response: In Fig. 3-A in the original manuscript, intracellular NP was detected 3 h, not

16 h, after IAV infection. We sincerely apologize for the mistake, and changed the Legends (lines 1014-1016 in the revised manuscript). As shown in Fig. 3a in the revised manuscript, newly synthesized NP was exclusively detected in the nucleus 3 h after single-cycle infection (not multi-cycle infection), which corresponds to an early stage of infection as judged by the expression time course of each viral protein (Fig. 3c). Then, after 16 h of infection, NP was mainly detected in the cytosol as shown in Fig. 3d. PVF-tet did not affect the localization of NP throughout this time period including the early stage. Furthermore, we performed an additional experiment using bafilomycin A1, which functions at a very early stage of the infection to suppress the fusion process after endocytosis of the virus. As shown in Fig.3b in the revised manuscript, bafilomycin A1 treatment could inhibit IAV cytotoxicity when administered up to 1 h after initial IAV infection (single-cycle infection), but could not when administered after 5 h. In contrast, PVF-tet treatment successfully inhibited the cytotoxicity even when administered up to 9 h after infection. All of these observations indicate that PVF-tet functions in a later stage of infection. We replaced Figs. 3a and 3b in the original manuscript, and accordingly changed the Results, Methods, and Legends (lines 158-165, 533-535, and 1020-1026, respectively) in the revised manuscript.

- 9) Why did only HA protein in the untreated virus sample decrease whereas other viral proteins increase in 16 h post-infection in Fig. 3c (Left pane of Fig. 3c)? It is difficult to understand why PVF-TET (right panel of Fig. 3c) could stick HA inside cells but not inhibit other viral protein production (All other viral proteins were increased in cells in the right panel of Fig. 3c. How can we see the antiviral activity of PVF-TET).

Response: As the reviewer points out, in the absence of PVF-tet, the amount of newly synthesized HA peaked at 9 h after single-cycle infection and subsequently declined, while other viral proteins accumulated. After 9 h of infection, these viral proteins started to be assembled into progeny virus particles and then released into the culture medium. Accordingly, because of the relatively low amount of newly synthesized HA compared to other viral proteins, the total amount of HA present in cells was markedly reduced to be used for the production of progeny virus particles, indicating that the amount of HA is a key factor for the virus production. Consistent with this observation,

the accumulation of HA in a vacuole-like structure induced by PVF-tet, which results in the shortage of HA available for the progeny virus production, can function as an effective anti-viral machinery.

PVF-tet, which was designed to specifically bind to the receptor binding region of HA trimer, did not bind to other viral proteins as shown in Figs. 3d and 3e, where FITC-PVF-tet co-localized with HA in the vacuole-like structure, but not with other viral proteins. Furthermore, intracellular localization of these proteins was not affected by PVF-tet treatment (Fig. 3d). Thus, even in the presence of PVF-tet, the accumulation of these proteins does not represent the sequestration of these proteins.

10) In Line 170, Fig. 3e, in uninfected cells, PVF-TET mainly localized at the membrane. This tends to suggest that PVF-TET may not penetrate into cells efficiently? What is the mechanism of PVF-TET uptake? If PVF-TET can really go into cells to bind to the newly synthesized HA to interfere viral replication, the most important and direct experiment should be done by infecting cells with 2 MOI for 1h, then treating infected cells at 2 h post-infection with the PVF-TET, and removing the PVF-TET media at 6h post infection. After these treatment, evaluate if the PVF-TET go into cells to inhibit viral infection and accumulate the newly synthesized HA in the vacuole at 8 h or 10 h post-infection.

Response: Following the reviewer's comment, we examined the mechanism of the uptake of PVF-tet. In Supplementary Fig. 4g in the revised manuscript, MDCK cells were incubated with FITC-labeled PVF-tet in the presence of Alexa594-labeled dextran, which is widely known to be incorporated into cells via micropinocytosis, for 1 h at 4 °C or 37 °C. Although both PVF-tet and dextran were exclusively detected on the plasma membrane at 4 °C, PVF-tet localized in the endocytic vesicles containing dextran at 37 °C, indicating that PVF-tet can be incorporated into the cells via endocytotic pathway.

Following the reviewer's suggestion, we performed the additional experiments. In Supplementary Fig. 4c in the revised manuscript, MDCK cells were infected with IAV at 10 MOI for 1 h, and then treated with FITC-PVF-tet at 2 h post-infection. After 4 h incubation, the cells were washed, and cultured for 4 h. Microscopic analysis showed that PVF-tet could still induce the formation of vacuole-like structure in which PVF-tet

co-localized with newly synthesized HA. In addition, in Supplementary Fig. 4d in the revised manuscript, we examined the effect of the same procedures on the anti-viral activity of PVF-tet. MDCK cells were infected with IAV at 10 MOI for 1 h, and then treated with PVF-tet at 2 h post-infection. After the indicated time periods, the cells were washed, and further cultured up to 24 h post infection. Under this condition, the treatment of PVF-tet still efficiently inhibited the cytotoxicity in a treatment-time dependent manner. All of these results clearly indicate that PVF-tet can penetrate into the cells to bind to newly synthesized HA. We added Supplementary Figs. 4c, 4d and 4g, and accordingly changed the Results (lines 183-185 and 191-195) in the revised manuscript.

12) In line 267, it seems that 3-MA treatment did not inhibit the acidification of HA-containing structure. The 3-MA inhibited the formation of big autophagosome or amphisome in Fig. 6c.

Response: As shown in Fig.6c in the original manuscript, the 3-MA treatment strongly inhibited the acidification of the PVF-tet-induced HA-containing structure (left panel), and also the formation of the HA-containing vacuoles, i.e. big autophagosomes or amphisomes (right panel), as the reviewer points out. In order to further clarify the involvement of PI3K in the formation of the HA-containing vacuoles, we examined the effect of SAR405, a more specific PI3K inhibitor, and the effects of gene-knockout of *PIK3C3* and *ULK1*, a direct activator of PI3K, as shown in Figs. 6c and 6d, and Supplementary Figs. 9 and 10 in the revised manuscript.

13) In line 269, why did 3-MA suppress the antiviral activity of PVF-TET, but 3-MA treated samples had low HA expression in the samples with or without PVF treatment in Fig. 6e? Was it due to the cytotoxicity of 3MA?

Response: As the reviewer points out in Fig. 6e in the original manuscript, the relative amount of intracellular HA was decreased by 3-MA treatment in the presence or absence of PVF-tet, although it is not statistically significant. That is not due to the cytotoxicity of 3-MA, because the treatment of uninfected MDCK cells with 5 mM

3-MA for 16 h did not affect the cell viability. The decrease of the relative amount of HA may reflect the suppressive effect of 3-MA on the viral protein synthesis, which has already been shown in the previous report (Int J Biochem Cell Biol. 72, 100-108, 2016). However, even though the treatment of 3-MA has some suppressive effects on the amount of HA, the treatment of PVF-tet holds the capacity to induce the accumulation of substantial amount of HA, which localizes in the small vesicles, but not in the vacuole-like structure, in the presence of 3-MA, as shown in Fig.6c (right panel). Thus, the PVF-tet-induced accumulation of HA occurs prior to the formation of the vacuole-like structure containing HA, further confirming that the formation of the structure is essential for the anti-viral activity of PVF-tet.

14) In line 280, it is not clear how many LD50s were used and what method was used to inoculate the PVF-tet to mice. Does the co-injection mean that the PVF-tet was premixed with the virus? If the antiviral mechanism of PVF-tet is to bind to newly synthesized HA, there is no need to premix virus with PVF-tet to do the animal experiment? This is another highly controversial point in this study.

Response: In Fig.7 in the original manuscript, mice were intranasally infected with 2000 pfu IAV strain PR8, which is equivalent to 10 LD₅₀ values. Following the reviewer's comment, we added the information in Methods and Legends (lines 687-690, 1115-1116, respectively) in the revised manuscript. In Fig.7 in the original manuscript, (D)PVF-tet was premixed with the virus solution and then injected to mice. Following reviewer's suggestion, we examined whether (D)PVF-tet can protect mice from the lethality even when administered after virus infection. As shown in Supplementary Fig. 11b in the revised manuscript, intranasal administration of (D)PVF-tet after 6 h post infection still partially attenuated the weight loss and rescued 50% of mice from mortality. We added Supplementary Fig. 11b, and accordingly changed the Results (lines 324-326) in the revised manuscript.

Minor issue:

1) For the animal experiment, allowing a weight loss of 40% without euthanasia is inhumane and against animal ethics.

Response:All the animal experiments were approved by the animal ethics committee of Doshisha University prior to their commencement, and performed in accordance with approved protocol. In Fig.7a in the original manuscript, the weight loss for all the mice was within 40%. Following the reviewer's suggestion, in Supplementary Figs. 11a and 11b in the revised manuscript, we euthanized the mice with weight loss of over 30%.

- 2) This is quite wrong to say that the amphisome is an antiviral machinery. The sequestration of HA by PVF-TET into the amphisome is the key mechanism leading to the interruption of the viral life cycle. Claiming a host antiviral machinery is not a scientific description of the finding and this must be deleted.

Response: As the reviewer comments, it is not suitable to simply describe the amphisome as an anti-viral machinery, because in general the amphisome functions as an intermediate organelle observed in various type of cells, which, in many cases, subsequently fuses with the lysosome to degrade the vesicular contents (ref. 24-26 in the original manuscript). Also in alveolar type-II epithelial cells, the amphisome functions to mature into the lamellar body, a lipid-sorting organelle that supplies alveolar surfactant (ref. 28-30 in the original manuscript). In this study, PVF-tet induced the accumulation of HA within a unique structure, “the inducible amphisome,” whose features are consistent with, but not equivalent to, the conventional amphisome: the inducible amphisome is highly acidified compared to the conventional amphisome, and HA accumulated in it is not degraded. In order to clarify this point, we would like to describe this type of amphisome that functions as an anti-viral machinery as “the inducible amphisome” in the original and the revised manuscript.

The authors did not demonstrate how they can differentiate between mature HA and newly synthesized HA. Both of which can be present inside infected cell. Thus their conclusion about PVF-TET binding to only new synthesized HA is unjustified.

Response: Following the reviewer's comment, we examined the effect of PVF-tet treatment on mature HA, which is present on the plasma membrane and ready to assemble with other viral proteins to form progeny virus particles. As shown in Supplementary Fig. 4e in the revised manuscript, in the absence of PVF-tet, substantial amount of mature HA was found to be present on the outer surface of the plasma membrane 16 h after infection, and intracellular HA (newly synthesized HA) was detected in small vesicles to be transported to the plasma membrane. In contrast, in the presence of PVF-tet, mature HA was not detected in the plasma membrane, but the marked accumulation of intracellular HA in a vacuole-like structure was observed, indicating that PVF-tet sequestered newly synthesized HA in the structure to hamper its vesicular transport to the plasma membrane. We added Supplementary Fig. 4e, and accordingly changed the Results (lines 185-187) in the revised manuscript.

Response to the comments of the reviewer #2:

Reviewer #2 (Remarks to the Author):

Based on a peptide library screening, the authors selected one peptide (PVF-tet) for mechanistic investigations. The peptide was shown to cause accumulation of influenza virus HA in a vesicular compartment which was identified, by extensive confocal microscopy studies, as an amphosome. PVF-tet displayed activity against influenza virus in cell culture and mouse models.

The data package is impressive with a wide variety of highly specialized and complementary methodologies. I feel that the data are new, interesting and thoroughly elaborated. Overall, the conclusions are supported by the data. The PVF-tet part is perhaps quite specific, but this is compensated by the broader relevance of the ABCA3 experiments.

Response: We would like to thank the reviewer #2 for reviewing our manuscript and offering the constructive suggestions that help strengthen our conclusion. We have addressed all of the reviewer's concerns in the point-by-point responses as follows, especially with the clear data about the peptides' structure-activity relationship as described in the response to the detailed comments no. 3.

Specific comments:

1) Why was a tetravalent strategy used, considering that HA is a homotrimer?

Response: It is possible to synthesize divalent, tetravalent, or even octavalent peptide library: Fmoc-Lys(Fmoc)-OH is used in the first single cycle, two cycles, or three cycles to create two, four, or eight branches, respectively, in the peptide chain for subsequent library synthesis by standard BOP/HOB coupling chemistry. Following this method, it is impossible to synthesize trivalent one. As shown in Fig. 2a in the original manuscript, PVF-tet, a tetravalent peptide, but not its monomeric form, efficiently binds to HA trimer, confirming the strategic rationale to use a tetravalent peptide library for screening.

2) By which mechanism can this peptide enter into cells (line 317)?

Response: To address the reviewer's question, we examined the mechanism of the uptake of PVF-tet. In Supplementary Fig. 4g in the revised manuscript, MDCK cells were incubated with FITC-labeled PVF-tet in the presence of Alexa594-labeled dextran, which is widely known to be incorporated into cells via micropinocytosis, for 1 h at 4 °C or 37 °C. Although both PVF-tet and dextran were exclusively detected on the plasma membrane at 4 °C, PVF-tet localized in the endocytic vesicles containing dextran at 37 °C, indicating that PVF-tet can be incorporated into the cells via endocytotic pathway. Previous studies have shown that monovalent or polyvalent peptide with poly-Args holds potency to penetrate into the cells via endocytic pathway (Biochemistry, 41, 7925-7930, 2002; Biochemistry, 46, 492-501, 2007; Molecular Therapy, 17, 1868-1876, 2009). PVF-tet has two tandem Args in its HA-binding motif, which make 8 Args in the tetravalent structure, probably contributing to its cell-permeable property. We added Supplementary Fig. 4g, and accordingly changed the Results (lines 191-195) in the revised manuscript.

3) The authors SHOULD comment on the peptides' structure-activity relationship, i.e. why do some peptide sequences work better than others (cfr Figure 1).

Response: As the reviewer points out, among the identified tetravalent peptides only PVD-tet showed very low anti-viral activity, although all of them have similar binding affinities for HA and equally inhibited the binding between HA and a receptor mimic polymer, as shown in Fig. 1e in the original manuscript. To address the reviewer's question, we compared the cell-permeability of the tetravalent peptides as a possible factor to evaluate the structure-activity relationship. As shown in Supplementary Fig. 4f in the revised manuscript, only FITC-PVD-tet failed to penetrate into the cells compared to the others. As described above, the presence of poly-Args in the peptide molecules is attributed to the cell-permeability. Among the identified tetravalent peptides, only PVD-tet has an Asp in its HA-binding motif, which may diminish the electrostatic role of the tandem Args in the cell-permeability, resulting in the loss of the

anti-viral activity. We added Supplementary Fig. 4f, and accordingly changed the Results (lines 190-198) in the revised manuscript.

- 4) Any idea on which part of HA PVF-tet is binding? The relevance of PVF-tet as an antiviral concept would be hugely increased if the authors can demonstrate that the antiviral activity is not restricted to A/PR/8 but also valid for other HA subtypes.

Response: Based on the observation that PVF-tet inhibited the binding between HA and α 2-3 sialyllactose-polymer, it is reasonable to conclude that PVF-tet binds to the receptor-binding region of HA. Furthermore, as shown in Supplementary Fig. 1c, PVF-tet also efficiently inhibited the binding of HA to α 2-6 sialyllactose-polymer, suggesting that PVF-tet can widely function to inhibit the interaction between HA and sialic acid on the receptor binding region. We added Supplementary Fig. 1c, and accordingly changed the Results (lines 119-123) in the revised manuscript.

We have already found that PVF-tet bound to recombinant HA derived from highly virulent avian H5N1 A/chicken/Yamaguchi/7/2004 strain, which exclusively binds to α 2-3 sialyllactose-polymer, and also to its human-tropic mutant with two mutations (Q226L and G228S), which exclusively binds to α 2-6 sialyllactose-polymer. Thus, we believe that based on the HA-binding motif of PVF-tet, highly customized binding motifs for HA of H5N1 virus could be developed by using affinity maturation-based screening of tetravalent peptides synthesized on a membrane. In this paper, however, we would like to focus on H1N1 virus.

- 5) The peptide should be tested for cytotoxicity in the MDCK cells. Then, calculate the antiviral EC50 or EC90 value and the ratio between cytotoxic and antiviral concentration (= selectivity index).

Response: Following reviewer's suggestion, we examined the cytotoxicity of the identified tetravalent peptides in MDCK cells. As shown in Supplementary Fig. 1d in the revised manuscript, none of these tetravalent peptides showed cytotoxicity up to 60 μ M, which is a practically available maximum-dose of each peptide. Following the reviewer's comment, we added Supplementary Fig. 1d, and changed the Legends for Fig.

1f with each EC50 value (lines 985-988) in the revised manuscript.

- 6) Some experiments lack a control compound. For instance: the 1-5-9h experiment (Fig. 3b) would be more convincing if bafilomycin was included. The mouse study should have included an antiviral drug like an NAI. Another relevant control is an inactive peptide analogue like PVD-tet [or rather its (D)form].

Response: Following reviewer's comments, we performed additional experiments. As shown in Fig. 3b in the revised manuscript, we additionally examined the effect of bafilomycin A1. MDCK cells were infected with IAV at 4 °C for 1h to synchronize the IAV binding to the cells. After extensive washing, the cells were treated with each compound at the indicated time periods after washing. We found that the treatment with bafilomycin A1 5 h after infection completely reduced the anti-viral activity. We replaced Fig. 3b in the original manuscript, and accordingly changed the Results, Methods, and Legends (lines 162-165, 533-535, and 1020-1026, respectively) in the revised manuscript.

As shown in Supplementary Fig. 11a in the revised manuscript, we examined the effect of zanamivir, an established NA inhibitor, on the lethality of IAV infection in mice. We added Supplementary Fig. 11a, and accordingly changed the Results (lines 322-324) in the revised manuscript.

- 7) Mouse studies: what was the rationale to do coinjection of virus with peptide? Would the peptide not work when administered a few hours after virus infection?

Response: Following the reviewer's suggestion, we examined whether (D)PVF-tet can protect mice from the lethality even when administered after virus infection. As shown in Supplementary Fig. 11b, intranasal administration of (D)PVF-tet after 6 h post infection still partially attenuated the weight loss and rescued 50% of mice from mortality. We added Supplementary Fig. 11b, and accordingly changed the Results (lines 324-326) in the revised manuscript.

- 8) Line 875. Why was NP detection done at 16 h p.i.? If the purpose was to investigate

virus entry, NP staining should have been done at 1 h p.i. At 16 h p.i., a second replication round has started; the lack of inhibition by PVF-tet indicates that its antiviral effect is not convincing.

Response: In Fig. 3a in the original manuscript, intracellular NP was detected 3 h, not 16 h, after IAV infection. We sincerely apologize for the mistake, and changed the Legends (lines 1014-1016 in the revised manuscript). As shown in Fig. 3a in the revised manuscript, newly synthesized NP was exclusively detected in the nucleus 3 h after infection, which corresponds to an early stage of infection as judged by the expression time-course of each viral protein (Fig. 3c).

- 9) Line 180. Why is this consistent? What could be the reason why HA is not degraded within these vesicles?

Response: As the reviewer points out, the sentence in line 180 in the original manuscript is obscure. To clarify this point, we changed the sentence as shown in the revised manuscript (lines 206-210).

To address the reviewer's question, we examined whether the HA-containing vacuole-like structure contains lysosomal enzymes. We found that cathepsin B, one of the representative lysosomal cysteine proteases, was detected in the structure by immunocytochemistry, indicating that the structure has a feature of autolysosomes, which are formed after fusion of autophagosomes with the lysosome. To elucidate why HA is not degraded in the vacuole-like structure that contains lysosomal proteases, the effect of PVF-tet on the oligomer-formation of HA was examined. As shown in Supplementary Fig. 2, PVF-tet, but not PVF-mono, could induce the formation of highly clustered HA after incubation with HA, supporting that PVF-tet functions to induce the oligomer-formation of HA through the intramolecular and intermolecular interaction. The PVF-tet induced oligomer-formation of HA in the vacuole-like structure might be a possible reason why HA in the structure is resistant to proteolytic degradation. We added Supplementary Fig. 2, and accordingly changed the Results (lines 132-135, and 208-210, in the revised manuscript).

Some minor points:

10) Line 27: IAV cytopathicity (better than: cytotoxicity). Change also at other sites in the text.

Response: Following the reviewer's suggestion, we replaced the word "cytotoxicity" with "cytopathicity" in the revised manuscript.

11) Line 39-40: spelling error in neuraminidase. ... is responsible for the release of ... (more correct than: budding).

Response: We sincerely apologize for the spelling error. Following the reviewer's suggestion, we changed the sentence as follows in the revised manuscript (lines 35-38): the most common currently used therapeutic agents are inhibitors of neuraminidase (NA), a viral coat protein that is responsible for the release of newly synthesized virus from the plasma membrane of infected cells.

12) Line 49. The multivalent interaction is not limited to one HA trimer, but is likely further strengthened by several HA trimers showing simultaneous binding to cell surface sialoglycans.

Response: Following the reviewer's comment, we changed the sentence as follows in the revised manuscript (lines 45-49): Thus, the HA trimer can bind three sialic acid molecules through a multivalent interaction, sometimes referred to as the "clustering effect," which increases its binding affinity for its receptor by thousands fold. Consequently, several HA trimers on the virus particle simultaneously bind to its receptor to further strengthen the clustering effect.

13) Introduction: the background literature is rather old. Check and update as much as possible.

Response: Following the reviewer's suggestion, we updated the background literature in Introduction in the revised manuscript.

14) Line 139: reformulate. Cleavage is required to render the HA fusion-competent.

Response: Following the reviewer's suggestion, we changed the sentence as follows in the revised manuscript (lines 138-141): under a low multiplicity of infection (MOI), HA on virus particles released from infected cells must be cleaved into two fragments, HA1 and HA2, by trypsin present in the culture medium, which enables the virus to be fused with the endosome-membrane after endocytosis by the target cells.

15) Line 161. Was virus infection synchronized?

Response: In Fig.3b in the original manuscript, we did not synchronize the virus infection. Following reviewer's comments, we re-performed the experiments with bafilomycin A1 as shown in the response to the comment no. 6. The experiments were performed after synchronizing the IAV binding to the cells.

16) Line 124. Why was the test limited to an α 2,3 sialoglycan; an α 2,6 analogue is equally relevant.

Response: Following the reviewer's comment, we performed additional experiments with α 2-6 sialyllactose as shown in the response to the comment no. 4.

Response to the comments of the reviewer #3:

Reviewer #3 (Remarks to the Author):

The authors demonstrate that tetravalent peptides that have been identified in screens for influenza virus hemagglutinin (HA) binders inhibit viral replication. This inhibition is associated with HA sequestration into a vesicular compartment that contains lysosomal (LAMP1) and autophagosomal (LC3) markers. HA is associated with lipid rafts and overexpression of the lamellar body inducing ABCA3 transporter assembles similar compartments and inhibits influenza virus replication. From these data the authors conclude that an amphisome results from these treatments that could be explored for therapeutic approaches against influenza virus.

These are potentially interesting findings on a new therapeutic strategy against influenza A virus. However, it remains a speculation that the observed vesicles are amphisomes and unclear how these compartments restrict influenza A virus infection.

In summary, the presented data show accumulation of HA in LC3 and LAMP1 positive vesicles after tetravalent peptide treatment. This is associated with restriction of influenza virus infection, but more data on the identity of these vesicles and how they inhibit infection should be provided.

Response: We would like to thank the reviewer #3 for reviewing our manuscript and offering the constructive suggestions that help strengthen our conclusion. Throughout the comment, the reviewer #3 raised 2 major concerns.

- 1) The shortness of data showing the observed vesicles are amphisomes. We clarified this point as described in the response to the detailed comments no. 1, 3, 4, and 5.
- 2) Unclearness about the mechanism of how these compartments restrict influenza A virus infection. We clarified this point as described in the response to the detailed comments no. 2.

We have addressed all of the reviewer's concerns in the point-by-point responses as follows.

Major comments:

- 1) The multivalent HA binders might aggregate newly synthesized HA in the ER or Golgi, which cannot be removed by retrograde translocation and proteasomal degradation in the cytosol. ERphagy or autophagy of Golgi might take care of these aggregates. Can the authors find ER or Golgi proteins apart from calnexin or p230 in their so-called amphisomes? Does endocytosed material gain access to these vesicles? Depending on this distinction the authors can call the vesicles amphisomes or just autophagosomes that have engulfed ER or Golgi plus aggregated HA.

Response: Following the reviewer's suggestion, we examined co-localization of HA with additional organelle markers, such as (Supplementary Fig. 5c) HSP47 (an ER marker), (Supplementary Fig. 5f) GS28 (a *trans*-Golgi network marker), and (Supplementary Fig. 5g) GM130 (a *cis*-Golgi network marker). As shown in Supplementary Fig. 5 in the revised manuscript, none of these markers did not localize with the HA-containing vacuole-like structure induced by PVF-tet treatment, indicating that autophagy of the ER or the Golgi is not involved in the formation of the structure. We added Supplementary Fig. 5, and accordingly changed the Results (lines 200-204) in the revised manuscript.

Following the reviewer's suggestion, we examined whether dextran, which is widely known to be incorporated into cells via micropinocytosis and transported to the endosomes, can be incorporated into the HA-containing vacuole-like structure induced by PVF-tet. As shown in Supplementary Fig. 8c in the revised manuscript, in the presence of PVF-tet Alexa594-labeled dextran was efficiently incorporated into the HA-containing vacuole-like structure, while in the absence of PVF-tet dextran did not localize with HA. This observation further confirms that the HA-containing vacuole-like structure induced by PVF-tet is an amphisome, which is generally formed by the fusion of various endosomes with autophagosomes. We added Supplementary Fig. 8c, and accordingly changed the Results (lines 290-292) in the revised manuscript.

- 2) The antiviral effect of the tetravalent peptide remains enigmatic. Why is HA not degraded in the LAMP1 positive acidified vesicles? How is HA inactivated? Can less

HA be found on the cell surface of tetravalent peptide treated influenza virus infected cells after HA sequestration?

Response: To address the reviewer's question, we examined whether the HA-containing vacuole-like structure contains lysosomal enzymes. We found that cathepsin B, one of the representative lysosomal cysteine proteases, was detected in the structure by immunocytochemistry, indicating that the structure has a feature of autolysosomes, which are formed after fusion of autophagosomes with the lysosome. To elucidate why HA is not degraded in the vacuole-like structure that contains lysosomal proteases, the effect of PVF-tet on the oligomer-formation of HA was examined. As shown in Supplementary Fig. 2, PVF-tet, but not PVF-mono, induced the formation of highly clustered HA after incubation with HA, supporting that PVF-tet functions to induce the oligomer-formation of HA through the intramolecular and intermolecular interaction. The PVF-tet induced oligomer-formation of HA in the vacuole-like structure might be a possible reason why HA in the structure is resistant to proteolytic degradation. We added Supplementary Fig. 2, and accordingly changed the Discussion (lines 132-135, and 208-210) in the revised manuscript.

To address the reviewer's question, we examined the effect of PVF-tet treatment on the localization of HA in cells. As shown in Supplementary Fig. 4e in the revised manuscript, in the absence of PVF-tet, substantial amount of HA was found to be present on the outer surface of the plasma membrane 16 h after infection, and intracellular HA was detected in small vesicles to be transported to the plasma membrane. In contrast, in the presence of PVF-tet, HA was not detected on the plasma membrane, but the marked accumulation of intracellular HA in a vacuole-like structure was observed, indicating that PVF-tet sequestered newly synthesized HA in the structure to hamper its vesicular transport to the plasma membrane for the virus production. We added Supplementary Fig. 4e, and accordingly changed the Results (lines 185-187) in the revised manuscript.

- 3) The authors employ PI3 kinase inhibitors (3-MA and wortmannin) to block autophagy. These are, however, by no means specific for this process. In order to argue that the autophagic machinery is involved in producing the HA containing vesicles specific Atg

silencing should be performed?

Response: Following the reviewer's suggestion, we examined the effects of gene-knockout of *PIK3C3* and *ULK1*, a direct activator of PI3K, on the anti-viral activity of PVF-tet. As shown in Figs. 6c and 6d, and Supplementary Figs. 9a and 9b in the revised manuscript, not only the formation of HA-containing vacuole-like structure induced by PVF-tet, but also the dose-dependent anti-viral activity of PVF-tet were significantly suppressed in the *PIK3C3*- and *ULK1*-knockout MDCK cells, clearly indicating the involvement of the autophagic machinery in the anti-viral activity of PVF-tet. We added Figs. 6c and 6d, and Supplementary Figs. 9a and 9b, and accordingly changed the Results, Methods and Legends (lines 296-310, 633-648, and 1097-1112, respectively) in the revised manuscript.

In order to further clarify this point, we also examined the effect of SAR405, a recently developed specific inhibitor for class III PI3K. As shown in Supplementary Fig. 9c in the revised manuscript, the dose-dependent anti-viral activity of PVF-tet were completely suppressed by SAR405 treatment, further confirming the involvement of the autophagic machinery in the anti-viral activity of PVF-tet. We added Supplementary Fig. 9c, and accordingly changed the Results (lines 310-312) in the revised manuscript.

- 4) LC3 conjugation to membranes does not automatically indicate autophagy. This attachment has been also described for a subset of phagosomes in a process called LC3 associated phagocytosis (LAP). In the present manuscript the authors do not distinguish between LAP and autophagy. Only autophagy depends on the ULK1 complex and components of this complex (ULK1 or Fip200) could be silenced. Autophagosomes contain also often sequestosome 1/p62 as a prototypic autophagy cargo. Therefore, the presence of this protein in the HA positive vesicles could be assessed.

Response: Following the reviewer's suggestion, we examined the effects of gene-knockout of *ULK1* as shown in the response to the comment no. 3.

Following reviewer's suggestion, we examined whether p62/SQSTM1 is detected in the HA-containing vacuole-like structure induced by PVF-tet. As shown in Supplementary Fig. 8b in the revised manuscript, p62/SQSTM1 was partially

accumulated in the HA-containing vacuole-like structure induced by PVF-tet treatment, further confirming the involvement of the autophagic machinery in the anti-viral activity of PVF-tet. We added Supplementary Fig. 8b, and accordingly changed the Results (lines 288-290) in the revised manuscript.

- 5) Finally, the *in vivo* data do not connect the restriction of infection to characteristic accumulations of the *in vitro* observed vesicles. Does LC3-II also accumulate in pneumocytes type II after tetravalent peptide treatment during influenza infection?

Response: Following the reviewer's comment, we examined the content of LC3-I and LC3-II in the cell-population enriched with alveolar type-II epithelial cells isolated from IAV infected, or IAV and (D)PVF-tet co-injected mice. As shown in Supplementary Figs. 11d and 11e in the revised manuscript, LC3-II production was enhanced by IAV infection, but the ratio of LC3-II to LC3-I, which correlates the progression of autophagy, was substantially increased in IAV and (D)PVF-tet co-injected mice compared to IAV infected mice, consistent with the *in vitro* observation as shown in Fig. 6a. Combined with the results obtained from Fig. 7, these observations indicate that (D)PVF-tet facilitates the formation of amphisomes with accumulated HA through the autophagic machinery to exert the anti-IAV activity *in vivo*. We added Supplementary Figs. 11d, and 11e, and accordingly changed the Results (lines 339-346) in the revised manuscript.

Minor comments:

- 1) It is unclear why the original descriptions of autophagic membrane accumulation during influenza A virus infection are not cited in this study: Beale et al., *Cell Host Microbe* 2014; Gannage et al., *Cell Host Microbe* 2009.

Response: Following the reviewer's suggestion, we cited the two papers in the Results and Discussion (lines 276-279 and 395-398, respectively) in the revised manuscript.

Reviewers' comments:

Reviewer #1 (Remarks to the Author):

These are our main assessment:

A. The authors have answered the majority of our questions. But they did not answer some critical ones but just deleted their previous figs or admitted that they have made some mistakes.

1) For example, the Fig. 2b and Fig.3 (right figure) used 10 MOI and 24h, but the cell control viability was 20% in Fig. 2b and 50% in Fig 3c right. What is the problem? The PVF-Tet cannot efficiently protect cells in low MOI (Fig.2c left), but more efficiently protect cells in high MOI. What is the reason? After they changed their legends in Fig 2, with or without trypsin, it may have answered the question. However, if the trypsin can cause significant difference in findings, why they did not describe it in the original manuscript!

2) In Fig. 3-A in the original manuscript, intracellular NP was detected 3 h, not 16 h (which is now cited as the time for first detection of NP) after IAV infection. Such serious mistakes in the initial manuscript are quite unacceptable and throws doubt on the reliability of the findings in this work!

3) The author did not answer our question 7 and 8. They just said they made very important mistake for question 7, and did new experiments. The absence of effect of PVF-TET on the NP expression can be due to the low antiviral efficiency of PVF-TET when using MOI 2. Obviously, the PVF-TET's antiviral efficiency is low, which may not significantly affect the NP ratio in infected cells. If they want to make a clear case to show that the PVF-TET cannot affect the binding and entry, at least another method is needed to show that the PVF-tet does not affect the viral RNA replication within one life cycle. This is so important for the conclusion of the whole study that the authors must provide very clear evidence. However, the author only used cell survival at 24h post infection, which can therefore can only claim that PVF-TET have antiviral activity, but does not indicate that PVF-TET cannot affect entry even with the time of addition assay.

4) A major contradiction is that PVF-TET can block 100% of virus release in Fig 4E, but the antiviral activity shown in the preceding in vitro and then the mouse experiments is poor. The authors need to explain this contradiction very clearly

B. But the most important problems are:

1) They screened out the PVF-tet by a binding assay to HA receptor and showed that the PVF-TET can really bind to the HA receptor binding domain. Then they claimed that the PVF-tet did not bind to viral HA, but only bind to newly synthesized intracellular HA. How can this happen at all? They must support their claim by showing structural differences between the HA receptor binding domains between the newly synthesized intracellular HA and that of the HA on released virus? This is a key problem in the study!!!. They paid a lot of attention on the downstream experiments to claim that the PVF-tet only bind to newly synthesized intracellular HA, but they did not clearly verify if the PVF-tet cannot bind to virus and affect viral entry. This must be answered before the paper can be considered for publication. Otherwise, the conclusion can be wrong.

2) The antiviral efficacy of PVF-TET was really not good, only protected 50% of H1N1-infected mice when PVF-TET was administrated at 6 h post infection. There are many published antiviral compounds and antiviral peptides which are better than this protection and put doubt on the significance of this findings.

3) 3) Moreover some of the experimental design are not reasonable. Note that if the authors cannot claim any new antiviral mechanism and show no data of broad protection in mice infected by other influenza virus, it would be doubtful if the work can be considered for publication in a reputable journal. I am sorry that I cannot be helpful in finding a way out for the authors on this occasion.

Warm regards.

Reviewer #2 (Remarks to the Author):

The authors have adequately and thoroughly addressed all my comments. This includes a few new experiments which create added value. I have only two minor comments:

- Dextran is a marker of macropinocytosis (not: micropinocytosis)
- To clearly show the experimental difference between Fig. 7a and Suppl. Fig. 11b, specify in the Legend the time when (D)PVF-tet was administered (together with virus or at 6 h p.i.).

Reviewer #3 (Remarks to the Author):

Manuscript Nr: NCOMMS-18-17953A

Omi et al., "The Inducible Amphisome Isolates Viral Hemagglutinin and Defends Against Influenza A Virus Infection"

The authors demonstrate that tetravalent peptides that have been identified in screens for influenza virus hemagglutinin (HA) binders inhibit viral replication. This inhibition is associated with HA sequestration into a vesicular compartment that contains lysosomal (LAMP1) and autophagosomal (LC3) markers. HA is associated with lipid rafts and overexpression of the lamellar body inducing ABCA3 transporter assembles similar compartments and inhibits influenza virus replication. From these data the authors conclude that an amphisome results from these treatments that could be explored for therapeutic approaches against influenza virus.

In their revised manuscript version, the authors have addressed most of my comments. Mainly, they have characterized the amphisomes in more detail, they document reduced HA surface expression after HA aggregation, they perform more specific ULK1 and PI3 kinase silencing and demonstrate also LC3 accumulation in vivo. These additional experimental data have significantly strengthened the presented manuscript.

Response to the comments of the reviewer #1:

Reviewer #1 (Remarks to the Author):

These are our main assessment:

A. The authors have answered the majority of our questions. But they did not answer some critical ones but just deleted their previous figs or admitted that they have made some mistakes.

1) For example, the Fig. 2b and Fig.3 (right figure) used 10 MOI and 24h, but the cell control viability was 20% in Fig. 2b and 50% in Fig 3c right. What is the problem? The PVF-Tet cannot efficiently protect cells in low MOI (Fig.2c left), but more efficiently protect cells in high MOI. What is the reason? After they changed their legends in Fig 2, with or without trypsin, it may have answered the question. However, if the trypsin can cause significant difference in findings, why they did not describe it in the original manuscript!

2) In Fig. 3-A in the original manuscript, intracellular NP was detected 3 h, not 16 h (which is now cited as the time for first detection of NP) after IAV infection. Such serious mistakes in the initial manuscript are quite unacceptable and throws doubt on the reliability of the findings in this work!

3) The author did not answer our question 7 and 8. They just said they made very important mistake for question 7, and did new experiments. The absence of effect of PVF-TET on the NP expression can be due to the low antiviral efficiency of PVF-TET when using MOI 2. Obviously, the PVF-TET's antiviral efficiency is low, which may not significantly affect the NP ratio in infected cells. If they want to make a clear case to show that the PVF-TET cannot affect the binding and entry, at least another method is needed to show that the PVF-tet does not affect the viral RNA replication within one life cycle. This is so important for the conclusion of the whole study that the authors must provide very clear evidence. However, the author only used cell survival at 24h post infection,

which can therefore only claim that PVF-TET have antiviral activity, but does not indicate that PVF-TET cannot affect entry even with the time of addition assay.

4) A major contradiction is that PVF-TET can block 100% of virus release in Fig 4E, but the antiviral activity shown in the preceding in vitro and then the mouse experiments is poor. The authors need to explain this contradiction very clearly

B. But the most important problems are:

1) They screened out the PVF-tet by a binding assay to HA receptor and showed that the PVF-TET can really bind to the HA receptor binding domain. Then they claimed that the PVF-tet did not bind to viral HA, but only bind to newly synthesized intracellular HA. How can this happen at all? They must support their claim by showing structural differences between the HA receptor binding domains between the newly synthesized intracellular HA and that of the HA on released virus? This is a key problem in the study!!!. They paid a lot of attention on the downstream experiments to claim that the PVF-tet only bind to newly synthesized intracellular HA, but they did not clearly verify if the PVF-tet cannot bind to virus and affect viral entry. This must be answered before the paper can be considered for publication. Otherwise, the conclusion can be wrong.

2) The antiviral efficacy of PVF-TET was really not good, only protected 50% of H1N1-infected mice when PVF-TET was administrated at 6 h post infection. There are many published antiviral compounds and antiviral peptides which are better than this protection and put doubt on the significance of this findings.

3) 3) Moreover some of the experimental design are not reasonable. Note that if the authors cannot claim any new antiviral mechanism and show no data of broad protection in mice infected by other influenza virus, it would be doubtful if the work can be considered for publication in a reputable journal. I am sorry that I cannot be helpful in finding a way out for the authors on this occasion.

Response: We would like to thank the reviewer #1 for reviewing our manuscript and offering the fair and constructive criticisms that help strengthen our conclusion. We have addressed all of the reviewer's concerns in the point-by-point responses as follows.

A. The authors have answered the majority of our questions. But they did not answer some critical ones but just deleted their previous figs or admitted that they have made some mistakes.

1)For example, the Fig. 2b and Fig.3 (right figure) used 10 MOI and 24h, but the cell control viability was 20% in Fig. 2b and 50% in Fig 3c right. What is the problem? The PVF-Tet cannot efficiently protect cells in low MOI (Fig.2c left), but more efficiently protect cells in high MOI. What is the reason? After they changed their legends in Fig 2, with or without trypsin, it may have answered the question. However, if the trypsin can cause significant difference in findings, why they did not describe it in the original manuscript!

Response: The viability values of cells cultured without any inhibitors under the condition of single-cycle infection were 43 ± 2 , 24 ± 4 , 19 ± 6 , and $55 \pm 3\%$, in Figs. 1b, 1f, 2b and 2c-right panel in the revised manuscript, respectively, each of which was determined by three independent experiments. In each group, the experiments were sequentially performed under the similar conditions, such as subculture conditions of cells and virus conditions, to minimize the biological errors. These average values, however, varied among the different groups, as the reviewer points out. To clarify this point, we plotted all the data obtained so far as shown in the attached Fig. 1. Under the condition of single-cycle infection (10 MOI for 24 h), the average cell viability value in each group is apt to vary compared to the condition of multi-cycle infection (0.001 MOI for 48 h). A possible reason is that under the condition of single-cycle infection, where infectious progeny virus is not produced, the cell viability can be more directly affected by the condition of the virus used, probably explaining the different values of cell viability among these groups, such as Figs. 1b, 1f, 2b or 2c-right panel.

As the reviewer suggests, PVF-tet did not efficiently protect cells under the low MOI

condition, because PVF-tet can be easily cleaved by trypsin present in the culture medium as shown in Supplementary Fig. 3 in the revised manuscript. Although we already described the presence of trypsin in the culture medium used for the low MOI condition in the original manuscript (lines 137-143 and 441-445), we added this information in the legend for Fig. 2c in the 1st version of the revised manuscript to further clarify this point. Under the low MOI condition, the inhibitory effect of (D)PVF-tet was also diminished (Fig. 2c left panel) compared to the high MOI condition (Fig. 2c right panel), even though (D)PVF-tet is resistant to trypsin. To clarify this point, we examined the effect of FBS, which is present in the culture medium for the high MOI condition but not in the medium for the low MOI condition, on the incorporation of (D)PVF-tet into MDCK cells. As shown in the attached Fig. 2, FITC-labeled (D)PVF-tet was more efficiently incorporated into MDCK cells through endocytosis, when the cells were cultured in the medium for the high MOI condition compared to the medium for the low MOI condition. A previous report indicates that the incorporation of a cell permeable peptide can be affected by the presence of FBS (Bioconjugate Chem., 19, 656-664, 2008), explaining the reduction of the anti-viral activity of (D)PVF-tet under the low MOI condition.

2) In Fig. 3-A in the original manuscript, intracellular NP was detected 3 h, not 16 h (which is now cited as the time for first detection of NP) after IAV infection. Such serious mistakes in the initial manuscript are quite unacceptable and throws doubt on the reliability of the findings in this work!

Response: In Fig. 3a in the original manuscript and Fig. 3a upper panel in the revised manuscript, we measured the number of NP-positive cells, in which newly synthesized NP was exclusively detected in the nucleus 3 h after infection as shown in Fig. 3a lower panel in the revised manuscript. In contrast, after 16 h of infection, NP was mainly detected in the cytosol as shown in Fig. 3d in the revised manuscript. Because these two images are completely different, the incubation time in the Fig. 3a cannot be 16 h after infection. We sincerely apologize to the reviewers for the careless mistake.

3) The author did not answer our question 7 and 8. They just said they made very important mistake for question 7, and did new experiments. The absence of effect of PVF-TET on the NP expression can be due to the low antiviral efficiency of PVF-TET when using MOI 2. Obviously, the PVF-TET's antiviral efficiency is low, which may not significantly affect the NP ratio in infected cells. If they want to make a clear case to show that the PVF-TET cannot affect the binding and entry, at least another method is needed to show that the PVF-tet does not affect the viral RNA replication within one life cycle. This is so important for the conclusion of the whole study that the authors must provide very clear evidence. However, the author only used cell survival at 24h post infection, which can therefore can only claim that PVF-TET have antiviral activity, but does not indicate that PVF-TET cannot affect entry even with the time of addition assay.

Response: Following the reviewer's suggestion, we examined the effect of PVF-tet on the viral mRNA synthesis within one life cycle. As shown in Supplementary Fig. 4b in the 2nd version of the revised manuscript, the relative amount of NP mRNA was markedly increased in a time-dependent manner after IAV infection. Even in a very early stage of infection (after 1 h incubation at 37 °C), PVF-tet did not affect the NP mRNA level, while bafilomycin A1 completely suppressed the induction. In addition, we also examined the effect of PVF-tet on the fusion process after endocytosis of the virus. We prepared IAV particles labeled with two different lipophilic tracers, C18-Rhodamine B (red fluorescence) and C18-DiO (green fluorescence), which can be detected as red particles because under this condition fluorescence of DiO is quenched by that of Rhodamine B through the Förster resonance energy transfer. After fusion of the virus with the endosome membrane, these two tracers diffuse into the endosome membrane, allowing the endosome to be detected by both fluorescence as a result of DiO dequenching. As shown in Supplementary Fig. 4a in the 2nd version of the revised manuscript, after 90 min incubation at 37 °C co-localization of red- and green-fluoresce was efficiently detected in vehicle and PVF-tet treated cells, but not in bafilomycin A1 treated cells, further indicating that PVF-tet does not affect the binding or entry of the virus. We added Supplementary Figs. 4a and 4b, and accordingly changed the Results (lines 162-164) in the 2nd version of the revised manuscript.

4) A major contradiction is that PVF-TET can block 100% of virus release in Fig 4E, but the antiviral activity shown in the preceding in vitro and then the mouse experiments is poor. The authors need to explain this contradiction very clearly

Response: To further clarify the inhibitory effect of PVF-tet on the vesicular transport of newly synthesized HA to the plasma membrane, we added low magnification images of cells. As shown Supplementary Fig. 5c left panel in the 2nd version of the revised manuscript, PVF-tet clearly reduced the amount of cell surface HA (green), and concomitantly induced the accumulation of intracellular HA (red) in the vacuole-like structure, indicating that PVF-tet sequestered newly synthesized HA in the structure to hamper its vesicular transport to the plasma membrane. Under the same condition, where MDCK cells were infected with IAV at 10 MOI for 1 h and then cultured with PVF-tet, PVF-tet (20 μ M) efficiently inhibited the IAV cytopathicity to increase the cell viability from $18 \pm 4\%$ to $103 \pm 9\%$, or from $35 \pm 3\%$ to $85 \pm 9\%$ 24 h after infection, as shown in Fig. 6d or Supplementary Fig. 10c in the 2nd version of the revised manuscript, respectively. This protective effect of PVF-tet is consistent with the marked inhibition of the intracellular transport of newly synthesized HA to the plasma membrane to suppress the viral propagation. We added Supplementary Fig. 5c left panel, and accordingly changed the Results (lines 190-192) in the 2nd version of the revised manuscript.

As the reviewer points out, (D)PVF-tet rescued 50% of mice from mortality, while zanamivir rescued 100% mice (Supplementary Fig. 12a in the 2nd version of the revised manuscript), although (D)PVF-tet efficiently inhibited IAV cytopathicity with comparable efficacy to zanamivir in MDCK cells (Fig. 1f in the revised manuscript). The presence of proteases, such as TM6RSS4, trypsin, and HAT in the respiratory tract (Curr Opin Virol. 24, 16-24, 2017) might affect the stability of (D)PVF-tet after intranasal administration to reduce its efficacy in vivo. In addition, as shown in the attached Fig. 2, the efficacy of incorporation of PVF-tet into the target cells can be affected by the presence of serum proteins, so the epithelial environment of the alveolar space, where the alveolar type-II cells, major target cells of IAV infection, are present with large amount of pulmonary surfactant but less amount of serum proteins, might also negatively affect the incorporation of PVF-tet to the alveolar type-II cells. However, the precise mechanisms

remains to be clarified.

B. But the most important problems are:

1) They screened out the PVF-tet by a binding assay to HA receptor and showed that the PVF-TET can really bind to the HA receptor binding domain. Then they claimed that the PVF-tet did not bind to viral HA, but only bind to newly synthesized intracellular HA. How can this happen at all? They must support their claim by showing structural differences between the HA receptor binding domains between the newly synthesized intracellular HA and that of the HA on released virus? This is a key problem in the study!!!. They paid a lot of attention on the downstream experiments to claim that the PVF-tet only bind to newly synthesized intracellular HA, but they did not clearly verify if the PVF-tet cannot bind to virus and affect viral entry. This must be answered before the paper can be considered for publication. Otherwise, the conclusion can be wrong.

Response: To address the reviewer's concern, we examined the inhibitory effect of PVF-tet on the hemagglutinating activity of HA present on infectious IAV particles. As shown in Supplementary Fig. 4e in the 2nd version of the revised manuscript, PVF-tet did not inhibit the hemagglutinating activity, while fetuin efficiently inhibited the activity in a dose dependent manner. This observation indicates that PVF-tet does not bind to the functioning HA present on IAV particles, although PVF-tet efficiently binds to the functioning recombinant HA and then inhibits its receptor-binding activity (as shown in Fig. 1e in the revised manuscript). The difference is not attributed to the structural difference of the receptor-binding region, because these HAs are functionally equivalent. A major difference between these HAs is the state of HA; HA present on infectious IAV particles is cleaved into HA1 and HA2 fragments, but recombinant HA or newly synthesized intracellular HA is not. Thus, the resultant HA2 fragment might hamper the accessibility and/or binding of PVF-tet to the receptor-binding region of the HA1 fragment present on infectious IAV. Previous studies based on the X-ray crystallography demonstrate that the cleavage of HA does not affect the structure of the receptor-binding region of HA (Science, 303, 1866-1870. 2004; Science, 303, 1838-1842. 2004), further supporting this contention.

Consistently, proteolytic cleavage of recombinant HA into HA1 and HA2 fragments markedly reduced the binding of PVF-tet to HA as shown in Supplementary Fig. 4d in the 2nd version of the revised manuscript. However, the precise mechanism by which the cleavage of HA results in the reduction of the PVF-tet binding to HA remains to be clarified. We added Supplementary Fig. 4e, and accordingly changed the Results (lines 173-175) in the 2nd version of the revised manuscript.

As shown in the response to the reviewer's comment A-3, we performed additional experiments, and showed that PVF-tet did not affect the binding and fusion of IAV or also NP mRNA level even in a very early stage of infection.

2) The antiviral efficacy of PVF-TET was really not good, only protected 50% of H1N1-infected mice when PVF-TET was administrated at 6 h post infection. There are many published antiviral compounds and antiviral peptides which are better than this protection and put doubt on the significance of this findings.

Response: Almost all the synthetic HA inhibitors developed so far, including sialic acid polymers or HA-binding peptides (ref. 12-14 in the 2nd version of the revised manuscript; *Biochemistry*, 32, 2967-2978, 1993; *Virology*, 226, 66-76, 1996; *Virology*, 248, 264-274, 1998; *J Virol.*, 80, 11960-11967, 2006; *Int J Biol Sci.*, 5, 543-548, 2009; *J Virol.*, 84, 4277-4288, 2010; *J Med Chem.*, 57, 8332-8339, 2014; *J Virol.*, 88, 1447-1460, 2014), are targeting HA present on infectious IAV particles to inhibit viral binding and/or fusion process. PVF-tet is the first compound that targets newly synthesized HA and inhibits viral propagation *in vitro* and *in vivo* through a unique mechanism involving autophagy induction. Furthermore, based on the unique mechanism underlying the function of PVF-tet, we identified another type of anti-viral amphisome, which can be induced even in the absence of PVF-tet. We believe that these findings could be useful to provide a new strategy to combat IAV infection.

3) Moreover some of the experimental design are not reasonable. Note that if the authors cannot claim any new antiviral mechanism and show no data of broad protection in mice

infected by other influenza virus, it would be doubtful if the work can be considered for publication in a reputable journal. I am sorry that I cannot be helpful in finding a way out for the authors on this occasion.

Response: To apply the unique anti-viral concept identified here to another type of influenza virus, it is necessary to develop a tetravalent peptide customized to HA of each intended virus. We have already found that PVF-tet bound to recombinant HA derived from highly virulent avian H5N1 A/chicken/Yamaguchi/7/2004 strain, which exclusively binds to α 2-3 sialyllactose-polymer, and also to its human-tropic mutant with two mutations (Q226L and G228S), which exclusively binds to α 2-6 sialyllactose-polymer. Thus, it is highly possible that based on the HA-binding motif of PVF-tet, binding motifs customized to HA of H5N1 virus could be developed by using affinity maturation-based screening of tetravalent peptides synthesized on a membrane. On the other hand, we have already found that human-tropic HA of H5N1 virus, which was overexpressed in MDCK cells stably overexpressing ABCA3, was also accumulated in ABCA3-positive structure as shown in the attached Fig. 3 (confidential data), suggesting that the strategy to stimulate the formation of the anti-viral amphisomes might be also useful to inhibit the propagation of another type of influenza virus. In this paper, however, we would like to focus on H1N1 virus, because we believe that the significance of our findings is to propose a novel concept to combat IAV infection.

Attached Figure 1

Attached Figure 2

Attached Figure 3 **[Confidential]**

Attached Figure 1. The cell viability under high or low MOI condition.

MDCK cells were infected with IAV strain PR8 at 10 or 0.001 MOI for 24 h or 48 h, respectively, without any inhibitors. Data are presented as a percentage of the control value without infection. Individual dots indicate the values obtained from independent experiments. A group with the same color represents a group of sequentially performed experiments in each figure indicated. Box-plot elements are defined as follows: center line, median; box limits, upper and lower quartiles; whiskers, 1.5x interquartile range.

Attached Figure 2. The effect of FBS on the incorporation of (D)PVF-tet.

MDCK cells were incubated with FITC-labeled (D)PVF-tet (2.5 μ M) and Alexa594-labeled dextran (250 μ g/ml) in MEM supplemented with 10% heat-inactivated FBS, 50 units/ml penicillin and 50 μ g/ml streptomycin (Trypsin(-)-MEM), or serum-free MEM supplemented with 1 μ g/ml trypsin, 0.2% BSA, 25 mM HEPES, 50 units/ml penicillin, and 50 μ g/ml streptomycin (Trypsin(+)-MEM) for 1 h at 37 °C. Fluorescent images were analyzed using laser scanning confocal microscopy. Dashed lines indicate the nucleus. Scale bar represents 20 μ m.

Attached Figure 3. Accumulation of human-tropic HA of H5N1 virus in MDCK cells stably overexpressing ABCA3.

MDCK-derived clones stably overexpressing ABCA3mVenus cells (clone 6) were transfected with the respective expression vectors (pcDNA3.1(-)) encoding HA of H1N1 PR8 (H1 HA; upper panel) or HA of H5N1 A/chicken/Yamaguchi/7/2004 strain (H5 HA; lower panel) using ScreenFect A plus transfection reagent. After 48 h, intracellular localization of each HA and ABCA3mVenus was analyzed by immunocytochemistry. Dashed lines indicate the nucleus. Scale bar represents 20 μ m.

Response to the comments of the reviewer #2:

Reviewer #2 (Remarks to the Author):

The authors have adequately and thoroughly addressed all my comments. This includes a few new experiments which create added value. I have only two minor comments:

- Dextran is a marker of macropinocytosis (not: micropinocytosis)
- To clearly show the experimental difference between Fig. 7a and Suppl. Fig. 11b, specify in the Legend the time when (D)PVF-tet was administered (together with virus or at 6 h p.i.).

Response: We would like to thank the reviewer #2 for reviewing our manuscript and offering the constructive suggestions. We have followed the reviewer's suggestions in the point-by-point responses as follows.

- Dextran is a marker of macropinocytosis (not: micropinocytosis)

Response: Following the reviewer's suggestion, we replaced the word "micropinocytosis" with "macropinocytosis" in the 2nd version of the revised manuscript.

- To clearly show the experimental difference between Fig. 7a and Suppl. Fig. 11b, specify in the Legend the time when (D)PVF-tet was administered (together with virus or at 6 h p.i.).

Response: Following the reviewer's suggestion, we clarified the experimental condition in the Legend for Fig. 7a, b (line 1124) and Supplementary Fig. 12a, c, d-e in the 2nd version of the revised manuscript.

Response to the comments of the reviewer #3:

Reviewer #3 (Remarks to the Author):

Manuscript Nr: NCOMMS-18-17953A

Omi et al., “The Inducible Amphisome Isolates Viral Hemagglutinin and Defends Against Influenza A Virus Infection”

The authors demonstrate that tetravalent peptides that have been identified in screens for influenza virus hemagglutinin (HA) binders inhibit viral replication. This inhibition is associated with HA sequestration into a vesicular compartment that contains lysosomal (LAMP1) and autophagosomal (LC3) markers. HA is associated with lipid rafts and overexpression of the lamellar body inducing ABCA3 transporter assembles similar compartments and inhibits influenza virus replication. From these data the authors conclude that an amphisome results from these treatments that could be explored for therapeutic approaches against influenza virus.

In their revised manuscript version, the authors have addressed most of my comments. Mainly, they have characterized the amphisomes in more detail, they document reduced HA surface expression after HA aggregation, they perform more specific ULK1 and PI3 kinase silencing and demonstrate also LC3 accumulation in vivo. These additional experimental data have significantly strengthened the presented manuscript.

Response: We would like to thank the reviewer #3 for reviewing our manuscript and offering the constructive suggestions.

Reviewers' comments:

Reviewer #1 (Remarks to the Author):

Dear Editor,

Thanks for asking me to review NCOMMS-18-17953B.

The authors have tried to address most of the questions raised. But the publication of a paper in Nature communication requires a novel observation with proof of mechanism by up-to-standard experiments.

There are very important omissions or mistakes in the paper which put serious doubts on the conclusions of this paper:

1. The authors have not been able to demonstrate the mechanism of why PVF-TET binding only to newly synthesized HA and not to proteolytically cleaved HA. The authors explained that HA2 may hamper the binding of PVF-tet to the receptor binding sites, but they also mentioned that X-ray demonstrated that the cleavage of HA does not affect the structure of the receptor binding site. So the authors in effect provided a reason not supporting their explanation. In contrast, the structure data indicated that HA2 will not hamper the receptor binding site located at HA1 head region.

Firstly, they did not answer the key question why PVF-tet only binds to newly synthesized HA, but not binds to released viral HA. They did not show clear evidence of PVF-tet cannot bind to viral HA (Sup Fig 4c just indicated that the binding of PVF-tet is weaker than that of fetuin).

Secondly, they did not clearly indicated that PVF-tet reduced the viral release in the supernatant by counting plaque forming unit. But they used indirect methods of cell viability, NP and HA protein staining. These are not the good methods for quantification the reduction of viral release. If the author's conclusion is the PVF-TET can target new viral HA at the later stage, why did the authors pretreat cells before viral infection and then test the viral release at 16h post-infection in Fig.2d? They should treat the infected cells with PVF-TET after viral infection to see if the PVF-TET can significantly inhibit viral release. In addition, new Supplementary Fig. 4e did not include the important control. Without virus, can PVF-tet block the TRBC precipitation?

2. The authors have not tried to reproduce the “so called novel antiviral machineries amphisomes” by transfecting the host cells with labelled HA and treat it with the antiviral peptide PVF-TET to reproduce the amphisomes and see if the treated cells become resistant to infection by influenza virus. Without such experiments, how can we claim that a new antiviral machinery is formed by PVF-TET and non-proteolytically activated (newly synthesized) HA.

3. I quote from the author’s response to my query, ‘A possible reason is that under the condition of single-cycle infection, where infectious progeny virus is not produced’. This is completely wrong. After 24 hour infection of cell line by 10MOI, there must have many virus replication cycle already. Examination of a single cycle infection can only take place latest by 6 to 8 hours post infection.

Reviewer #2 (Remarks to the Author):

This is incorporated in my attached report below:

Reviewer #1**Authors' response in blue**
Input of Reviewer #2 in red

1) For example, the Fig. 2b and Fig.3 (right figure) used 10 MOI and 24h, but the cell control viability was 20% in Fig. 2b and 50% in Fig 3c right. What is the problem? The PVF-Tet cannot efficiently protect cells in low MOI (Fig.2c left), but more efficiently protect cells in high MOI. What is the reason? After they changed their legends in Fig 2, with or without trypsin, it may have answered the question. However, if the trypsin can cause significant difference in findings, why they did not describe it in the original manuscript!

Response: The viability values of cells cultured without any inhibitors under the condition of single-cycle infection were 43 ± 2 , 24 ± 4 , 19 ± 6 , and $55 \pm 3\%$, in Figs. 1b, 1f, 2b and 2c-right panel in the revised manuscript, respectively, each of which was determined by three independent experiments. In each group, the experiments were sequentially performed under the similar conditions, such as subculture conditions of cells and virus conditions, to minimize the biological errors. These average values, however, varied among the different groups, as the reviewer points out. To clarify this point, we plotted all the data obtained so far as shown in the attached Fig. 1. Under the condition of single-cycle infection (10 MOI for 24 h), the average cell viability value in each group is apt to vary compared to the condition of multi-cycle infection (0.001 MOI for 48 h). A possible reason is that under the condition of single-cycle infection, where infectious progeny virus is not produced, the cell viability can be more directly affected by the condition of the virus used, probably explaining the different values of cell viability among these groups, such as Figs. 1b, 1f, 2b or 2c-right panel. As the reviewer suggests, PVF-tet did not efficiently protect cells under the low MOI 4 condition, because PVF-tet can be easily cleaved by trypsin present in the culture medium as shown in Supplementary Fig. 3 in the revised manuscript. Although we already described the presence of trypsin in the culture medium used for the low MOI condition in the original manuscript (lines 137-143 and 441-445), we added this information in the legend for Fig. 2c in the 1st version of the revised manuscript to further clarify this point. Under the low MOI condition, the inhibitory effect of (D)PVF-tet was also diminished (Fig. 2c left panel) compared to the high MOI condition (Fig. 2c right panel), even though (D)PVF-tet is resistant to trypsin. To clarify this point, we examined the effect of FBS, which is present in the culture medium for the high MOI condition but not in the medium for the low MOI condition, on the incorporation of (D)PVF-tet into MDCK cells. As shown in the attached Fig. 2, FITC-labeled (D)PVF-tet was more efficiently incorporated into MDCK cells through endocytosis, when the cells were cultured in the medium for the high MOI condition compared to the medium for the low MOI condition. A previous report indicates that the incorporation of a cell permeable peptide can be affected by the presence of FBS (Bioconjugate Chem., 19, 656-664, 2008), explaining the reduction of the anti-viral activity of (D)PVF-tet under the low MOI condition.

- I am not really concerned about some inter-experimental variability in the cell viability of the virus control.
- Trypsin issue: the authors could (i) specify in Fig. 2c above the charts the conditions, i.e. plus or minus trypsin, % FBS and low or high MOI. (ii) clarify in line 142: if (D)-PVF-tet is resistant to trypsin, then why is its activity not better at the low MOI? Was FBS present in this experiment?

The authors should revise the text and figures to provide concise, clear and consistent descriptions of experimental 'details' that are important to understand the manuscript's messages. They now show data from variations in experimental setups, and this makes the text confusing.

2) In Fig. 3-A in the original manuscript, intracellular NP was detected 3 h, not 16 h (which is now cited as the time for first detection of NP) after IAV infection. Such serious mistakes in the initial manuscript are quite unacceptable and throws doubt on the reliability of the findings in this work!

Response: In Fig. 3a in the original manuscript and Fig. 3a upper panel in the revised manuscript, we measured the number of NP-positive cells, in which newly synthesized NP was exclusively detected in the nucleus 3 h after infection as shown in Fig. 3a lower panel in the revised manuscript. In contrast, after 16 h of infection, NP was mainly detected in the cytosol as shown in Fig. 3d in the revised manuscript. Because these two images are completely different, the incubation time in the Fig. 3a cannot be 16 h after infection. We sincerely apologize to the reviewers for the careless mistake.

The wrong time point is an unfortunate mistake but I believe that the NP pictures are self-explanatory.

3) The author did not answer our question 7 and 8. They just said they made very important mistake for question 7, and did new experiments. The absence of effect of PVF-TET on the NP expression can be due to the low antiviral efficiency of PVF-TET when using MOI 2. Obviously, the PVF-TET's antiviral efficiency is low, which may not significantly affect the NP ratio in infected cells. If they want to make a clear case to show that the PVF-TET cannot affect the binding and entry, at least another method is needed to show that the PVF-tet does not affect the viral RNA replication within one life cycle. This is so important for the conclusion of the whole study that the authors must provide very clear evidence. However, the author only used cell survival at 24h post infection, which can therefore can only claim that PVF-TET have antiviral activity, but does not indicate that PVF-TET cannot affect entry even with the time of addition assay.

Response: Following the reviewer's suggestion, we examined the effect of PVF-tet on the viral mRNA synthesis within one life cycle. As shown in Supplementary Fig. 4b in the 2nd version of the revised manuscript, the relative amount of NP mRNA was markedly increased in a time-dependent manner after IAV infection. Even in a very early stage of infection (after 1 h incubation at 37 °C), PVF-tet did not affect the NP mRNA level, while bafilomycin A1 completely suppressed the induction. In addition, we also examined the effect of PVF-tet on the fusion process after endocytosis of the virus. We prepared IAV particles labeled with two different lipophilic tracers, C18-Rhodamine B (red fluorescence) and C18-DiO (green fluorescence), which can be detected as red particles because under this condition fluorescence of DiO is quenched by that of Rhodamine B through the Förster resonance energy transfer. After fusion of the virus with the endosome membrane, these two tracers diffuse into the endosome membrane, allowing the endosome to be detected by both fluorescence as a result of DiO dequenching. As shown in Supplementary Fig. 4a in the 2nd version of the revised manuscript, after 90 min incubation at 37 °C co-localization of red- and green-fluoresce was efficiently detected in vehicle and PVF-tet treated cells, but not in bafilomycin A1 treated cells, further indicating that PVF-tet does not affect the binding or entry of the virus. We added Supplementary Figs. 4a and 4b, and accordingly changed the Results (lines 162-164) in the 2nd version of the revised manuscript.

In lines 162-164, specify: did not affect the fusion process, as monitored with fluorescently labeled virus, ...

In my view, this experiment adequately proves that PVF-tet does not act during virus entry.

- 4) A major contradiction is that PVF-TET can block 100% of virus release in Fig 4E, but the antiviral activity shown in the preceding in vitro and then the mouse experiments is poor. The authors need to explain this contradiction very clearly

Response: To further clarify the inhibitory effect of PVF-tet on the vesicular transport of newly synthesized HA to the plasma membrane, we added low magnification images of cells. As shown Supplementary Fig. 5c left panel in the 2nd version of the revised manuscript, PVF-tet clearly reduced the amount of cell surface HA (green), and concomitantly induced the accumulation of intracellular HA (red) in the vacuole-like structure, indicating that PVF-tet sequestered newly synthesized HA in the structure to hamper its vesicular transport to the plasma membrane. Under the same condition, where MDCK cells were infected with IAV at 10 MOI for 1 h and then cultured with PVF-tet, PVF-tet (20 μ M) efficiently inhibited the IAV cytopathicity to increase the cell viability from $18 \pm 4\%$ to $103 \pm 9\%$, or from $35 \pm 3\%$ to $85 \pm 9\%$ 24 h after infection, as shown in Fig. 6d or Supplementary Fig. 10c in the 2nd version of the revised manuscript, respectively. This protective effect of PVF-tet is consistent with the marked inhibition of the intracellular transport of newly synthesized HA to the plasma membrane to suppress the viral propagation. We added Supplementary Fig. 5c left panel, and accordingly changed the Results (lines 190-192) in the 2nd version of the revised manuscript. As the reviewer points out, (D)PVF-tet rescued 50% of mice from mortality, while zanamivir rescued 100% mice (Supplementary Fig. 12a in the 2nd version of the revised manuscript), although (D)PVF-tet efficiently inhibited IAV cytopathicity with comparable efficacy to zanamivir in MDCK cells (Fig. 1f in the revised manuscript). The presence of proteases, such as TMRRSS4, trypsin Clara, and HAT in the respiratory tract (Curr Opin Virol. 24, 16-24, 2017) might affect the stability of (D)PVF-tet after intranasal administration to reduce its efficacy in vivo. In addition, as shown in the attached Fig. 2, the efficacy of incorporation of PVF-tet into the target cells can be affected by the presence of serum proteins, so the epithelial environment of the alveolar space, where the alveolar type-II cells, major target cells of IAV infection, are present with large amount of pulmonary surfactant but less amount of serum proteins, might also negatively affect the incorporation of PVF-tet to the alveolar type-II cells. However, the precise mechanisms remains to be clarified.

It looks to me that the authors' long answer does not address the Reviewer's comment (but is there a Figure 4e? Only Fig. 2d shows antiviral activity based on virus yield?). The Reviewer seems critical about how relevant is the antiviral activity of (D)PVF-tet, especially when compared to that of a well-established drug like zanamivir.

I am not sure that there is a contradiction. Fig. 2d shows a fold increase in the virus control of about 600 which is reduced to about 50 under PVF-tet. I thus estimate that PVF-tet causes about one-log₁₀ reduction in virus yield, which is not a very impressive antiviral effect.

In this respect, I agree with the Reviewer that the (D)PVF-tet concept does not result in a very strong antiviral effect.

5) They screened out the PVF-tet by a binding assay to HA receptor and showed that the PVF-TET can really bind to the HA receptor binding domain. Then they claimed that the PVF-tet did not bind to viral HA, but only bind to newly synthesized intracellular HA. How can this happen at all? They must support their claim by showing structural differences between the HA receptor binding domains between the newly synthesized intracellular HA and that of the HA on released virus? This is a key problem in the study!!!. They paid a lot of attention on the downstream experiments to claim that the PVF-tet only bind to newly synthesized intracellular HA, but they did not clearly verify if the PVF-tet cannot bind to virus and affect viral entry. This must be answered before the paper can be considered for publication. Otherwise, the conclusion can be wrong.

Response: To address the reviewer's concern, we examined the inhibitory effect of PVF-tet on the hemagglutinating activity of HA present on infectious IAV particles. As shown in Supplementary Fig. 4e in the 2nd version of the revised manuscript, PVF-tet did not inhibit the hemagglutinating activity, while fetuin efficiently inhibited the activity in a dose dependent manner. This observation indicates that PVF-tet does not bind to the functioning HA present on IAV particles, although PVF-tet efficiently binds to the functioning recombinant HA and then inhibits its receptor-binding activity (as shown in Fig. 1e in the revised manuscript). The difference is not attributed to the structural difference of the receptor-binding region, because these HAs are functionally equivalent. A major difference between these HAs is the state of HA; HA present on infectious IAV particles is cleaved into HA1 and HA2 fragments, but recombinant HA or newly synthesized intracellular HA is not. Thus, the resultant HA2 fragment might hamper the accessibility and/or binding of PVF-tet to the receptor-binding region of the HA1 fragment present on infectious IAV. Previous studies based on the X-ray crystallography demonstrate that the cleavage of HA does not affect the structure of the receptor-binding region of HA (Science, 303, 1866-1870. 2004; Science, 303, 1838-1842. 2004), further supporting this contention.

Consistently, proteolytic cleavage of recombinant HA into HA1 and HA2 fragments markedly reduced the binding of PVF-tet to HA as shown in Supplementary Fig. 4d in the 2nd version of the revised manuscript. However, the precise mechanism by which the cleavage of HA results in the reduction of the PVF-tet binding to HA remains to be clarified. We added Supplementary Fig. 4e, and accordingly changed the Results (lines 173-175) in the 2nd version of the revised manuscript.

As shown in the response to the reviewer's comment A-3, we performed additional experiments, and showed that PVF-tet did not affect the binding and fusion of IAV or also NP mRNA level even in a very early stage of infection.

This is to me an extremely valid Reviewer comment that must be solved.

The text is indeed very confusing and unclear regarding the interaction between PVF-tet and HA. PVF-tet was identified by an HA binding assay (with recombinant HA0? This is not clearly mentioned). The compound is not active in the hemagglutination assay (with virus in which presumably most HA is present under the form of HA1/HA2 although we are not sure about this) but is, strangely, active in Fig. 1e showing a binding assay with recombinant HA0 (???) and a receptor mimic.

The authors are right that the receptor binding site is not different between HA0 and HA1/HA2. Thus, they hypothesize that PVF-tet cannot bind to cleaved HA1/HA2 protein whereas it does bind to uncleaved HA0. This assumption could easily be verified by SPR, comparing binding of the recombinant protein to either the HA0 or cleaved (= treated with trypsin) HA1/HA2 form.

If the hypothesis proves to be correct, how can it be explained? It is very hard to rationalize that a compound binding at a stem region (cfr many anti-HA stem antibodies) could interfere with receptor binding.

I agree with the Reviewer that this is a serious lack in the study.

6) The antiviral efficacy of PVF-TET was really not good, only protected 50% of H1N1-infected mice when PVF-TET was administrated at 6 h post infection. There are many published antiviral compounds and antiviral peptides which are better than this protection and put doubt on the significance of this findings.

Response: Almost all the synthetic HA inhibitors developed so far, including sialic acid polymers or HA-binding peptides (ref. 12-14 in the 2nd version of the revised manuscript; *Biochemistry*, 32, 2967-2978, 1993; *Virology*, 226, 66-76, 1996; *Virology*, 248, 264-274, 1998; *J Virol.*, 80, 11960-11967, 2006; *Int J Biol Sci.*, 5, 543-548, 2009; *J Virol.*, 84, 4277-4288, 2010; *J Med Chem.*, 57, 8332-8339, 2014; *J Virol.*, 88, 1447-1460, 2014), are targeting HA present on infectious IAV particles to inhibit viral binding and/or fusion process. PVF-tet is the first compound that targets newly synthesized HA and inhibits viral propagation *in vitro* and *in vivo* through a unique mechanism involving autophagy induction. Furthermore, based on the unique mechanism underlying the function of PVF-tet, we identified another type of anti-viral amphisome, which can be induced even in the absence of PVF-tet. We believe that these findings could be useful to provide a new strategy to combat IAV infection.

It seems not totally fair to compare a new antiviral concept like PVF-tet (which leads to HA sequestering in amphisomes) with more established small molecule concepts in the literature. In Fig. 7e, there is less body weight loss in mice receiving the peptide and virus together; these mice have a survival rate of 50%.

The authors do not really present (D)PVF-tet as a strong antiviral compound, rather they use the molecule to identify the inducible amphisome as a new antiviral concept. I agree with this presentation.

7) Moreover some of the experimental design are not reasonable. Note that if the authors cannot claim any new antiviral mechanism and show no data of broad protection in mice infected by other influenza virus, it would be doubtful if the work can be considered for publication in a reputable journal. I am sorry that I cannot be helpful in finding a way out for the authors on this occasion.

Response: To apply the unique anti-viral concept identified here to another type of influenza virus, it is necessary to develop a tetravalent peptide customized to HA of each intended virus. We have already found that PVF-tet bound to recombinant HA derived from highly virulent avian H5N1 A/chicken/Yamaguchi/7/2004 strain, which exclusively binds to α 2-3 sialyllactose-polymer, and also to its human-tropic mutant with two mutations (Q226L and G228S), which exclusively binds to α 2-6 sialyllactose-polymer. Thus, it is highly possible that based on the HA-binding motif of PVF-tet, binding motifs customized to HA of H5N1 virus could be developed by using affinity maturation-based screening of tetravalent peptides synthesized on a membrane. On the other hand, we have already found that human-tropic HA of H5N1 virus, which was overexpressed in MDCK cells stably overexpressing ABCA3, was also accumulated in ABCA3-positive structure as shown in the attached Fig. 3 (confidential data), suggesting that the strategy to stimulate the formation of the anti-viral amphisomes might be also useful to inhibit the propagation of another type of influenza virus. In this paper, however, we would like to focus on H1N1 virus, because we believe that the significance of our findings is to propose a novel concept to combat IAV infection.

See my comment above regarding 'strong' antiviral effect.

Regarding broad activity against more than one (not only H1N1) influenza virus, the authors added a confidential Figure 3 to show amphisome localization for a human-tropic H5 HA. Why are they so reluctant to mention the data in the manuscript?

I agree with the Reviewer that adding (D)PVF-tet activity data for another virus subtype (which I prefer to experiments with HA only) would make the study much more significant. I would be satisfied with in vitro data.

In the answer above, the authors explain that PVF-test does bind to H5 HA, so it should also be active in an antiviral assay with H5N1 virus?

Summary analysis

I agree with Reviewer #1's criticisms on the confusing information regarding the precise PVF-tet/HA interaction and broad applicability of this peptide concept. The authors must solve these issues, particularly the first criticism.

Some other remarks (eg. the peptide is not as potent as some other reported inhibitors) are a lesser concern to me since the focus of the manuscript is on presenting a new concept.

Some confusing information regarding some experiments can be solved by adequate presentation of the Figures (e.g. by adding next to the Figures a visual of the experimental setup).

Response to the comments of reviewer #1:

We would like to thank reviewer #1 for reviewing our manuscript and offering comments that help strengthen our conclusions. We have provided point-by-point responses to the reviewer's concerns as follows.

Reviewers' comments:

Reviewer #1 (Remarks to the Author):

Dear Editor,

Thanks for asking me to review NCOMMS-18-17953B.

The authors have tried to address most of the questions raised. But the publication of a paper in Nature communication requires a novel observation with proof of mechanism by up-to-standard experiments.

There are very important omissions or mistakes in the paper which put serious doubts on the conclusions of this paper:

1. The authors have not been able to demonstrate the mechanism of why PVF-TET binding only to newly synthesized HA and not to proteolytically cleaved HA. The authors explained that HA2 may hamper the binding of PVF-tet to the receptor binding sites, but they also mentioned that X-ray demonstrated that the cleavage of HA does not affect the structure of the receptor binding site. So the authors in effect provided a reason not supporting their explanation. In contrast, the structure data indicated that HA2 will not hamper the receptor binding site located at HA1 head region.

Firstly, they did not answer the key question why PVF-tet only binds to newly synthesized HA, but not binds to released viral HA. They did not show clear evidence of PVF-tet cannot bind to viral HA (Sup Fig 4c just indicated that the binding of PVF-tet is weaker than that of fetuin).

Response: To answer the reviewer's concern, we performed additional experiments to further define the interaction between HA and PVF-tet. We showed that the binding of PVF-tet to HA was reduced when the HA was cleaved by trypsin (Supplementary Fig. 5a, middle panel). PVF-tet did not bind to infectious IAV particles with HA1/HA2 (PR8-HA1/HA2), but it did bind with remarkable potency to non-infectious IAV particles with HA0 (PR8-HA0), clearly indicating that the binding of PVF-tet to virus particles depends on the state of the HA on the particles (Supplementary Fig. 5b, middle panel).

As the reviewers point out, the important question is why PVF-tet efficiently binds to HA0 but not to HA1/HA2. To answer that question, we examined the effect of HA cleavage on the receptor-binding activity of fetuin, a highly sialylated glycoprotein. As shown in Supplementary Fig. 5a, the binding of fetuin to HA was also reduced by the cleavage of HA, although to a substantially lesser degree than the binding of PVF-tet. Consistently, fetuin exhibited much more binding to IAV PR8-HA0 than to infectious IAV PR8-HA1/HA2, whereas asialofetuin exhibited no binding to either virus (Supplementary Fig. 5b, right panel). The difference in the sialic acid-dependent binding of fetuin to IAV PR8-HA0 and IAV PR8-HA1/HA2 cannot be attributed to a structural difference in the receptor-binding site of HA (HA-RBS); which consists of a 130-loop, a 150-loop, a 190-helix including L194, and a 220-loop; because previous studies demonstrated that the cleavage of HA does not affect the structure of the HA-receptor binding site (RBS) (Supplementary Fig. 6a; Science, 303, 1866-1870. 2004; Science, 303, 1838-1842. 2004). Consistent with that observation, the solvent accessibility of the HA-RBS was equally low in HA0 and HA1/HA2 (Supplementary Fig. 6b, right panels). However, the crystallographic B-factors, which reflect atomic fluctuation and provide information about protein dynamics, of the HA0-RBS structure were higher than those of the HA1/HA2-RBS structure (Supplementary 6b, left panels). That observation was confirmed by a similar analysis using average B-factors based on the structural information of various H1 and H3 subtypes (Supplementary Fig. 6c). All of these results imply that the higher atomic fluctuation of the HA0-RBS compared with that of the HA1/HA2-RBS leads to a more efficient interaction with sialic acid or PVF-tet, although the crystal structures of the HA0-RBS and the HA1/HA2-RBS are the same. The cleavage of HA changes the conformation of the RBS to one with lower atomic fluctuation, which PVF-tet can hardly access, whereas the binding of sialic acid is less susceptible to the conformational change.

We added Supplementary Figs. 5 and 6 and changed the text of the Results section (lines 173-183) in the resubmitted version of the manuscript.

Secondly, they did not clearly indicated that PVF-tet reduced the viral release in the supernatant by counting plaque forming unit. But they used indirect methods of cell viability, NP and HA protein staining. These are not the good methods for quantification the reduction of viral release. If the author's conclusion is the PVF-TET can target new viral HA at the later stage, why did the authors pretreat cells before viral infection and then test the viral release at 16h post-infection in Fig.2d? They should treat the infected cells with PVF-TET after viral infection to see if the PVF-TET can significantly inhibit viral release.

Response: We performed a plaque-forming assay to evaluate the inhibitory effects of PVF-tet treatment and ABCA3 over-expression on the release of virus particles into the supernatant, as shown in Fig. 2d and Fig. 4d, respectively. Fig. 2d shows that PVF-tet efficiently inhibits IAV propagation. Fig. 3 indicates that PVF-tet inhibits IAV propagation by functioning at a later stage of infection. In light of those results, we performed experiments in which cells were treated with PVF-tet after viral infection, as shown in Figs. 5a and 6, and Supplementary Figs. 7, 9, 12b, 12c, 13b, 13c, and 16.

In addition, new Supplementary Fig. 4e did not include the important control. Without virus, can PVF-tet block the TRBC precipitation?

Response: Following the reviewer's comment, we added data showing that PVF-tet did not block RBC precipitation in the absence of IAV (Supplementary Fig. 4c).

2. The authors have not tried to reproduce the "so called novel antiviral machineries amphisomes" by transfecting the host cells with labelled HA and treat it with the antiviral peptide PVF-TET to reproduce the amphisomes and see if the treated cells become resistant to infection by influenza virus. Without such experiments, how can we claim that a new antiviral machinery is formed by PVF-TET and non-proteolytically activated (newly synthesized) HA.

Response: The formation of the anti-viral amphisome is induced by PVF-tet or by the over-expression of ABCA3 in response to IAV infection. Even if the amphisome were reproduced with transfected HA and PVF-tet, subsequent IAV infection would induce further amphisome formation depending on the amount of PVF-tet left in the cells, making it difficult to adequately evaluate the anti-viral activity of the reproduced amphisome. Moreover, the crucial point for the anti-viral activity is whether newly synthesized HA, which is produced after IAV infection, can bind with PVF-tet to be sequestered or not, regardless of the presence or absence of reproduced amphisome.

3. I quote from the author's response to my query, 'A possible reason is that under the condition of single-cycle infection, where infectious progeny virus is not produced'. This is completely wrong. After 24 hour infection of cell line by 10MOI, there must have many virus replication cycle already. Examination of a single cycle infection can only take place latest by 6 to 8 hours post infection.

Response: As described in the manuscript, we used MEM supplemented with 10% FBS, but without trypsin, as the culture medium for the single-cycle infection. It is generally well established that no infectious progeny virus can be produced under such conditions (with 10% FBS but not trypsin; Proc. Natl. Acad. Sci. USA, 95, 10224-, 1998; J. Virol., 76, 8682-, 2002). To clarify this point, we added a visual depiction of the experimental setup in the resubmitted version of the manuscript (Fig. 2b and 2c).

Response to the comments of the reviewer #2:

We would like to thank the reviewer #2 for reviewing our manuscript and offering the fair and constructive suggestions. We have followed the reviewer's suggestions in the point-by-point responses as follows.

Reviewer #1

Author's response in blue

Input of Reviewer #2 in red

1) For example, the Fig. 2b and Fig.3 (right figure) used 10 MOI and 24h, but the cell control viability was 20% in Fig. 2b and 50% in Fig 3c right. What is the problem? The PVF-Tet cannot efficiently protect cells in low MOI (Fig.2c left), but more efficiently protect cells in high MOI. What is the reason? After they changed their legends in Fig 2, with or without trypsin, it may have answered the question. However, if the trypsin can cause significant difference in findings, why they did not describe it in the original manuscript!

Response: The viability values of cells cultured without any inhibitors under the condition of single-cycle infection were 43 ± 2 , 24 ± 4 , 19 ± 6 , and $55 \pm 3\%$, in Figs. 1b, 1f, 2b and 2c- right panel in the revised manuscript, respectively, each of which was determined by three independent experiments. In each group, the experiments were sequentially performed under the similar conditions, such as subculture conditions of cells and virus conditions, to minimize the biological errors. These average values, however, varied among the different groups, as the reviewer points out. To clarify this point, we plotted all the data obtained so far as shown in the attached Fig. 1. Under the condition of single-cycle infection (10 MOI for 24 h), the average cell viability value in each group is apt to vary compared to the condition of multi-cycle infection (0.001 MOI for 48 h). A possible reason is that under the condition of single-cycle infection, where infectious progeny virus is not produced, the cell viability can be more directly affected by the condition of the virus used, probably explaining the different values of cell viability among these groups, such as Figs. 1b, 1f, 2b or 2c-right panel. As the reviewer suggests, PVF-tet did not efficiently protect cells under the low MOI 4 condition, because PVF-tet can be easily cleaved by trypsin present in the culture medium as shown in Supplementary Fig. 3 in the revised manuscript. Although we already described the presence of trypsin in the culture medium used for the low MOI condition in the original manuscript (lines 137-143 and 441-445), we added this information in the legend for Fig. 2c in the 1st version of the revised manuscript to further clarify this point. Under the low MOI condition, the inhibitory

effect of (D)PVF-tet was also diminished (Fig. 2c left panel) compared to the high MOI condition (Fig. 2c right panel), even though (D)PVF-tet is resistant to trypsin. To clarify this point, we examined the effect of FBS, which is present in the culture medium for the high MOI condition but not in the medium for the low MOI condition, on the incorporation of (D)PVF-tet into MDCK cells. As shown in the attached Fig. 2, FITC-labeled (D)PVF-tet was more efficiently incorporated into MDCK cells through endocytosis, when the cells were cultured in the medium for the high MOI condition compared to the medium for the low MOI condition. A previous report indicates that the incorporation of a cell permeable peptide can be affected by the presence of FBS (Bioconjugate Chem., 19, 656-664, 2008), explaining the reduction of the anti-viral activity of (D)PVF-tet under the low MOI condition.

- I am not really concerned about some inter-experimental variability in the cell viability of the virus control.
- Trypsin issue: the authors could (i) specify in Fig. 2c above the charts the conditions, i.e. plus or minus trypsin, % FBS and low or high MOI. (ii) clarify in line 142: if (D)-PVF-tet is resistant to trypsin, then why is its activity not better at the low MOI? Was FBS present in this experiment?

The authors should revise the text and figures to provide concise, clear and consistent descriptions of experimental 'details' that are important to understand the manuscript's messages. They now show data from variations in experimental setups, and this makes the text confusing.

Response: Following the reviewer's suggestion (i), we added the experimental setups above the charts in Fig. 2c.

FBS was excluded from the medium for the low MOI condition to promote multi-cycle infection. As shown in Supplementary Fig. 8c in the resubmitted version of the manuscript, the incorporation of PVF-tet was enhanced by the presence of FBS, which explains why (D)PVF-tet was less effective under the low MOI condition, which lacked FBS (Fig. 2c), than under the high MOI condition. That observation is consistent with a previous report indicating that cell-permeable peptides can be efficiently endocytosed into cells in the presence of FBS (Bioconjugate Chem., 19, 656-664, 2008). Following the reviewer's suggestion (ii), we clarified this point in the resubmitted version of the manuscript (line 209-215).

Following the reviewer's suggestion, we clarified experimental details in the resubmitted version of the manuscript (line 141-142, Figs. 2b-2d, and Supplementary Fig. 7b).

2) In Fig. 3-A in the original manuscript, intracellular NP was detected 3 h, not 16 h (which is now cited as the time for first detection of NP) after IAV infection. Such serious mistakes in the initial manuscript are quite unacceptable and throws doubt on the reliability of the findings in this work!

Response: In Fig. 3a in the original manuscript and Fig. 3a upper panel in the revised manuscript, we measured the number of NP-positive cells, in which newly synthesized NP was exclusively detected in the nucleus 3 h after infection as shown in Fig. 3a lower panel in the revised manuscript. In contrast, after 16 h of infection, NP was mainly detected in the cytosol as shown in Fig. 3d in the revised manuscript. Because these two images are completely different, the incubation time in the Fig. 3a cannot be 16 h after infection. We sincerely apologize to the reviewers for the careless mistake.

The wrong time point is an unfortunate mistake but I believe that the NP pictures are self-explanatory.

Response: We greatly appreciate the reviewer's comment.

3) The author did not answer our question 7 and 8. They just said they made very important mistake for question 7, and did new experiments. The absence of effect of PVF-TET on the NP expression can be due to the low antiviral efficiency of PVF-TET when using MOI 2. Obviously, the PVF-TET's antiviral efficiency is low, which may not significantly affect the NP ratio in infected cells. If they want to make a clear case to show that the PVF-TET cannot affect the binding and entry, at least another method is needed to show that the PVF-tet does not affect the viral RNA replication within one life cycle. This is so important for the conclusion of the whole study that the authors must provide very clear evidence. However, the author only used cell survival at 24h post infection, which can therefore can only claim that PVF-TET have antiviral activity, but does not indicate that PVF-TET cannot affect entry even with the time of addition assay.

Response: Following the reviewer's suggestion, we examined the effect of PVF-tet on the viral mRNA synthesis within one life cycle. As shown in Supplementary Fig. 4b in the 2nd version of the revised manuscript, the relative amount of NP mRNA was markedly increased in a time-dependent manner after IAV infection. Even in a very early stage of infection (after 1 h incubation at 37 °C), PVF-tet did not affect the NP mRNA level, while bafilomycin A1 completely suppressed the

induction. In addition, we also examined the effect of PVF-tet on the fusion process after endocytosis of the virus. We prepared IAV particles labeled with two different lipophilic tracers, C18-Rhodamine B (red fluorescence) and C18-DiO (green fluorescence), which can be detected as red particles because under this condition fluorescence of DiO is quenched by that of Rhodamine B through the Förster resonance energy transfer. After fusion of the virus with the endosome membrane, these two tracers diffuse into the endosome membrane, allowing the endosome to be detected by both fluorescence as a result of DiO dequenching. As shown in Supplementary Fig. 4a in the 2nd version of the revised manuscript, after 90 min incubation at 37 °C co-localization of red- and green-fluoresce was efficiently detected in vehicle and PVF-tet treated cells, but not in bafilomycin A1 treated cells, further indicating that PVF-tet does not affect the binding or entry of the virus. We added Supplementary Figs. 4a and 4b, and accordingly changed the Results (lines 162-164) in the 2nd version of the revised manuscript.

In lines 162-164, specify: did not affect the fusion process, as monitored with fluorescently labeled virus, ...

In my view, this experiment adequately proves that PVF-tet does not act during virus entry.

Response: Following the reviewer's suggestion, we added a sentence in the resubmitted version of the manuscript (lines 166-167) to specify the experiment. We greatly appreciate the reviewer's fair comment.

4) A major contradiction is that PVF-TET can block 100% of virus release in Fig 4E, but the antiviral activity shown in the preceding in vitro and then the mouse experiments is poor. The authors need to explain this contradiction very clearly

Response: To further clarify the inhibitory effect of PVF-tet on the vesicular transport of newly synthesized HA to the plasma membrane, we added low magnification images of cells. As shown Supplementary Fig. 5c left panel in the 2nd version of the revised manuscript, PVF-tet clearly reduced the amount of cell surface HA (green), and concomitantly induced the accumulation of intracellular HA (red) in the vacuole-like structure, indicating that PVF-tet sequestered newly synthesized HA in the structure to hamper its vesicular transport to the plasma membrane. Under the same condition, where MDCK cells were infected with IAV at 10 MOI for 1 h and then cultured with PVF-tet, PVF-tet (20 μ M) efficiently inhibited the IAV cytopathicity to increase the cell viability from $18 \pm 4\%$ to $103 \pm 9\%$, or from $35 \pm 3\%$ to $85 \pm 9\%$ 24 h after infection, as shown in

Fig. 6d or Supplementary Fig. 10c in the 2nd version of the revised manuscript, respectively. This protective effect of PVF-tet is consistent with the marked inhibition of the intracellular transport of newly synthesized HA to the plasma membrane to suppress the viral propagation. We added Supplementary Fig. 5c left panel, and accordingly changed the Results (lines 190-192) in the 2nd version of the revised manuscript. As the reviewer points out, (D)PVF-tet rescued 50% of mice from mortality, while zanamivir rescued 100% mice (Supplementary Fig. 12a in the 2nd version of the revised manuscript), although (D)PVF-tet efficiently inhibited IAV cytopathicity with comparable efficacy to zanamivir in MDCK cells (Fig. 1f in the revised manuscript). The presence of proteases, such as TMRRSS4, trypsin Clara, and HAT in the respiratory tract (Curr Opin Virol. 24, 16-24, 2017) might affect the stability of (D)PVF-tet after intranasal administration to reduce its efficacy in vivo. In addition, as shown in the attached Fig. 2, the efficacy of incorporation of PVF-tet into the target cells can be affected by the presence of serum proteins, so the epithelial environment of the alveolar space, where the alveolar type-II cells, major target cells of IAV infection, are present with large amount of pulmonary surfactant but less amount of serum proteins, might also negatively affect the incorporation of PVF-tet to the alveolar type-II cells. However, the precise mechanisms remains to be clarified.

It looks to me that the authors' long answer does not address the Reviewer's comment (but is there a Figure 4e? Only Fig. 2d shows antiviral activity based on virus yield?). The Reviewer seems critical about how relevant is the antiviral activity of (D)PVF-tet, especially when compared to that of a well-established drug like zanamivir.

I am not sure that there is a contradiction. Fig. 2d shows a fold increase in the virus control of about 600 which is reduced to about 50 under PVF-tet. I thus estimate that PVF-tet causes about one-log₁₀ reduction in virus yield, which is not a very impressive antiviral effect.

In this respect, I agree with the Reviewer that the (D)PVF-tet concept does not result in a very strong antiviral effect.

Response: We agree with the reviewers' comments that (D)PVF-tet is not a very strong antiviral compound compared with a well-established anti-IAV agent such as zanamivir. However, using PVF-tet we could identify the inducible amphisome as a new antiviral concept, which has never been recognized before. Furthermore, following the reviewers' suggestions, we could present the broad applicability of this peptide concept in vitro and in vivo as described below.

5) They screened out the PVF-tet by a binding assay to HA receptor and showed that the PVF-TET can really bind to the HA receptor binding domain. Then they claimed that the PVF-tet did not bind to viral HA, but only bind to newly synthesized intracellular HA. How can this happen at all? They must support their claim by showing structural differences between the HA receptor binding domains between the newly synthesized intracellular HA and that of the HA on released virus? This is a key problem in the study!!!. They paid a lot of attention on the downstream experiments to claim that the PVF-tet only bind to newly synthesized intracellular HA, but they did not clearly verify if the PVF-tet cannot bind to virus and affect viral entry. This must be answered before the paper can be considered for publication. Otherwise, the conclusion can be wrong.

Response: To address the reviewer's concern, we examined the inhibitory effect of PVF-tet on the hemagglutinating activity of HA present on infectious IAV particles. As shown in Supplementary Fig. 4e in the 2nd version of the revised manuscript, PVF-tet did not inhibit the hemagglutinating activity, while fetuin efficiently inhibited the activity in a dose dependent manner. This observation indicates that PVF-tet does not bind to the functioning HA present on IAV particles, although PVF-tet efficiently binds to the functioning recombinant HA and then inhibits its receptor-binding activity (as shown in Fig. 1e in the revised manuscript). The difference is not attributed to the structural difference of the receptor-binding region, because these HAs are functionally equivalent. A major difference between these HAs is the state of HA; HA present on infectious IAV particles is cleaved into HA1 and HA2 fragments, but recombinant HA or newly synthesized intracellular HA is not. Thus, the resultant HA2 fragment might hamper the accessibility and/or binding of PVF-tet to the receptor-binding region of the HA1 fragment present on infectious IAV. Previous studies based on the X-ray crystallography demonstrate that the cleavage of HA does not affect the structure of the receptor-binding region of HA (Science, 303, 1866-1870. 2004; Science, 303, 1838-1842. 2004), further supporting this contention.

Consistently, proteolytic cleavage of recombinant HA into HA1 and HA2 fragments markedly reduced the binding of PVF-tet to HA as shown in Supplementary Fig. 4d in the 2nd version of the revised manuscript. However, the precise mechanism by which the cleavage of HA results in the reduction of the PVF-tet binding to HA remains to be clarified. We added Supplementary Fig. 4e, and accordingly changed the Results (lines 173-175) in the 2nd version of the revised manuscript.

As shown in the response to the reviewer's comment A-3, we performed additional experiments, and showed that PVF-tet did not affect the binding and fusion of IAV or also NP mRNA level even in a very early stage of infection.

This is to me an extremely valid Reviewer comment that must be solved.

The text is indeed very confusing and unclear regarding the interaction between PVF-tet and HA.

PVF-tet was identified by an HA binding assay (with recombinant HA0?). The compound is not active in the hemagglutination assay (with virus in which presumably most HA is present under the form of HA1/HA2 although we are not sure about this) but is, strangely, active in Fig. 1e showing a binding assay with recombinant HA0 (???) and a receptor mimic.

The authors are right that the receptor binding site is not different between HA0 and HA1/HA2. Thus, they hypothesize that PVF-tet cannot bind to cleaved HA1/HA2 protein whereas it does bind to uncleaved HA0. This assumption could easily be verified by SPR, comparing binding of the recombinant protein to either the HA0 or cleaved (= treated with trypsin) HA1/HA2 form.

If the hypothesis proves to be correct, how can it be explained? It is very hard to rationalize that a compound binding at a stem region (cfr many anti-HA stem antibodies) could interfere with receptor binding.

I agree with the Reviewer that this is a serious lack in the study.

Response: Recombinant HA0 was used for the screening and the characterization of the identified peptides (Figs. 1a, 1d, 1e, and 2a, and Supplementary Figs. 1 and 2). We added this information in the Methods (lines 503, 527, 565-566, 572 and 586) in the resubmitted version of the manuscript.

Following the reviewers' suggestions, we performed additional experiments to further define the interaction between HA and PVF-tet. We showed that the binding of PVF-tet to HA was reduced by the cleavage of HA by trypsin (Supplementary Fig. 5a, middle panel). We found that PVF-tet did not bind to infectious IAV particles with HA1/HA2 (PR8-HA1/HA2) but did bind with remarkable potency to non-infectious IAV particles with HA0 (PR8-HA0), clearly indicating that the binding of PVF-tet to virus particles depends on the state of HA on the particles (Supplementary Fig. 5b, middle panel).

As the reviewers point out, the important question is why PVF-tet efficiently binds to HA0 but not to HA1/HA2. To answer that question, we examined the effect of HA cleavage on the receptor-binding activity of fetuin, a highly sialylated glycoprotein. As shown in Supplementary Fig. 5a, the binding of fetuin to HA was also reduced by the cleavage of HA, although to a substantially lesser degree than the binding of PVF-tet. Consistently, fetuin exhibited much more binding to PR8-HA0 than to infectious PR8-HA1/HA2, whereas asialofetuin exhibited no binding to either virus (Supplementary Fig. 5b, right panel). The difference in the sialic acid-dependent binding of fetuin to PR8-HA0 and PR8-HA1/HA2 cannot be attributed to a structural difference in the receptor-binding site of HA (HA-RBS); which consists of a 130-loop, a 150-loop, a 190-helix including L194, and a 220-loop; because previous studies showed that the cleavage of HA does not affect the structure of the HA-RBS (Supplementary Fig. 6a; Science, 303, 1866-1870. 2004; Science, 303, 1838-1842. 2004). Consistent with that observation, HA0 and HA1/HA2 displayed equally

low solvent accessibility in the HA-RBS (Supplementary Fig. 6b, right panels). However, the crystallographic B-factors, which reflect atomic fluctuation and provide information about protein dynamics, of the HA0-RBS structure were higher than those of the HA1/HA2-RBS structure (Supplementary 6b, left panels). That observation was confirmed by a similar analysis using average B-factors based on the structural information of various H1 and H3 subtypes (Supplementary Fig. 6c). All of these results imply that the higher atomic fluctuation of the HA0-RBS compared with that of the HA1/HA2-RBS leads to a more efficient interaction with sialic acid or PVF-tet, although the crystal structures of the HA0-RBS and the HA1/HA2-RBS are the same. The cleavage of HA changes the conformation of the RBS to one with lower atomic fluctuation, which PVF-tet can hardly access, whereas the binding of sialic acid is less susceptible to the conformational change.

We added Supplementary Figs. 5 and 6 and changed the text of the Results section (lines 173-183) in the resubmitted version of the manuscript.

6) The antiviral efficacy of PVF-TET was really not good, only protected 50% of H1N1- infected mice when PVF-TET was administrated at 6 h post infection. There are many published antiviral compounds and antiviral peptides which are better than this protection and put doubt on the significance of this findings.

Response: Almost all the synthetic HA inhibitors developed so far, including sialic acid polymers or HA-binding peptides (ref. 12-14 in the 2nd version of the revised manuscript; *Biochemistry*, 32, 2967-2978, 1993; *Virology*, 226, 66-76, 1996; *Virology*, 248, 264-274, 1998; *J Virol.*, 80, 11960-11967, 2006; *Int J Biol Sci.*, 5, 543-548, 2009; *J Virol.*, 84, 4277- 4288, 2010; *J Med Chem.*, 57, 8332-8339, 2014; *J Virol.*, 88, 1447-1460, 2014), are targeting HA present on infectious IAV particles to inhibit viral binding and/or fusion process. PVF-tet is the first compound that targets newly synthesized HA and inhibits viral propagation in vitro and in vivo through a unique mechanism involving autophagy induction. Furthermore, based on the unique mechanism underlying the function of PVF-tet, we identified another type of anti-viral amphisome, which can be induced even in the absence of PVF-tet. We believe that these findings could be useful to provide a new strategy to combat IAV infection.

It seems not totally fair to compare a new antiviral concept like PVF-tet (which leads to HA sequestering in amphisomes) with more established small molecule concepts in the literature. In Fig. 7e, there is less body weight loss in mice receiving the peptide and virus together; these mice have a survival rate of 50%.

The authors do not really present (D)PVF-tet as a strong antiviral compound, rather they use the molecule to identify the inducible amphisome as a new antiviral concept. I agree with this presentation.

Response: We greatly appreciate the reviewer's fair comment.

7) Moreover some of the experimental design are not reasonable. Note that if the authors cannot claim any new antiviral mechanism and show no data of broad protection in mice infected by other influenza virus, it would be doubtful if the work can be considered for publication in a reputable journal. I am sorry that I cannot be helpful in finding a way out for the authors on this occasion.

Response: To apply the unique anti-viral concept identified here to another type of influenza virus, it is necessary to develop a tetravalent peptide customized to HA of each intended virus. We have already found that PVF-tet bound to recombinant HA derived from highly virulent avian H5N1 A/chicken/Yamaguchi/7/2004 strain, which exclusively binds to α 2-3 sialyllactose-polymer, and also to its human-tropic mutant with two mutations (Q226L and G228S), which exclusively binds to α 2-6 sialyllactose-polymer. Thus, it is highly possible that based on the HA-binding motif of PVF-tet, binding motifs customized to HA of H5N1 virus could be developed by using affinity maturation-based screening of tetravalent peptides synthesized on a membrane. On the other hand, we have already found that human-tropic HA of H5N1 virus, which was overexpressed in MDCK cells stably overexpressing ABCA3, was also accumulated in ABCA3-positive structure as shown in the attached Fig. 3 (confidential data), suggesting that the strategy to stimulate the formation of the anti-viral amphisomes might be also useful to inhibit the propagation of another type of influenza virus. In this paper, however, we would like to focus on H1N1 virus, because we believe that the significance of our findings is to propose a novel concept to combat IAV infection.

See my comment above regarding 'strong' antiviral effect.

Regarding broad activity against more than one (not only H1N1) influenza virus, the authors added a confidential Figure 3 to show amphisome localization for a human-tropic H5 HA. Why are they so reluctant to mention the data in the manuscript?

I agree with the Reviewer that adding (D)PVF-tet activity data for another virus subtype (which I prefer to experiments with HA only) would make the study much more significant. I would be satisfied with in vitro data.

In the answer above, the authors explain that PVF-test does bind to H5 HA, so it should also be active in an antiviral assay with H5N1 virus?

Response: Following the reviewer's suggestion, we performed additional experiments to show the broad applicability of the peptide concept. As shown in Supplementary Fig. 16 in the resubmitted version of the manuscript, PVF-tet reduced the cytopathicity of other IAV strains; mouse-adapted A/California/04/2009 (H1N1 pdm), A/Tokyo/UTHP013/2016 (H1N1 pdm), and mouse-adapted A/Aichi/2/1968 (H3N2); through the formation of inducible amphisomes, all of which contained newly synthesized HA and PVF-tet. Furthermore, (D)PVF-tet rescued mice from the lethality of two mouse-adapted IAV strains, A/California/04/2009 and A/Aichi/2/1968 (H3N2), as shown in Fig. 7c in the resubmitted version of the manuscript. Those results clearly demonstrate the broad applicability of the peptide concept. We added Fig. 7c and Supplementary Fig. 16 and accordingly changed the Results and the Methods (lines 345-350, and 714-717, respectively) in the resubmitted version of the manuscript.

Based on the data obtained using human-tropic HA of H5N1 virus, PVF-tet is expected to be active in an antiviral assay with human-tropic H5N1 virus, but we have not tested that hypothesis experimentally because of the limitation of the biosafety level in our research facility.

We greatly appreciate the reviewers' constructive suggestions, which strengthen the new strategic concept to combat IAV infections.

Summary analysis

I agree with Reviewer #1's criticisms on the confusing information regarding the precise PVF-tet/HA interaction and broad applicability of this peptide concept. The authors must solve these issues, particularly the first criticism.

Some other remarks (e.g., the peptide is not as potent as some other reported inhibitors) are a lesser concern to me since the focus of the manuscript is on presenting a new concept.

Some confusing information regarding some experiments can be solved by adequate presentation of the Figures (e.g. by adding next to the Figures a visual of the experimental setup).

Reviewers' comments:

Reviewer #2 (Remarks to the Author):

No further comments.

Reviewer #4 (Remarks to the Author):

This manuscript reports the development and characterization of a multivalent, peptide-based inhibitor of influenza A virus, PVF-tet. The authors find that treatment of IAV infected cells with PVF-tet leads to the accumulation of HA in vacuole-like compartments within the cell. The sequestration of HA in these compartments leads to a reduction in shed virus, as well as an increase in cell survival. The authors also show that similar sequestration of HA can occur via other means, including the overexpression of ABCA3, a protein involved in surfactant production and the formation of lamellar bodies. By co-localizing proteins associated with autophagosomes (LC3-II, p62) with these structures and by showing that their formation is blocked by inhibitors of autophagy, the authors conclude that HA is being sequestered within amphisomes, compartments formed by the fusion of endosomal vesicles with autophagosomes.

The observations reported in this manuscript – in particular, that a peptide which enters into the cells and binds multivalently to HA can lead to its sequestration in amphisomes and promote cell survival – is novel and will be of interest to researchers in the areas of anti-viral therapeutics, virus replication, and innate immunity. A limitation of this manuscript is the lack of mechanistic insight into how PVF-tet interacts with HA to drive its sequestration, and how this sequestration specifically enhances cell survival.

To address the first of these limitations, the authors have performed new experiments and analysis to try to understand the interesting observation that PVF-tet binds with much higher affinity to HA0 than to HA1/HA2. This observation is supported in the revised manuscript by the addition of supplemental figure 5, which shows that the difference in binding to HA0 versus HA1/HA2 is not unique to PVF-tet, but also applies to sialic acid (measured using fetuin). This appears to be the case both for recombinant HA, as well as for HA on the surface of virus particles. This suggests that differential affinity for the receptor binding site in HA0 versus HA1/HA2 may be a general phenomenon, and makes supplemental figure 5 a strong addition to the paper.

To understand what the underlying cause for the observed differences in binding might be, the authors provide a plausible hypothesis: although the HA receptor binding site structure is unaffected by cleavage into HA1/HA2, fluctuations in the receptor binding site may be dampened in HA1/HA2, making it harder for both PVF-tet and sialic acid to bind. While I believe that this is a reasonable model, the evidence offered in support - based on a comparison of B-factors across different HA structures - is fairly unconvincing. Comparing B-factors across structures of different HAs (often with different resolutions) is non-trivial, and I am not convinced that it can be done in this case in a meaningful way. Although some normalization is surely necessary to make the comparison, I am not sure that subtracting the overall median B-factor for the entire structure is the appropriate way to do this. For example, if cleavage into HA1/HA2 increased flexibility in the membrane proximal region of HA in a way that was reflected in increased B-factor in this region, then subtracting the median value from the overall structure could artificially suppress the apparent flexibility in the vicinity of the receptor binding site. This example may or may not be relevant to the specific comparisons in supplemental figure 6, but it illustrates one of several possible pitfalls in the approach. At a minimum, these pitfalls need to be discussed.

As it stands now, the author's description of this analysis (e.g. page 9, line 180) arrives at too strong a conclusion for the evidence they present. Such definitive statements would require that the authors obtain more direct evidence of a difference in flexibility (for example, using hydrogen exchange mass spectrometry).

Additional comments:

1) How does sequestration of HA promote cell survival following high MOI (~10) infection? If the standard definition of MOI as fully-infectious particles (i.e. pfu) per cell is used here (and the authors should indicate if this is the case), the infectious burden on these cells would be quite high, and yet figure 2 shows 80-100% survival at 24 hours post infection. This is the case despite the fact that the synthesis and localization of other viral proteins (including NP, M2, and NA, which have previously been shown to modulate IAV-induced cell death – see for example 10.1016/j.chom.2009.09.005; 10.1038/cddis.2013.89) appears to be unaffected. This seems to suggest that the shedding of virus or the cell surface localization of HA may be direct contributors to cell death. If so, this is an interesting observation that is fairly central to the focus of this paper, yet there is no specific discussion of how PVF-tet and HA sequestration may be contributing to cell survival.

2) A direct comparison of low vs. high MOI infections (or better yet, a full growth curve) would strengthen the paper. As it stands, the action of PVF-tet cannot easily be disentangled from the effects of trypsin and FBS. I understand the technical limitations that make this difficult (FBS inhibits trypsin and also enhances uptake of PVF-tet), but there are ways around this. For example, performing experiments using a virus like A/WSN/33 in which HA can be cleaved by plasmin from

FBS in the absence of trypsin (see for example 10.1073/pnas.95.17.10224) would be one way to do this.

3) The description of PVF-tet as being cell-permeable (e.g. page 10, line 215) is confusing to me, since it primarily appears to enter the cell via endocytosis or macropinocytosis after binding to the cell surface (figure 3e, supplementary figure 8). In this case, it would remain topologically on the side of the membrane opposite the cytoplasm. In fact, it seems like it would be counterproductive if it were able to passively enter the cytoplasm, since it would diminish its ability to bind to the HA extracellular domain.

4) Although the authors discuss other recent developments in peptide-based inhibitors of IAV that target HA, they omit a relevant reference (10.1038/nbt.3907). In this work, Strauch et al. demonstrate two features which the author's claim have previously been uncharacterized: multivalent attachment to HA (page 3, lines 52-53) and in vivo efficacy (page 17, line 384).

Response to comments from Reviewer #4:

We would like to thank Reviewer #4 for considering our manuscript and offering constructive comments that help strengthen our conclusions. We have provided point-by-point responses to the Reviewer's concerns below.

Reviewers' comments:

Reviewer #4 (Remarks to the Author):

This manuscript reports the development and characterization of a multivalent, peptide-based inhibitor of influenza A virus, PVF-tet. The authors find that treatment of IAV infected cells with PVF-tet leads to the accumulation of HA in vacuole-like compartments within the cell. The sequestration of HA in these compartments leads to a reduction in shed virus, as well as an increase in cell survival. The authors also show that similar sequestration of HA can occur via other means, including the overexpression of ABCA3, a protein involved in surfactant production and the formation of lamellar bodies. By co-localizing proteins associated with autophagosomes (LC3-II, p62) with these structures and by showing that their formation is blocked by inhibitors of autophagy, the authors conclude that HA is being sequestered within amphisomes, compartments formed by the fusion of endosomal vesicles with autophagosomes.

The observations reported in this manuscript – in particular, that a peptide which enters into the cells and binds multivalently to HA can lead to its sequestration in amphisomes and promote cell survival – is novel and will be of interest to researchers in the areas of anti-viral therapeutics, virus replication, and innate immunity. A limitation of this manuscript is the lack of mechanistic insight into how PVF-tet interacts with HA to drive its sequestration, and how this sequestration specifically enhances cell survival.

To address the first of these limitations, the authors have performed new experiments and analysis to try to understand the interesting observation that PVF-tet binds with much higher affinity to HA0 than to HA1/HA2. This observation is supported in the revised manuscript by the addition of supplemental figure 5, which shows that the difference in binding to HA0 versus HA1/HA2 is not unique to PVF-tet, but also applies to sialic acid (measured using fetuin). This appears to be the case both for recombinant HA, as well as for HA on the surface of virus particles. This suggests that differential affinity for the receptor binding site in HA0 versus HA1/HA2 may be a general phenomenon, and makes supplemental figure 5 a strong addition to the paper.

To understand what the underlying cause for the observed differences in binding might be, the authors provide a plausible hypothesis: although the HA receptor binding site structure is unaffected by cleavage into HA1/HA2, fluctuations in the receptor binding site may be dampened in HA1/HA2, making it harder for both PVF-tet and sialic acid to bind. While I believe that this is a reasonable model, the evidence offered in support - based on a comparison of B-factors across different HA structures - is fairly unconvincing. Comparing B-factors across structures of different HAs (often with different resolutions) is non-trivial, and I am not convinced that it can be done in this case in a meaningful way. Although some normalization is surely necessary to make the comparison, I am not sure that subtracting the overall median B-factor for the entire structure is the appropriate way to do this. For example, if cleavage into HA1/HA2 increased flexibility in the membrane proximal region of HA in a way that was reflected in increased B-factor in this region, then subtracting the median value from the overall structure could artificially suppress the apparent flexibility in the vicinity of the receptor binding site. This example may or may not be relevant to the specific comparisons in supplemental figure 6, but it illustrates one of several possible pitfalls in the approach. At a minimum, these pitfalls need to be discussed.

Response: In response to the Reviewer's comment that comparing B-factors across structures of different HAs is non-trivial, we compared the overall structures of HA0 and HA1/HA2 derived from H1HA or H3HA separately, as shown in Supplementary Fig. 6. For H1HA, the crystal structure of HA0 from strain A/Brevig Mission/1/1918 (PDB: 1RD8) was compared with those of HA1/HA2 from strains A/Puerto Rico/8/1934 (PDB: 1RU7) and A/California/04/2009 (PDB: 3LZG). For H3HA, the structure of HA0 from strain A/Aichi/2/1968 (PDB: 1HA0) was compared with that of HA1/HA2 from the same strain (PDB: 3VUN and 2VIU). Importantly, in addition to H1HA from strains A/Puerto Rico/8/1934, H1HA from A/California/04/2009 and H3HA from H3N2 strain A/Aichi/2/1968 were shown to form inducible amphisomes in the presence of PVF-tet after infection, as described in Supplementary Fig. 16. Consistent with these observations, in all cases, the normalized B-factors in the HA0-RBS structure were higher than those in the HA1/HA2-RBS structure, where the normalization of B-factors was performed based on the conventional "z-score normalization" (Acta Cryst. D, 70, 2413-9, 2014). This finding implies that the higher atomic fluctuation of HA0 RBS compared with that of HA1/HA2 RBS leads to a more efficient interaction with PVF-tet (Supplementary Fig. 5) even though the crystal structures of HA0 RBS and HA1/HA2 RBS are the same. As the Reviewer points out, we cannot rule out the possibility that, after normalization, the relatively higher fluctuation seen in the membrane proximal region (MPR) of HA1/HA2 is reflected by the suppression of apparent flexibility in the vicinity of the RBS, as observed in 1RU7. However, in 3LZG, where the fluctuation level in the MPR is lower than that of 1RU7, the fluctuation level of the RBS is even lower than that of 1RU7, substantially weakening the possibility.

As it stands now, the author's description of this analysis (e.g. page 9, line 180) arrives at too strong a conclusion for the evidence they present. Such definitive statements would require that the authors obtain more direct evidence of a difference in flexibility (for example, using hydrogen exchange mass spectrometry).

Response: In response to the Reviewer's suggestion, we have revised the text of the Results section (lines 176–181), as shown in the resubmitted version of the manuscript.

Additional comments:

1) How does sequestration of HA promote cell survival following high MOI (~10) infection? If the standard definition of MOI as fully-infectious particles (i.e. pfu) per cell is used here (and the authors should indicate if this is the case), the infectious burden on these cells would be quite high, and yet figure 2 shows 80-100% survival at 24 hours post infection. This is the case despite the fact that the synthesis and localization of other viral proteins (including NP, M2, and NA, which have previously been shown to modulate IAV-induced cell death – see for example 10.1016/j.chom.2009.09.005; 10.1038/cddis.2013.89) appears to be unaffected. This seems to suggest that the shedding of virus or the cell surface localization of HA may be direct contributors to cell death. If so, this is an interesting observation that is fairly central to the focus of this paper, yet there is no specific discussion of how PVF-tet and HA sequestration may be contributing to cell survival.

Response: As per the Reviewer's suggestion, we defined multiplicity of infection (MOI) as the number of fully infectious particles (i.e., pfu) per cell in the resubmitted version of the manuscript (lines 500-501). As the Reviewer mentions, in A549 lung epithelial cells, M2 enhances IAV-induced cell death by inhibiting macroautophagy (Cell Host Microbe, 6, 367-380, 2009). NP induces cell death by interacting with Clusterin, a negative regulator of Bax, to free and activate Bax, thus resulting in efficient virus replication (Cell Death Dis., 4, e562, 2013). In MDCK cells, after the transport of newly synthesized HA to the plasma membrane, HA has been reported to bind to sulfatide, one of the major sulfated glycolipids present in abundance in lipid rafts of the plasma membrane to induce apoptosis (J. Virol., 82, 5940-5950, 2008; PLoS One, 8, e61092, 2013). Blocking the formation of this complex with anti-sulfatide or anti-HA antibody effectively inhibits virus amplification, suggesting that the interaction of HA with sulfatide on the plasma membrane is primarily involved in virus-induced cell death in MDCK cells. Thus, HA sequestration induced by PVF-tet or ABCA3 overexpression can block this pathway to promote cell survival. Following the Reviewer's suggestion, we discussed this point in the resubmitted version of the manuscript (lines 417–424).

2) A direct comparison of low vs. high MOI infections (or better yet, a full growth curve) would strengthen the paper. As it stands, the action of PVF-tet cannot easily be disentangled from the effects of trypsin and FBS. I understand the technical limitations that make this difficult (FBS inhibits trypsin and also enhances uptake of PVF-tet), but there are ways around this. For example, performing experiments using a virus like A/WSN/33 in which HA can be cleaved by plasmin from FBS in the absence of trypsin (see for example [10.1073/pnas.95.17.10224](https://doi.org/10.1073/pnas.95.17.10224)) would be one way to do this.

Response: Following the Reviewer's suggestion, we directly compared the effect of PVF-tet on cytopathicity induced by low or high MOI infection in Caco-2 cells. Studies have shown that, in Caco-2 cells, IAV can efficiently replicate even in the presence of FBS probably due to the presence of endogenous proteases that cleave HA0 to produce infectious viruses with HA1/HA2 (Virology, 313, 198-212, 2003). As shown in Supplementary Fig. 8d, PVF-tet equally inhibited the cytopathicity induced by both low (0.001 MOI for 96 h) and high MOI (10 MOI for 72 h) under the same conditions with 10% FBS, indicating that the anti-viral activity of PVF-tet is independent of the MOI. We added the data and mentioned this point in the resubmitted version of the manuscript (lines 212–215).

3) The description of PVF-tet as being cell-permeable (e.g. page 10, line 215) is confusing to me, since it primarily appears to enter the cell via endocytosis or macropinocytosis after binding to the cell surface (figure 3e, supplementary figure 8). In this case, it would remain topologically on the side of the membrane opposite the cytoplasm. In fact, it seems like it would be counterproductive if it were able to passively enter the cytoplasm, since it would diminish its ability to bind to the HA extracellular domain.

Response: As the Reviewer mentions, PVF-tet is incorporated into cells via the endocytic pathway because the incorporation was observed at 37°C but not at 4°C. Then PVF-tet co-localized with dextran (Supplementary Fig. 8b). Following the Reviewer's suggestion, we revised “cell-permeable” (lines 205, 211 and 216) to “cell-penetrating” in the resubmitted version of the manuscript.

4) Although the authors discuss other recent developments in peptide-based inhibitors of IAV that target HA, they omit a relevant reference ([10.1038/nbt.3907](https://doi.org/10.1038/nbt.3907)). In this work, Strauch et al. demonstrate two features which the author's claim have previously been uncharacterized: multivalent attachment to HA (page 3, lines 52-53) and in vivo efficacy (page 17, line 384).

Response: Following the Reviewer's comment, we cited the reference (10.1038/nbt.3907) and revised the Discussion section accordingly (lines 380–385) in the resubmitted version of the manuscript.

REVIEWERS' COMMENTS:

Reviewer #4 (Remarks to the Author):

In response to the previous review, the authors have revised their presentation and discussion of the B-factor analysis (Supplemental Figure 6), performed additional experiments to measure cell viability at high and low MOI under matching conditions (10% FBS and no trypsin; Supplemental Figure 8d), and clarified points in the discussion which I found incomplete or confusing. The primary conclusions of the paper have now been demonstrated and presented clearly enough, even if more work will be needed to shed light on the underlying mechanisms.

Response to comments from Reviewer #4:

Reviewers' comments:

Reviewer #4 (Remarks to the Author):

In response to the previous review, the authors have revised their presentation and discussion of the B-factor analysis (Supplemental Figure 6), performed additional experiments to measure cell viability at high and low MOI under matching conditions (10% FBS and no trypsin; Supplemental Figure 8d), and clarified points in the discussion which I found incomplete or confusing. The primary conclusions of the paper have now been demonstrated and presented clearly enough, even if more work will be needed to shed light on the underlying mechanisms.

Response: We would like to thank the reviewer for reviewing our manuscript and offering constructive comments.